# BLESSING FROM EXPERTS: SUPER REINFORCEMENT LEARNING IN CONFOUNDED ENVIRONMENTS

## ABSTRACT

We introduce *super reinforcement learning* in the batch setting, which takes the observed action as input for enhanced policy learning. In the presence of unmeasured confounders, the recommendations from human experts recorded in the observed data allow us to recover certain unobserved information. Including this information in the policy search, the proposed super reinforcement learning will yield a *super-policy* that is guaranteed to outperform both the standard optimal policy and the behavior one (e.g., the expert's recommendation). Furthermore, to address the issue of unmeasured confounding in finding super-policies, a number of non-parametric identification results are established. Finally, we develop two super-policy learning algorithms and derive their corresponding finite-sample regret guarantees.

## 1 INTRODUCTION

Offline reinforcement learning (RL) aims to find a sequence of optimal policies by leveraging the batch data (Sutton & Barto, 2018; Levine et al., 2020). In many high-stake domains such as medical studies (Kosorok & Laber, 2019), it is very costly or dangerous to interact with the environment for online data collection, and learning must rely entirely on pre-collected observational or experimental data. Recently, there is a surging interest in studying offline RL theories and methods. Most existing solutions rely on the unconfoundedness assumption that excludes the existence of latent variables that confound the action-reward/-next-observation associations. However, in practice we often encounter unmeasured confounding, under which most existing RL algorithms will lead to sub-optimal polices.

In this paper, we study offline policy learning in *confounded* contextual bandits and sequential decision making. Existing works on policy learning focused on searching an optimal policy that purely depends on the past history, ignoring the recommended action given by the human expert in the observed data. In many applications, there is a common belief that human decision-makers have access to important information that is not recorded in the observed data when taking an action (Kleinberg et al., 2018). For example, in the urgent care, clinicians leverage visual observations or communications with patients to recommend treatments, where such unstructured information is hard to quantify and often not recorded (McDonald, 1996). Another motivating example is given by the deep brain stimulation (DBS Lozano et al., 2019). Due to recent advances in DBS technology, it becomes feasible to instantly collect electroencephalogram data, based on which we are able to provide adaptive stimulation to specific regions in the brain so as to treat patients with neurological disorders including Parkinson's disease, essential tremor, etc. In these applications, the patient is allowed to determine the behavior policy (e.g., when to turn on/off the stimulation, for how long, etc) based on information only known to herself (e.g., how she feels), therefore generating batch data with unmeasured confounders. We notice that despite challenges in policy learning with latent confounders, human recommendations may capture certain unobserved information as discussed in aforementioned applications. Including this information as input of the policy can enhance policy learning, which is indeed "a blessing from experts". Therefore, in this paper, we ask

**Is it possible to consistently learn an optimal policy that takes both the data history and human recommendation at the current time as input for better decision making?**

We will answer the above question affirmatively. Specifically, we first introduce a novel framework called super RL, which compared with the standard RL additionally takes the human's recommendation as input for policy learning. In confounded environments, super RL can embrace the blessing

from experts. In other words, it leverages the human expertise in discovering unobserved information for enhanced policy learning. The resulting policy, which we call *super-policy*, is guaranteed to outperform the standard optimal one learned from without using the human expertise and the behavior policy that may depend on the hidden state. To implement the proposed super-policy for decision making in the future, we require the human expert to recommend an action at each time, which is commonly seen in practice. The super-policy then takes this action and other observations as input and override the recommendation produced by the expert. Second, to address the challenge of partial observability or unmeasured confounding, we establish several non-parametric identification results in finding these super-policies in various confounded environments, leveraging the recent development in causal inference (Tchetgen Tchetgen et al., 2020). Notably, our identification results prove that the super-policy is learnable from the observed data despite the presence of unmeasured confounding. Finally, we develop two super RL algorithms and derive the corresponding finite-sample regret guarantees that are polynomial in terms of all relevant parameters in finding a desirable super-policy.

## 2 RELATED WORK

There is an increasing interest in studying off-policy evaluation (OPE) and learning in sequential decision making problem with unmeasured confounding. Specifically, Zhang & Bareinboim (2016) introduced the causal RL framework and the confounded Markov decision process (MDP) with memoryless unmeasured confounding, under which the Markov property holds in the observed data. Along this direction, many OPE and learning methods are proposed using instrumental or mediator variables (Chen & Zhang, 2021; Liao et al., 2021; Li et al., 2021; Wang et al., 2021; Shi et al., 2022; Fu et al., 2022; Yu et al., 2022). In addition, partial identification bounds for the off-policy's value have been established based on sensitivity analysis (Namkoong et al., 2020; Kallus & Zhou, 2020; Bruns-Smith, 2021). Another streamline of research focuses on general confounded POMDP models to allow for both unmeasured confounding and partial observability. Several point identification results were established (Tennenholtz et al., 2020; Bennett & Kallus, 2021; Nair & Jiang, 2021; Shi et al., 2021; Ying et al., 2021; Miao et al., 2022). However, none of the aforementioned works study policy learning with the help of human expertise, i.e., taking recommended action in the observed data for decision making. Different from these works, we tackle the policy learning problem from a unique perspective and propose a novel super RL framework by leveraging human expertise in discovering certain unobserved information to further improve decision making. We also rigorously establish the super-optimality of the proposed super-policy over the standard optimal policy and the behavior policy. Our paper is also related to a line of works on policy learning and evaluation with partial observability using spectral decomposition and predictive state representation related methods (see e.g., Littman & Sutton, 2001; Song et al., 2010; Boots et al., 2011; Hsu et al., 2012; Singh et al., 2012; Anandkumar et al., 2014; Jin et al., 2020; Cai et al., 2022; Lu et al., 2022; Uehara et al., 2022a;b). Nonetheless, these methods require the no-unmeasured-confounders assumption.

Finally, our proposal is motivated by the work of Stensrud & Sarvet (2022) that introduced the concept of superoptimal treatment regime in contextual bandits. They used an instrumental variable approach for discovering such regime. However, their method can only be applied in a restrictive single-stage decision making setting with binary actions. In contrast, our super-RL framework is generally applicable to both confounded contextual bandits and sequential decision making allowing arbitrarily many actions. It is also worth mentioning that the proposed super RL differs from the recently proposed safe RL via human intervention (Saunders et al., 2017), where human intervention is performed to override bad actions recommended by the intelligent agent. We aim to leverage the human expertise in the previously collected data for intelligent agents to make better decisions.

## 3 SUPER RL: A CONTEXTUAL BANDIT EXAMPLE

In this section, we introduce the super-policy in confounded contextual bandits (e.g., single-stage decision making with unmeasured confounders). Consider a random tuple $(S, U, A, \{R(a)\}_{a \in \mathcal{A}})$, where $S$ and $U$ denote the observed and unobserved features respectively, $A$ denotes the action whose space is given by a finite set $\mathcal{A}$, and $\{R(a)\}_{a \in \mathcal{A}}$ denotes a set of the potential/counterfactual rewards under $A = a$, representing the reward that the agent would receive had action $a$ been taken. The observed reward, denoted by $R$, can then be written as $R = \sum_{a \in \mathcal{A}} R(a)\mathbb{I}(A = a)$.

Table 1: Policy values under different choices of $\epsilon$ in the toy example. In general, $\mathcal{V}(\pi_b) = 0.6 - 1.2\epsilon$, $\mathcal{V}(\pi^*) = 0.4$, $\mathcal{V}(\nu^*) = |0.7 - \epsilon| + |\epsilon - 0.3|$. Bold values are the largest under different settings.

| Policy Value | $\mathcal{V}(\pi_b)$ | $\mathcal{V}(\pi^*)$ | $\mathcal{V}(\nu^*)$ |
|---|---|---|---|
| $\epsilon = 0.5$ | 0.0 | **0.4** | **0.4** |
| $\epsilon = 0$ | 0.6 | 0.4 | **1.0** |
| $\epsilon = 1$ | -0.6 | 0.4 | **1.0** |

Denote the spaces of $S$ and $U$ by $\mathcal{S}$ and $\mathcal{U}$ respectively. Let $\pi : \mathcal{S} \to \mathcal{P}(\mathcal{A})$ denote a policy depending only on the observed information $S$, where $\mathcal{P}(\mathcal{A})$ refers to the class of all probability distributions over $\mathcal{A}$. In particular, $\pi(a \mid s)$ refers to the probability of choosing an action $a$ given that $S = s$. In the batch setting, we are given i.i.d. copies of $(S, A, R)$, where the action $A$ is generated by some behavior policy $\pi^b : \mathcal{S} \times \mathcal{U} \to \mathcal{P}(\mathcal{A})$ that depends on both observed and unobserved features. Since $U$ is unobserved, nearly all existing solutions focused on finding an optimal policy $\pi^*$ given by

$$\pi^*(a^* \mid s) = 1 \quad \text{if} \quad a^* = \text{argmax}_{a \in \mathcal{A}} \mathbb{E}\left[R(a) \mid S = s\right] \quad \forall s \in \mathcal{S}, \tag{1}$$

assuming the uniqueness of the maximization in equation 1 for every $s \in \mathcal{S}$. In addition, notice that $U$ may confound the causal relationship of the action-reward in the observational data. Ignoring this latent confounder will produce a biased estimator of $\pi^*$.

As discussed earlier, in this paper, we aim to find an optimal policy that leverages the input of human expertise, since actions generated by the behavior policy depend on the latent information. In particular, we search a *super-policy* $\nu^*$ in a larger policy class $\Omega = \{\nu : \mathcal{S} \times \mathcal{A} \to \mathcal{P}(\mathcal{A})\}$ such that

$$\nu^*(a^* \mid s, a') = 1 \quad \text{if} \quad a^* = \text{argmax}_{a \in \mathcal{A}} \mathbb{E}\left[R(a) \mid S = s, A = a'\right] \quad \forall (s, a') \in \mathcal{S} \times \mathcal{A}. \tag{2}$$

The two optimal optimal policies are equivalent when unconfoundedness assumption holds. When this condition is violated, $\mathbb{E}\left[R(a) \mid S = s, A = a'\right] \neq \mathbb{E}\left[R(a) \mid S = s\right]$ in general. More importantly, it follows from Proposition 1 of Stensrud & Sarvet (2022) that the value under $\nu^*$ is no worse and often larger than that under $\pi^*$. This yields the super-optimality of $\nu^*$ over $\pi^*$. It is also worth mentioning that in the presence of latent confounders, there is *no* guarantee that the standard optimal policy $\pi^*$ outperforms the behavior policy $\pi^b$ because $\pi^b$ depends on the unobserved information. To the contrary, since $\pi^b \in \Omega$, the proposed super-policy is always better than $\pi^b$. Specifically, let $\mathcal{V}(\nu)$ be the value under the intervention of a generic policy $\nu$, i.e., $\mathcal{V}(\nu) = \sum_{a \in \mathcal{A}} \mathbb{E}[R(a)\nu(a \mid S, A)]$. We have the following lemma that demonstrates the super-optimality of $\nu^*$ over both $\pi^*$ and $\pi^b$.

**Lemma 3.1** (Super-Optimality). $\mathcal{V}(\nu^*) \geq \max\{\mathcal{V}(\pi^b), \mathcal{V}(\pi^*)\}$.

Intuitively speaking, the super-optimality of $\nu^*$ comes from the use of unobserved information $U$ contained in $\pi^b$. We consider the following toy example to elaborate.

**Toy Example**: Assume $S$ and $U$ independently follow a Bernoulli distribution with success probability 0.5. Suppose the action is binary and the behavior policy satisfies $\mathbb{P}(A = 1|S, U = 1) = \mathbb{P}(A = 0|S, U = 0) = 1 - \varepsilon$ for some $0 \leq \varepsilon \leq 1$. Let $R = 8(A - 0.5)(S - 0.2)(U - 0.3)$. In this example, the parameter $\varepsilon$ measures the degree of unmeasured confounding. When $\varepsilon = 0.5$, the behavior policy does not depend on $U$ and the no-unmeasured-confounders assumption is automatically satisfied. Otherwise, this condition is violated. In particular, when $\varepsilon = 0$ or 1, we can fully recover the latent confounder based on the recommended action. Table 1 summarizes the policy values of $\pi^b$, $\pi^*$ and $\nu^*$ under different $\varepsilon$, in which the super-optimality holds.

Despite its appealing property, it is generally impossible to learn the super-policy $\nu^*$ without any further assumptions, since the counterfactual effect $\mathbb{E}\left[R(a) \mid S = s, A = a'\right]$ is not identifiable from the observed data due to unmeasured confounding. Toward that end, we adopt the proximal causal inference framework developed by Tchetgen Tchetgen et al. (2020). Specifically, we assume the existence of certain action and reward proxies $Z \in \mathcal{Z}$ and $W \in \mathcal{W}$ in additional to $(S, A, R)$. These proxies are required to satisfy the following assumptions (Miao et al., 2018b):

**Assumption 1.** (a) $R \perp\!\!\!\perp Z \mid (S, U, A)$; (b) $W \perp\!\!\!\perp (Z, A) \mid (S, U)$, $W \not\perp\!\!\!\perp U \mid S$; (c) $R(a) \perp\!\!\!\perp A \mid (S, U)$ for $a \in \mathcal{A}$; (d) There exists a bridge function $q : \mathcal{W} \times \mathcal{A} \times \mathcal{S} \to \mathbb{R}$ such that

$$\mathbb{E}\left[q(W, a, S) \mid U, S, A = a\right] = \mathbb{E}\left[R \mid U, S, A = a\right]. \tag{3}$$

Assumptions 1(a)-(b) are standard in proximal causal inference, requiring these proxies to meet certain conditional independence conditions. Assumptions 1(c), called latent unconfoundedness, is mild as we allow $U$ to be unobserved. The last assumption can be satisfied when some completeness and regularity conditions hold. See Miao et al. (2018a) and also Lemma 3.3 below for more details. Then the following lemma allows us to consistently learn the super-policy $\nu^*$ from the observed data.

**Lemma 3.2.** Under Assumption 1, we have $\mathbb{E}\left[R(a) \mid S=s, A=a'\right] = \mathbb{E}[q(W,a,S) \mid S=s, A=a']$, which further leads to that $\mathcal{V}(\nu) = \mathbb{E}\left[\sum_{a \in \mathcal{A}} q(W,a,S)\nu(a \mid S, A)\right]$ for any $\nu \in \Omega$.

In practice, one may want to include as many confounders in the policy as possible to achieve the largest super-optimality. Hence under this proximal causal inference framework, with some abuse of notation, we further extend the policy class to $\Omega = \{\nu : \mathcal{S} \times \widetilde{\mathcal{Z}} \times \mathcal{A} \to \mathcal{P}(\mathcal{A})\}$ and consider the corresponding super-policy $\nu^*$ as

$$\nu^*(a^* \mid s, \tilde{z}, a') = 1 \quad \text{if} \quad a^* = \mathrm{argmax}_{a \in \mathcal{A}} \mathbb{E}\left[R(a) \mid S=s, \widetilde{Z}=\tilde{z}, A=a'\right], \quad (4)$$

where $\widetilde{Z}$ is a subset of $Z$ that continues to exist when we implement the super-policy. In applications where the action proxy is no longer available in future decision making, equation 4 is reduced to equation 2. We also remark that different from $Z$, $W$ is obtained after intervention. As such, it does not make sense to include $W$ in the super-policy. The following corollary allows us to identify $\nu^*$.

**Corollary 3.1.** Under Assumption 1, the policy value under a given $\nu \in \Omega$ is given by $\mathcal{V}(\nu) = \mathbb{E}\left[\sum_{a \in \mathcal{A}} q(W,a,S)\nu(a \mid S, A, \widetilde{Z})\right]$. In addition, the optimal policy $\nu^*$ is given by

$$\nu^*(a^* \mid s, \tilde{z}, a') = 1 \quad \text{if} \quad a^* = \mathrm{argmax}_{a \in \mathcal{A}} \mathbb{E}\left[q(W,a,S) \mid S=s, \widetilde{Z}=\tilde{z}, A=a'\right]. \quad (5)$$

It can be seen from Corollary 3.1 that to identify the super-policy, it remains to estimate the bridge function $q$ defined in Assumption 1(d). One can impose the following completeness condition.

**Assumption 2.** For any squared-integrable function $g$ and for any $(s,a) \in \mathcal{S} \times \mathcal{A}$, $\mathbb{E}[g(U) \mid Z, S=s, A=a] = 0$ almost surely if and only if $g(U) = 0$ almost surely.

**Lemma 3.3.** Under Assumptions 1-2 and some regularity conditions (see Assumption 7 in Appendix E, solving the following linear integral equation

$$\mathbb{E}\left[q(W,a,S) \mid Z, S, A=a\right] = \mathbb{E}\left[R \mid Z, S, A=a\right], \quad (6)$$

for every $a \in \mathcal{A}$ with respect to $q$ gives a valid bridge function that satisfies Assumption 1(d).

Built upon Corollary 3.1 and Lemma 3.3, Algorithm 1 summarizes the procedure to find $\nu^*$ from a population perspective. Practical procedure that learns $\nu^*$ given samples can be found in Appendix B.

---

**Algorithm 1:** Identification of $\nu^*$ in confounded contextual bandits.

---

1 **Input:** i.i.d. copies of $(S, Z, A, R, W)$.
2 Compute $q$ by solving $\mathbb{E}\left[q(W,a,S) \mid Z, S, A=a\right] = \mathbb{E}\left[R \mid Z, S, A=a\right]$ for every $a \in \mathcal{A}$.
3 Compute $a^* = \mathrm{argmax}_{a \in \mathcal{A}} \mathbb{E}\left[q(W,a,S) \mid S=s, \widetilde{Z}=\tilde{z}, A=a'\right] \forall (s, \tilde{z}, a') \in \mathcal{S} \times \widetilde{\mathcal{Z}} \times \mathcal{A}$.
4 **Output:** $\nu^*$ with $\nu^*(a^* \mid s, z, a') = 1$ for any $(s, z, a')$.

---

## 4 SUPER RL IN SEQUENTIAL DECISION MAKING

### 4.1 MODEL SETUP AND SUPER-POLICIES IN SEQUENTIAL DECISION MAKING

In this section, we formally introduce the super-policy in confounded sequential decision making, demonstrate its super-optimality, and present several non-parametric identification results. Consider an episodic and confounded POMDP denoted by $\mathcal{M} = (\mathcal{S}, \mathcal{U}, \mathcal{A}, T, \mathcal{P}, r)$ where $\mathcal{S}$ and $\mathcal{U}$ denote the spaces of observed and unobserved features respectively, $\mathcal{A}$ denotes the action space, $T$ denotes the total length of horizon, $\mathcal{P} = \{\mathcal{P}_t\}_{t=1}^T$ where each $\mathcal{P}_t$ denotes transition kernel from $\mathcal{S} \times \mathcal{U} \times \mathcal{A}$ to $\mathcal{S} \times \mathcal{U}$ at time $t$, and $r = \{r_t\}_{t=1}^T$ denotes the set of reward functions over $\mathcal{S} \times \mathcal{U} \times \mathcal{A}$. The data following $\mathcal{M}$ can be summarized as $\{S_t, U_t, A_t, R_t\}_{t=1}^T$ where $S_t$ and $U_t$ correspond to the observed

and latent features at time $t$, $A_t$ and $R_t$ denote the action and the reward at time $t$. For simplicity, we assume the action space is discrete and all rewards are uniformly bounded, i.e., $|R_t| \leq R_{\max}$.

Given an offline dataset, our objective is to learn an (in-class) optimal policy to maximize the expected cumulative rewards. All existing works consider policies defined as a sequence of functions mapping from the past history (excluding the current action) to a probability mass function over the action space $\mathcal{A}$. Specifically, given a generic policy $\pi = \{\pi_t\}_{t=1}^T$, one can define its value function as

$$V_t^\pi(s, u) = \mathbb{E}^\pi \left[ \sum_{t'=t}^T R_{t'} \,\middle|\, S_t = s, U_t = u \right], \quad \text{for every } (s, u) \in \mathcal{S} \times \mathcal{U}, \qquad (7)$$

where $\mathbb{E}^\pi$ denotes the expectation with respect to the distribution whose action at each time $t$ follows $\pi_t$. Existing works aim to leverage the batch data to estimate an optimal policy that maximizes

$$\mathcal{V}(\pi) = \mathbb{E}[V_1^\pi(S_1, U_1)], \qquad (8)$$

where we use $\mathbb{E}$ to denote the expectation with respect to the initial data distribution. Under unmeasured confounding, the observed action $A_t$ in the batch data is generated by some behavior policy $\pi_t^b : \mathcal{S} \times \mathcal{U} \to \mathcal{P}(\mathcal{A})$ for $1 \leq t \leq T$. Let $\pi^b = \{\pi_t^b\}_{t=1}^T$.

To handle unmeasured confounding, we similarly assume the existence of certain reward proxies $\{W_t\}_{t=1}^T$ and action proxies $\{Z_t\}_{t=1}^T$ that can help identify policy values. In sequential decision making, as shown in Tennenholtz et al. (2020), past and future observations can be served as the two proxies in confounded partially observable Markov decision processes (POMDPs). As such, our method can be applied to most confounded decision-making problems where human agents will recommend actions in the future. Concrete examples of these proxies are given in later sections and Appendix A. Previous works such as Lu et al. (2022) focus on finding $\pi^* \in \Pi \equiv \{\pi = \{\pi_t\}_{t=1}^T \mid \pi_t : \mathcal{S} \times \mathcal{Z}_t \to \mathcal{P}(\mathcal{A})\}$ such that $\pi^* = \operatorname{argmax}_{\pi \in \Pi} \mathcal{V}(\pi)$. In particular, when $Z_t$s are certain current features that can serve as the action proxies (see Section 4.2), $\Pi$ corresponds to the class of stationary policies. When $Z_t$s are given by the entire data history (see Section 4.3), $\Pi$ corresponds to the class of general history-dependent policies. When $Z_t$s are given by the most recent $k$-step observations (see Section 4.4), $\Pi$ corresponds to the class of $k$-memory policies.

Motivated by the discussions in Section 3, we propose to learn a super policy $\nu^* \in \Omega \equiv \{\nu = \{\nu_t\}_{t=1}^T \mid \nu_t : \mathcal{S} \times \mathcal{Z}_t \times \mathcal{A} \to \mathcal{P}(\mathcal{A})\}$ which leverages human expertise for enhanced policy making that maximizes $\mathcal{V}(\nu)$. Here $\mathcal{A}$ in $\Omega$ reflects the action space at the current time point $t$. Actions recommended by the expert before time $t$ can be included in $Z_t$. See Section 4.3 for more details. When considering $\Omega$, the policy value $\mathcal{V}(\nu)$ indeed depends on $\pi^b$ as well because to implement the proposed super-policy we require the human agent to produce an action according to $\pi^b$ and then intervene using $\nu$. However, to ease notation, we omit $\pi^b$ when referring to $\mathcal{V}(\nu)$. Similar as before, since the super-policy additionally uses the expert's recommendation that depends on the unobserved information, we expect the super-policy $\nu^*$ to be superior to both $\pi^*$ and $\pi^b$, which is shown below.

**Theorem 4.1** (Super-Optimality). $\mathcal{V}(\nu^*) \geq \max\{\mathcal{V}(\pi^*), \mathcal{V}(\pi^b)\}$.

## 4.2 Identification of Stationary Super-Policies via Q-bridge Functions

Under unmeasured confounding, we apply the proximal causal inference framework to sequential decision making and make following assumptions to identify the policy value $\mathcal{V}(\nu)$ for each $\nu \in \Omega$.

**Assumption 3.** (a) (Markovianity) The process $\{S_t, U_t, A_t, R_t\}_{t=1}^T$ satisfies the Markov property, i.e., for any $t$, $(R_t, S_{t+1}, U_{t+1})$ depends on the past history only through $(S_t, U_t, A_t)$.

(b) (Reward proxy) $W_t \perp\!\!\!\perp (A_t, U_{t-1}, S_{t-1}) \mid (U_t, S_t)$, $W_t \not\perp\!\!\!\perp U_t \mid S_t$, for $1 \leq t \leq T$.

(c) (Action proxy) $Z_t \perp\!\!\!\perp (R_t, W_t, S_{t+1}, U_{t+1}, W_{t+1}) \mid (U_t, S_t, A_t)$ for $1 \leq t \leq T$.

Assumption 3 is satisfied by a wide range of confounded sequential decision making models. See Appendix A for detailed discussions. Specifically, Assumption (a) is mild. It essentially requires the data to be Markovian if we were to observe $\{U_t\}_{t=1}^T$. Assumptions (b) and (c) extends Assumption 1 to sequential decision making. In this section, we require the existence of current features that can serve as action proxy and focus on learning an optimal stationary policy. Alternatively, one can set the action proxy to past observations, as in Sections 4.3 – 4.5 and study history-dependent policies. Without loss of generality, we also assume these action proxies continue to be available when making decisions in the future. Otherwise, we can restrict the super-policy to be a function of $(S_t, A_t)$ only.

To identify $\mathcal{V}(\nu)$ and ultimately $\nu^*$ under unmeasured confounding, we rely on the existence of a class of $Q$-bridge functions $\{q_t^\nu\}_{t=1}^T$ defined over $\mathcal{W} \times \mathcal{S} \times \mathcal{A}$ such that for every $(s, u, a) \in \mathcal{S} \times \mathcal{U} \times \mathcal{A}$,

$$\mathbb{E}^\nu \left[ \sum_{t'=t}^T R_{t'} \,\Big|\, U_t, S_t, A_t \right] = \mathbb{E} \left[ \sum_{a \in \mathcal{A}} q_t^\nu(W_t, S_t, a)\nu_t(a \mid S_t, Z_t, A_t) \mid U_t, S_t, A_t \right]. \quad (9)$$

**Theorem 4.2** (Identification). If there exist $\{q_t^\nu\}_{t=1}^T$ that satisfy equation 9, then the value of policy $\nu$ can be identified by $\mathcal{V}(\nu) = \mathbb{E}[\sum_{a \in \mathcal{A}} q_1^\nu(W_1, S_1, a)\nu_1(a \mid S_1, Z_1, A_1)]$.

The following theorem proves the identifiability of these Q-bridge functions under certain completeness and regularity conditions. Together with Theorem 4.2, it forms the basis to learn the super-policy from the observed data. Let $V_t^\nu(W_t, S_t, Z_t, A_t) = q_t^\nu(W_t, S_t, a)\nu_t(a \mid S_t, Z_t, A_t)$.

**Theorem 4.3.** Under Assumption 3 and certain completeness and regularity (Assumptions 8, 9 and 10 in Appendix F), there always exist Q-bridge functions $\{q_t^\nu\}_{t=1}^T$ satisfying equation 9. In particular, set $q_{T+1}^\nu = 0$, $q_t^\nu$ can be obtained by solving the following linear integral equations for $t = T, \ldots, 1$,

$$\mathbb{E}\{q_t^\nu(W_t, S_t, A_t) - R_t - V_{t+1}^\nu(W_{t+1}, S_{t+1}, Z_{t+1}, A_{t+1}) \mid Z_t, S_t, A_t\} = 0. \quad (10)$$

### 4.3 IDENTIFICATION OF GENERAL HISTORY-DEPENDENT SUPER-POLICIES

In this section, we set $Z_t = \{O_{1:t}, A_{1:(t-1)}\}$, $S_t = \emptyset$ and $W_t$ to certain future features that can serve as a reward proxy that satisfies Assumption 3(b) (e.g., conditionally independent of the current action). The corresponding space of $Z_t$ is given by $\mathcal{Z}_t = \prod_{t'=1}^t \mathcal{O} \times \prod_{t'=1}^{t-1} \mathcal{A}$. Alternatively, one may set $Z_t = \{O_{1:(t-1)}, A_{1:(t-1)}\}$ and $W_t$ to the current observation as in Tennenholtz et al. (2020); Shi et al. (2021) to meet Assumption 3. The resulting model is reduced to a typical POMDP with unmeasured confounding and we present the identification results in Section 4.5. We focus on the case where $A_{1:(t-1)}$ in $Z_t$ are generated by the behavior policy instead of the super-policy. The policy class we focus on is given by $\Omega^{\text{history}} = \{\nu = \{\nu_t\}_{t=1}^T \mid \nu_t : \prod_{t'=1}^t (\mathcal{O} \times \mathcal{A}) \to \mathcal{P}(\mathcal{A})\}$, which includes all actions recommended by the expert for decision making but those generated by $\nu \in \Omega$. We leave the inclusion of these actions in the policy class as future work. To ease notation, we omit "history" in $\Omega^{\text{history}}$ when there is no confusion. Let $O_0$ denote some pre-collected observation before the decision process initiates. We impose the following additional assumption:

**Assumption 4.** (a) $Z_{t+1} \perp\!\!\!\perp O_0 \mid U_t, Z_t, A_t$, for $1 \leq t \leq T - 1$; (b) $W_t \perp\!\!\!\perp O_0 \mid U_t, Z_t, A_t$, for $1 \leq t \leq T$; (c) $O_t \in \mathcal{O}$ is generated from $U_t$ by some unknown map $\mathbb{H}_t : \mathcal{U} \to \mathcal{O}$.

Assumption 4(a)-(b) can be easily satisfied by initializing the decision process at $t = 2$. Assumption 4(c) is often imposed in POMDPs. Then we have the following identification results.

**Theorem 4.4.** Assume assumptions 3, 4, and certain completeness and regularity conditions in Appendix F hold. Define $q_{T+1}^\nu = 0$, and $\{q_t^\nu\}_{t=1}^T$ over $\mathcal{W} \times \prod_{t'=1}^t (\mathcal{O} \times \mathcal{A})$ as the solutions to the following linear integral equations:

$$\mathbb{E}\left\{ q_t^\nu(W_t, Z_t, A_t) - R_t - \sum_{a \in \mathcal{A}} q_{t+1}^\nu(W_{t+1}, Z_{t+1}, a)\nu_t(a \mid Z_{t+1}, A_{t+1}) \mid Z_t, O_0, A_t \right\} = 0, \quad (11)$$

for $t = T, T-1, \ldots, 1$. Then we could identify the policy value for $\nu \in \Omega^{\text{history}}$ as

$$\mathcal{V}(\nu) = \mathbb{E}\left[q_1^\nu(W_1, Z_1, A_1)\right]. \quad (12)$$

Theorem 4.4 allows us to identify general history-dependent policies.

### 4.4 IDENTIFICATION OF K-STEP HISTORY-DEPENDENT SUPER-POLICIES

In Section 4.3, we discuss how to identify the value of a history-dependent policy by taking $Z_t$ as past observations up to time $t$. As a result, the dimension of $Z_t$ increases linearly with $t$, resulting in the curse of dimensionality and history (Pineau et al., 2006). In this section, we consider a more practical class of policies that only use the most recent $k$-step observations. Policies of this type are widely used in practice (see e.g., Mnih et al., 2015; Berner et al., 2019).

To begin with, let $W_t$ be the future proxy reward that satisfies Assumption 3(b). For any $t \geq k + 1$, let $Z_t \in \mathcal{Z}_t$ denote the observed history from time $t - k$ up to time $t$, i.e., $(O_{(t-k):t}, A_{(t-k):(t-1)})$.

We further define $\tilde{Z}_t = Z_t \cap Z_{t+1} \in \tilde{\mathcal{Z}}_t$ as a subset of $Z_t$. Next, we define the $Q$-bridge functions $\{q_t^\nu\}_{t=k+1}^T$ over $\mathcal{W} \times \tilde{\mathcal{Z}}_t \times \mathcal{A}$ such that for every $(u, a) \in \mathcal{U} \times \mathcal{A}$ and $t \geq k+1$,

$$\mathbb{E}^{\nu_{t:T}}[\sum_{t'=t}^T R_{t'} \mid U_t, A_t] = \mathbb{E}[\sum_{a \in \mathcal{A}} q_t^\nu(W_t, \tilde{Z}_t, a)\nu_t(a \mid Z_t, A_t) \mid U_t, A_t]. \tag{13}$$

Under certain regularity conditions (Assumptions 13 and 14 specified in Appendix F), we are able to identify the $Q$-bridge functions $\{q_t^\nu\}_{t=k+1}^T$ through the following linear integral equations.

**Theorem 4.5.** Under Assumptions 3, 4(c), Assumptions 13 and 14 in Appendix F, there exist Q-bridge functions $\{q_t^\nu\}_{t=k+1}^T$ satisfying equation 13. In particular, set $q_{T+1}^\nu = 0$, $q_t^\nu$ can be obtained by solving the following linear integral equations for $t = T, \cdots, k+1$:

$$\mathbb{E}\{q_t^\nu(W_t, \tilde{Z}_t, A_t) - R_t - \sum_{a \in \mathcal{A}} q_{t+1}^\nu(W_{t+1}, \tilde{Z}_{t+1}, a)\nu_{t+1}(a \mid Z_{t+1}, A_{t+1}) \mid Z_t, A_t\} = 0. \tag{14}$$

As for $1 \leq t \leq k$, take $Z_t = \{O_{1:t}, A_{1:(t-1)}\}$, if additionally Assumptions 11, 12 in Appendix F and Assumption 4(a)-(b) on $O_0$ hold for $1 \leq t \leq k$, then there exist $\{q_t^\nu\}_{t=1}^k$ over $\mathcal{W} \times (\prod_{t'=1}^t)(\mathcal{O} \times \mathcal{A})$ as the solution to the following linear integral equation for $t = 1, \ldots, k$.

$$\mathbb{E}\{q_t^\nu(W_t, Z_t, A_t) - R_t - \sum_{a \in \mathcal{A}} q_{t+1}^\nu(W_{t+1}, Z_{t+1}, a)\nu_{t+1}(a \mid Z_{t+1}, A_{t+1}) \mid Z_t, O_0, A_t\} = 0, \tag{15}$$

where $O_0$ denotes some pre-collected observation defined in Section 4.3. Finally, the policy value can be identified as $\mathcal{V}(\nu) = \mathbb{E}[q_1^\nu(W_1, Z_1, A_1)]$.

We remark that the requirement for $O_0$ in Theorem 4.5 is much weaker than that in Theorem 4.4. In particular, here we only need Assumptions 4 (a)-(b), 11 and 12 to hold for the first $k$ steps. When $t \geq k+1$, we require the variability of $Z_t$ to cover the variability of $(U_t, \tilde{Z}_t)$, which to some extent requires the observation at $k$-th lag has sufficient variability relative to the variability of unobserved state at the current time $(U_t)$. As the lag $k$ increases, this assumption becomes more restrictive.

## 4.5 ALTERNATIVE IDENTIFICATION OF SUPER-POLICIES

In Section 4.2, we discuss how to identify the policy value via Q-bridge functions assuming the existence of certain future observations $(W_t)$ that can serve as reward proxy and are conditionally independent of the current action. As commented earlier, this condition can be relaxed by setting $W_t = O_t$, $Z_t = \{O_{1:(t-1)}, A_{1:(t-1)}\}$ and $S_t = \emptyset$. The resulting data generating process is reduced to the POMDP model studied in Tennenholtz et al. (2020). However, based on identification results in Sections 4.3-4.4, this rules out the dependence of the super-policy on the most recent observation, which could be restrictive. In the following, we provide a remedy for addressing this limitation.

For simplicity, we focus on identifying a given history-dependent super-policy $\nu = \{\nu_t\}_{t=1}^T$'s value, where $\nu_t : \prod_{t'=1}^t(\mathcal{O} \times \mathcal{A}) \to \mathcal{P}(\mathcal{A})$ depends on all the past observations and recommended actions. We consider a tabular setting where all random variables can only take finitely many values and use boldface letters $\boldsymbol{r} \in \mathbb{R}^{d_r}$, $\boldsymbol{u} \in \mathbb{R}^{d_u}$, $\boldsymbol{o} \in \mathbb{R}^{d_o}$ to represent the vectors consisting of all possible rewards, latent states and observations. Meanwhile, our results can be extended to general settings as well using value-bridge functions (Shi et al., 2021). Let $O_0$ denote some pre-collected observation. The following assumption summarizes the conditions for the model:

**Assumption 5.** (a) The process $\{U_t, A_t\}_{t=1}^T$ satisfies the Markov property; (b) For all $1 \leq t \leq T$, the observation $O_t$ is generated from $U_t$ by some unknown map $\mathbb{H}_t : \mathcal{U} \to \mathcal{O}$; (c) For all $1 \leq t \leq T$, $O_{t-1} \perp\!\!\!\perp (R_t, O_t, U_{t+1}) \mid (U_t, A_t)$.

We define the following matrices:

$$[\boldsymbol{P}_{o,a}^{(t,r)}]_{i,j} = \Pr(R_t = \boldsymbol{r}_i, O_t = o \mid A_t = a, O_{t-1} = \boldsymbol{o}_j), \qquad \boldsymbol{P}_{o,a}^{(r)} \in \mathbb{R}^{d_r \times d_o};$$

$$[\boldsymbol{P}_a^{(t)}]_{i,j} = \Pr(O_t = \boldsymbol{o}_i \mid A_t = a, O_{t-1} = \boldsymbol{o}_j), \qquad \boldsymbol{P}_a^{(t)} \in \mathbb{R}^{d_o \times d_o};$$

$$[\boldsymbol{P}_{o,a',a}^{(t,o)}]_{i,j} = \Pr(O_{t+1} = \boldsymbol{o}_i, O_t = \boldsymbol{o}, A_{t+1} = a' \mid A_t = a, O_{t-1} = \boldsymbol{o}_j), \qquad \boldsymbol{P}_{o,a',a}^{(t)} \in \mathbb{R}^{d_o \times d_o}$$

$$[\boldsymbol{P}_{a,u}^{(t)}]_{i,j} = \Pr(U_t = \boldsymbol{u}_i \mid A_t = a, O_{t-1} = \boldsymbol{o}_j), \qquad \boldsymbol{P}_{a,u}^{(t)} \in \mathbb{R}^{d_u \times d_o}.$$

**Theorem 4.6.** Under Assumption 5, as long as $\boldsymbol{P}_a^{(t)}$ and $\boldsymbol{P}_{a,u}^{(t)}$ are invertible for any $t = 1, \ldots, T$ and $a \in \mathcal{A}$, the value function $\mathcal{V}(\nu)$ for any $\nu \in \Omega$ is identifiable. In particular,

$$\mathcal{V}(\nu) = \sum_{t=1}^{T} \{ \sum_{o_1, a_1, a_1', \ldots, o_t, a_t, a_t'} (\prod_{k=1}^{t} \nu_k(a_k \mid o_k, a_k', \ldots, o_1, a_1')) \boldsymbol{r}^{\mathsf{T}}$$

$$(\boldsymbol{P}_{o_t, a_t}^{(t,r)} [\boldsymbol{P}_{a_1}^{(t)}]^{-1}) (\prod_{k=t-1}^{1} \boldsymbol{P}_{o_k, a_{k+1}', a_k}^{(k,o)} [\boldsymbol{P}_{a_k}^{(k)}]^{-1}) \Pr(O_1 = \boldsymbol{o}, A_1 = a_1) \}.$$

## 5 SUPER-POLICY LEARNING WITH REGRET GUARANTEE

Based on the established identification results, we introduce our super-policy learning algorithms and establish the corresponding finite-sample regret bounds. We only focus on settings described in Sections 3 and 4.2. Other settings can be similarly studied, which we will leave as the future work.

### 5.1 CONFOUNDED CONTEXTUAL BANDITS: REGRET GUARANTEES

We develop a practical algorithm in Appendix B, based on the minimax estimation (Dikkala et al., 2020). Let $\hat{\nu}^*$ denote the output of Algorithm 3 in Appendix B which relies on the estimation of the bridge function $q$ given by equation 6. Define the $\mathcal{L}_2$ norm of a generic function $f$ as $\|f\|_2 \equiv \sqrt{\mathbb{E}[f^2]}$. Let $g(S, Z, A\,;f) \equiv \mathbb{E}[f(W, S) \mid S, Z, A]$ for any $f$ defined over $\mathcal{W} \times \mathcal{S}$. For a given projection estimator $\hat{\mathbb{E}}$, let $\hat{g}(S, Z, A\,;f) \equiv \hat{\mathbb{E}}[f(W, S) \mid S, Z, A]$ denote the corresponding estimator. Define

$$p_{\max} = \sup_{u, s, z, a', \nu \in \Omega} \frac{\sum_{a \in \mathcal{A}} \pi_b(A = a \mid U = u, S = s) \nu(A' = a' \mid Z = z, S = s, A = a)}{\pi_b(A' = a' \mid U = u, S = s)}.$$

**Lemma 5.1.** Suppose $q$ belongs to certain function class $\mathcal{Q} \subset \mathcal{W} \times \mathcal{S} \times \mathcal{A}$. Define the projection error as $\xi_n := \sup_{q \in \mathcal{Q}, a \in \mathcal{A}} \|g[\cdot, \cdot, \cdot\,;\, q(\cdot, \cdot, a)] - \hat{g}[\cdot, \cdot, \cdot\,;\, q(\cdot, \cdot, a)]\|_2$, and the bridge function estimation error as $\zeta_n := \|q - \hat{q}\|_2$. Then we obtain the following regret decomposition

$$\mathcal{V}(\nu^*) - \mathcal{V}(\hat{\nu}^*) \le 2(\xi_n + p_{\max} \zeta_n).$$

Suppose $\hat{q}$ and the projection estimator are computed by the procedure described in Appendix B. When $\mathcal{Q}$ (the function space for $q$) and $\mathcal{G}$ (the function space for the projected function) are VC-subgraph classes, we have the following theorem for the regret guarantee. Results when $\mathcal{G}$ and $\mathcal{Q}$ are reproducing kernel Hilbert spaces (RKHSs) are provided in Appendix I.3.

**Theorem 5.1.** If the star-shaped spaces $\mathcal{G}$ and $\mathcal{Q}$ are VC-subgraph classes with VC dimensions $\mathbb{V}(\mathcal{G})$, and $\mathbb{V}(\mathcal{G})$ respectively. Under assumptions in Theorems I.2 and I.4, with probability at least $1 - \delta$,

$$\mathcal{V}(\hat{v}^*) - \mathcal{V}(v^*) \lesssim n^{-1/2} p_{\max} \sqrt{\log(1/\delta) + \max\{\mathbb{V}(\mathcal{G}), \mathbb{V}(\mathcal{Q})\}},$$

where for any two positive sequences $\{a_n\}_n$, $\{b_n\}_n$, $a_n \lesssim b_n$ means that there exists some constant $C > 0$ such that $a_n \le C b_n$ for any $n$.

### 5.2 CONFOUNDED SEQUENTIAL DECISION MAKING: REGRET GUARANTEES

Now we present our super-policy learning algorithm for the sequential setting introduced in Section 4.2. Given the identification results in Theorems 4.2 and 4.6, to obtain the super-policy $\nu^*$, one solution is to directly search over the space of super-policies that maximizes the estimated value, i.e., $\hat{\nu} = \operatorname{argmax}_{v \in \Omega} \widehat{\mathcal{V}}(\nu)$. However, when $T$ is large and models imposed for estimating bridge functions are complex (e.g., deep neural networks), direct optimizing $\widehat{\mathcal{V}}(\nu)$ requires extensive computational power. Motivated by Theorem 4.3, we propose a fitted-Q-iteration type algorithm (Algorithm 2) for practical implementation and estab-

lish the regret bound in finding the super policy $\nu^*$ under memoryless unmeasured confounding.

---

**Algorithm 2:** Super RL for the confounded POMDP

---

1 **Input:** Data $\mathcal{D} = \{\mathcal{D}_t\}_{t=1}^T$ with $\mathcal{D}_t = \{(S_{i,t}, Z_{i,t}, A_{i,t}, R_{i,t}, W_{i,t}, S_{i,t+1}, Z_{i,t+1}, W_{i,t+1})\}_{i=1}^n$.

2 Let $\hat{q}_{T+1} = 0$ and $\hat{\nu}_T^*$ be an arbitrary policy.

3 Repeat for $t = T, \dots, 1$:

4      Obtain an estimator $\hat{q}_t$ for $q_t$ via a min-max estimation method in Appendix I.1 using $\mathcal{D}_t$

5      Compute $\hat{\mathbb{E}}[q_t(W_t, S_t, a) \mid S_t = s, Z_t = z, A_t = a']$ for $a \in \mathcal{A}$ using the method in
     Appendix I.2 and obtain the estimated super policy $\hat{\nu}_t^*$ as for every $(a', z, s)$,

$$\hat{\nu}_t^*(a^* \mid s, z, a') = \mathbb{1}\left\{\mathrm{argmax}_{a \in \mathcal{A}}\, \hat{\mathbb{E}}[q_t(W_t, S_t, a) \mid S_t = s, Z_t = z, A_t = a']\right\}.$$

6 **Output:** $\hat{\nu}^* = \{\hat{\nu}_t^*\}_{t=1}^T$.

---

**Assumption 6** (Memoryless Unmeasured Confounding). For $2 \le t \le T$, $U_t$ is independent of past data history (including latent factors in the past) up to time $t-1$ given $S_t$.

We introduce some notations. Define $p_t^\nu$ and $p_t^{\pi_b}$ as the marginal distributions of all random variables at time $t$ under the policy $\nu$ and behavior policy $\pi_b$ respectively. Define constants $p_{t,\max} := \sup_{s,z,a} p_t^{\nu^*}(s, z, a)/p_t^{\pi_b}(s, z, a)$, and $p_{t,\max}^\omega = \sup_{s,z,a,\nu \in \Omega} \omega_t^\nu(s, z, a)$, where $\omega_t^\nu(s, z, a)$ denotes certain density ratio whose explicit form is given in equation 29 of Appendix H.

Let $\mathcal{Q}^{(t)}$ denote the class for modelling $q_t$. Define $g_t[S_t, Z_t, A_t \; ; \; q(\cdot, \cdot, a)] := \mathbb{E}[q(W_t, S_t, a) \mid S_t, Z_t, A_t]$ and $\hat{g}_t[S_t, Z_t, A_t \; ; \; q(\cdot, \cdot, a)] := \hat{\mathbb{E}}[q(W_t, S_t, a) \mid S_t, Z_t, A_t]$ for $q \in \mathcal{Q}^{(t)}$ and $a \in \mathcal{A}$. Consider two projection errors $\xi_{t,n} := \sup_{q \in \mathcal{Q}^{(t)}, a \in \mathcal{A}} \|g_t[\cdot, \cdot, \cdot \; ; \; q(\cdot, \cdot, a)] - \hat{g}_t[\cdot, \cdot, \cdot \; ; \; q(\cdot, \cdot, a)]\|_2$ and $\zeta_{t,n}$ which denotes the projected error related to the computation in line 4 of Algorithm 2. The exact definition of $\zeta_{t,n}$ is given in equation 37 of Appendix I. The finite-sample regret bound of $\hat{\nu}^*$ by Algorithm 2 relies on the following regret decomposition.

**Lemma 5.2.** Suppose $q_t \in \mathcal{Q}^{(t)}$ for $1 \le t \le T$ and $\hat{\nu}^*$ is computed via Algorithm 2. Then under Assumptions 3, 6, 8, 9 and 10, we obtain the following regret decomposition,

$$\mathcal{V}(\nu^*) - \mathcal{V}(\hat{\nu}^*) \lesssim \left(\sum_{t=1}^T 2 p_{t,\max}\xi_{t,n}\right) + \sqrt{T\sum_{t=1}^T (p_{t,\max}^\omega)^2 (\zeta_{t,n})^2}.$$

In Appendix I, we provide a detailed analysis of $\xi_{t,n}$ and $\zeta_{t,n}$ regarding to the critical radii of local Rademacher complexities of different spaces, when $\hat{q}_t$ is estimated by the conditional moment method and the projection $\mathbb{E}[q(W_t, S_t, a) \mid S_t, Z_t, A_t]$ is estimated by the empirical risk minimization. Here we provide a regret bound which is characterized by the VC dimensions. Let $\mathcal{G}^{(t)}$ be the space of testing functions in the min-max estimating procedure described in Appendix I.1, and $\mathcal{H}^{(t)}$ be the space of inner products between any policy $\nu \in \Omega$ and $q \in \mathcal{Q}^{(t)}$ with $\mathcal{H}^{(T+1)} = \{0\}$. See the exact definitions of $\mathcal{G}^{(t)}$ and $\mathcal{H}^{(t)}$ in Appendix I.1.

**Theorem 5.2.** If the star-shaped spaces $\mathcal{G}^{(t)}$, $\mathcal{H}^{(t+1)}$ and $\mathcal{Q}^{(t)}$ are VC-subgraph classes with VC dimensions $\mathbb{V}(\mathcal{G}^{(t)})$, $\mathbb{V}(\mathcal{H}^{(t+1)})$ and $\mathbb{V}(\mathcal{G}^{(t)})$ respectively for $1 \le t \le T$. Under assumptions specified in Theorems I.1 and I.3, with probability at least $1 - \delta$,

$$\mathcal{V}(\hat{\nu}^*) - \mathcal{V}(v^*) \lesssim \sum_{t=1}^T (p_{t,\max} + p_{t,\max}^\omega)(T - t + 1)^{2.5}\sqrt{\frac{\log(T/\delta) + \max\{\mathbb{V}(\mathcal{G}^{(t)}), \mathbb{V}(\mathcal{H}^{(t+1)}), \mathbb{V}(\mathcal{Q}^{(t)})\}}{n}}.$$

When $\mathcal{G}^{(t)}$, $\mathcal{Q}^{(t)}$ and $\mathcal{H}^{(t)}$ are RKHSs, we establish the corresponding results in Appendix I.3.

## 6 CONCLUSION

In this paper, we introduce super reinforcement learning, which takes the observed action as input for enhanced policy learning. We establish the identification results for the super-policy in various confounded environments. Practical algorithms are proposed to perform the super-policy learning and corresponding finite-sample regret guarantees are provided.

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

## A    POMDP STRUCTURES AND PROXY VARIABLES

### A.1    POMDP STRUCTURES

In Figure 1, we illustrate the general POMDP structure regarding to the variables $\{U_t, S_t, A_t, R_t\}_{t=1}^T$. Figure 2 provides an example of the POMDP structure under the memoryless assumption (Assumption 6). As Figure 2 shows, all the information from the past time steps is transited to the next step only through the current observed state $S_t$. Figure 3 provides an illustration for the causal relationship of all the variables involved in the confounded POMDP. At any time step $t$, the reward proxy $W_t$ is only related to the action $A_t$ through $S_t$ and $U_t$; the action proxy $Z_t$ is only related to the reward $R_t$ through $S_t$ and $U_t$. In Section A.2, we provide more illustrations about the relationship of proxy variables with other variables.

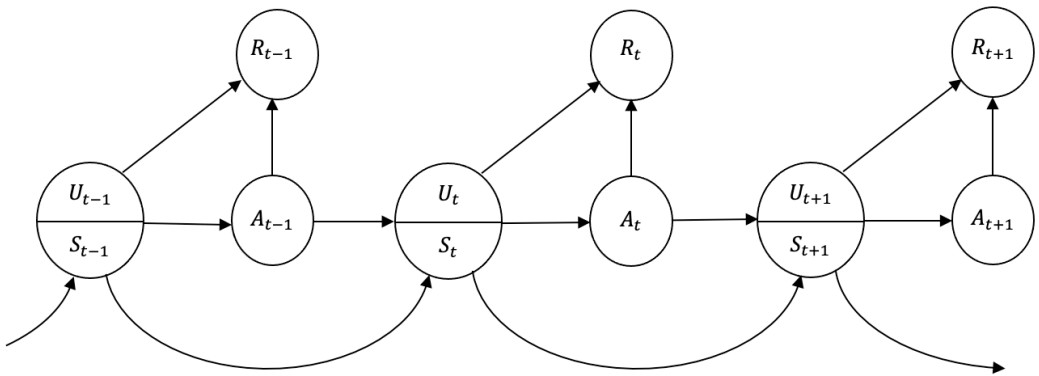

Figure 1: The data generation process under a typical POMDP.

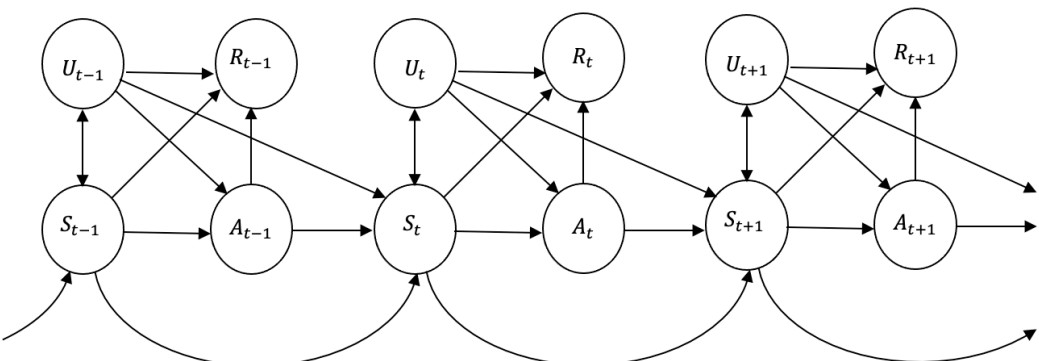

Figure 2: The data generation process under the memoryless POMDP.

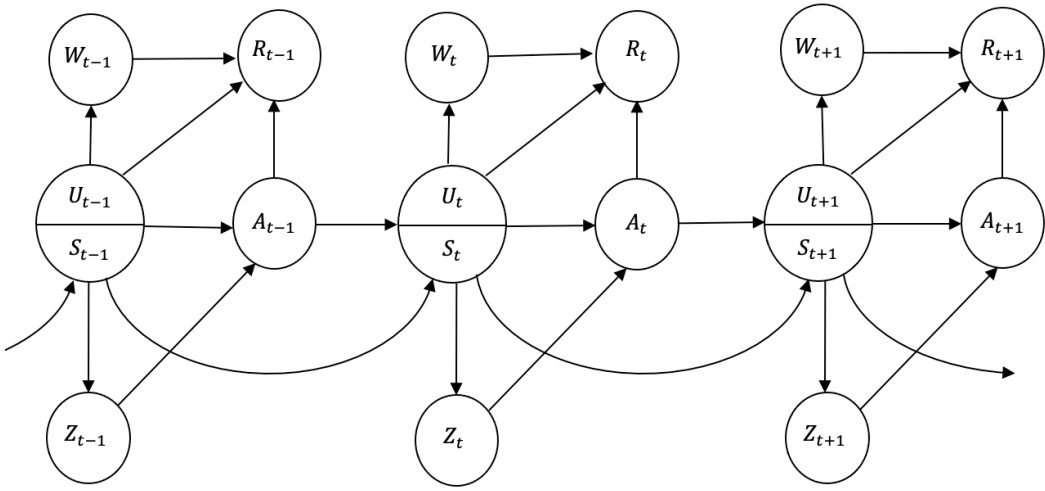

Figure 3: An illustration of the causal relationship of variables involved in the confounded POMDP.

## A.2 PROXY VARIABLES

In this section, we discuss several options for proxy variables $W_t$ and $Z_t$ satisfying the basic assumption (Assumption 3).

In Figure 4, we list some plausible causal relationship among $W_t$, $U_t$, $A_t$. We require the effect between $U_t$ and $W_t$ exists, but the effect between $W_t$ and $R_t$ is optional. For concrete examples of $W_t$, readers can refer to the discussion of type c variables in Tchetgen Tchetgen et al. (2020).

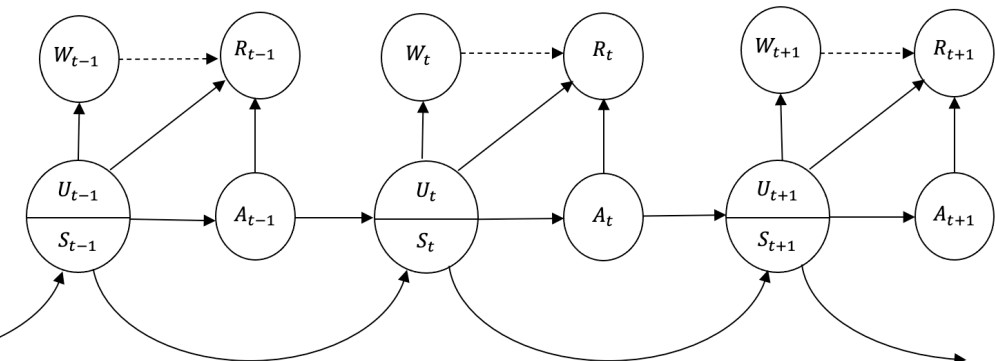

Figure 4: Causal relationship between $W_t$ and other variables. Dashed arrows indicate the causal effect is optional.

Once we determine $W_t$, we can select $Z_t$ accordingly. Figure 5 shows several different relationships of the action proxy $Z_t$ with other variables. In the left plot of Figure 5, $Z_t$ is one of the cause of $A_t$ and $Z_t \perp\!\!\!\perp (U_t, S_t) \mid A_t$. In this case, $Z_t$ can be considered as an instrumental variable for $A_t$. In the middle plot of Figure 5, $(U_t, S_t)$ is a direct cause for $Z_t$, the effect between $Z_t$ and $A_t$ can be in both directions and can be optional. As for the right plot in Figure 5, $Z_t$ is a direct effect of $U_t$ and $S_t$. And the effect between $Z_t$ and $A_t$ can be in both directions and can be optional. For concrete examples of choices of $Z_t$ in the observational study, readers can refer to the discussion of type b variables in Tchetgen Tchetgen et al. (2020). In Section 4.3 and 4.4, we also discuss the cases when $Z_t$ includes previous history.

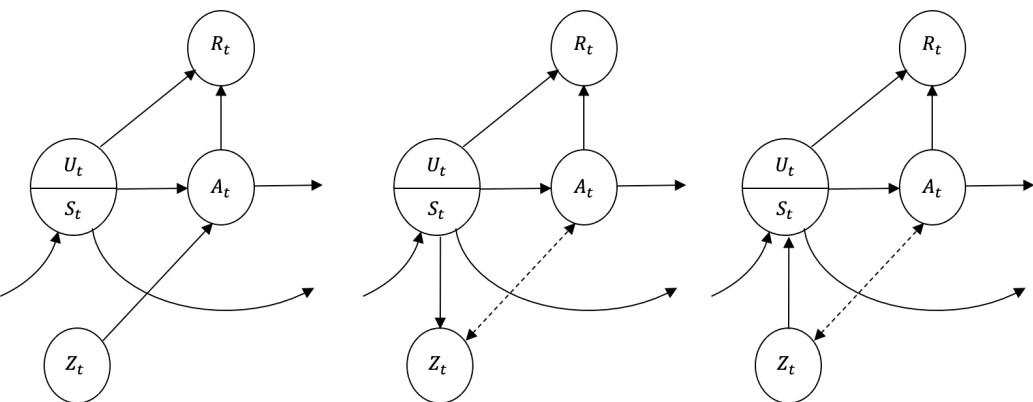

Figure 5: Different causal relationship between $Z_t$ and other variables. Dashed arrows indicate the causal effect is optional.

## B    LEARNING ALGORITHM FOR CONTEXTUAL BANDITS

In this section, we present the practical algorithm (Algorithm 3) for finding the super-policy in our contextual bandit example. The key step is to estimate the bridge function $q$ by the linear integral equation stated in Lemma 3.3. When $\mathcal{S} \times \mathcal{Z} \times \mathcal{A} \times \mathcal{W}$ are all finite and discrete, it can be straightforwardly estimated. In the following, we discuss the estimation when the general space is considered.

---

**Algorithm 3:** Learning Algorithm for the contextual bandits under unmeasured confounding

---

1 **Input:** Data $\mathcal{D} = (S_i, Z_i, A_i, R_i, W_i)_{i=1}^n$.
2 Obtain the estimation of the bridge function $\hat{q}$ by solving the estimation equation equation 6 using data $\mathcal{D}$
3 Implement any supervised learning method for estimating $\mathbb{E}\left[\hat{q}(W, S, a) \mid S, Z, A\right]$.
4 Compute $a^* = \text{argmax}_{a \in \mathcal{A}} \hat{\mathbb{E}}\left[\hat{q}(W, S, a) \mid S = s, Z = z, A = a'\right] \quad \forall(s, z, a') \in \mathcal{S} \times \mathcal{Z} \times \mathcal{A}$.
5 **Output:** $\hat{\nu}^*$ with $\hat{\nu}(a^* \mid s, z, a') = 1$ and $\hat{\nu}(\tilde{a} \mid s, z, a') = 0$ for $\tilde{a} \neq a^*$.

---

We consider the conditional moment estimation procedure in Dikkala et al. (2020), and propose to estimate $Q$-bridge function by

$$\hat{q} := \underset{q \in \mathcal{Q}}{\arg\min} \sup_{g \in \mathcal{G}} \tilde{\Psi}(q, g) - \lambda \left( \|g\|_{\mathcal{G}}^2 + \frac{U}{\Delta^2} \|g\|_{2,n}^2 \right) + \lambda\mu\|q\|_{\mathcal{Q}}^2, \tag{16}$$

where $\tilde{\Psi}(q, g) = \frac{1}{n} \sum_{i=1}^n \{q(W_i, S_i, A_i) - R_i\} g(Z_i, S_i, A_i)$, $\mathcal{Q}$ is the function space that we assume $q^*$ lies in, $\mathcal{G}$ is the function space where the test functions $g$ come from, $\lambda, \mu, \Delta, U > 0$ are some tuning parameters.

As for the projection $\hat{\mathbb{E}}[\hat{q}(W, S, a) \mid S = s, Z = z, A = a']$, the conditional moment framework can be also adopted to perform the estimation, here we propose to estimate it via the empirical risk minimization.

$$\hat{g}(\cdot, \cdot, \cdot; \ \hat{q}(\cdot, \cdot, a)) := \underset{g \in \mathcal{G}}{\arg\min} \frac{1}{n} \sum_{i=1}^n \left[g(S_i, Z_i, A_i) - \hat{q}(\cdot, \cdot, a)\right]^2 + \mu\|g\|_{\mathcal{G}}^2, \tag{17}$$

where $\hat{q}$ is defined in equation 16, $\mu > 0$ is a tuning parameter.

## C SIMULATIONS

### C.1 SIMULATION STUDY FOR CONTEXTUAL BANDITS

In this section, we conduct two simulation studies to evaluate the performance of the proposed super-policy. The first one is a contextual bandit example with tabular state values. We aim to demonstrate the super-policy performs better when the behavior policy reveals more information about the unmeasured confounders. The second one is a contextual bandit example with a continuous state space. It is used to demonstrate the performance of our algorithm using the bridge function.

**A contextual bandit with tabular state values**

Similar to the toy example described in Section 3, we take $S$ and $U$ as independent binary variables such that $\text{Pr}(S = 1) = 0.5$ and $\text{Pr}(U = 1) = 0.5$. The binary action $A$ is generated by the following conditional probabilities

$$\text{Pr}(A = 1 \mid U = 0) = \epsilon, \qquad \text{Pr}(A = 1 \mid U = 1) = 1 - \epsilon,$$

with different choices of $\epsilon \in [0, 1]$. The larger the $|\epsilon - 0.5|$ is, the more information of $U$ is revealed in the observed action $A$. Both the reward proxy $W$ and the action proxy $Z$ are binary and are generated according to the following conditional probabilities

$$\text{Pr}(W = 1 \mid U = 0) = 0.4, \qquad \text{Pr}(W = 1 \mid U = 1) = 0.6;$$

$$\text{Pr}(Z = 1 \mid U = 0) = 0.4, \qquad \text{Pr}(Z = 1 \mid U = 1) = 0.6.$$

Moreover, $W$ and $Z$ are conditionally independent given $U$. The observed reward is computed by $R = (U - 0.5)(A - 0.5) + \epsilon$ where $\epsilon \sim N(0, 0.5)$.

Three types of policy classes are considered.

1. SONLY: $\mathcal{S} \to \mathcal{P}(\mathcal{A})$. The policy only depends on the observed state $S$.
2. SZONLY: $\mathcal{S} \times \mathcal{Z} \to \mathcal{P}(\mathcal{A})$. The policy depends on on the observed state $S$ and the action proxy $Z$.
3. SUPER: $\mathcal{S} \times \mathcal{Z} \times \mathcal{A} \to \mathcal{P}(\mathcal{A})$. The super-policy class where the policy depends on the observed state $S$, the action proxy $Z_t$, and observed action $A$.

Table 2: Simulation results for the tabular setting described in C.1 under different choices of $\epsilon$. We replicate the simulation for 50 times. Mean regret values for estimated optimal policies under different policy classes are provided (and a smaller regret value indicates a better performance). Values in the parentheses are the standard deviations of the regret values.

|  | SONLY | SZONLY | SUPER |
|---|---|---|---|
| $\epsilon = 0.5$ | 0.25 (3.1e-04) | **0.21** (1.7e-02) | **0.21** (1.4e-02) |
| $\epsilon = 0.7$ | 0.25 (3.1e-04) | 0.22 (1.8e-02) | **0.18** (3.5e-02) |
| $\epsilon = 0.9$ | 0.25 (2.5e-04) | 0.24 (1.2e-02) | **0.17** (8.6e-02) |

Table 3: Simulation results for the continuous setting described in C.1 under different choices of $\epsilon$. The simulation is performed over 50 simulated datasets. Mean regret values for estimated optimal policies using different policy classes are provided. Smaller regret values indicate better performance. Values in the parentheses are the standard deviations of the regret values.

|  | SONLY | SZONLY | SUPER |
|---|---|---|---|
| $\epsilon = 0.5$ | 0.40 (9.6e-04) | 0.14 (6.1e-03) | **0.12** (2.5e-03) |
| $\epsilon = 0.7$ | 0.40 (9.2e-04) | 0.12 (5.9e-03) | **0.11** (3.5e-03) |
| $\epsilon = 0.9$ | 0.40 (1e-03) | 0.11 (1.3e-02) | **0.065** (1e-02) |

We implement Algorithm 1 to estimate the corresponding optimal policies for different policy classes. Note that for SONLY and SZONLY, we perform the projection step (line 4) by conditioning on $S$ and $(S, Z)$ respectively. Since this is a tabular setting, we use the empirical averages to approximate all the conditional expectations. In this simulation study, we consider the sample size $n = 5000$. As Table 2 shows, the super-policy produces smaller regret as $\epsilon$ deviates from $0.5$ more, while the estimated optimal policies such as SONLY and SZONLY do not change and have larger regrets.

**A contextual bandit with a continuous state**

In this setting, we take $S$ and $U$ as independent Gaussian random variables such that $S \sim N(0, 1)$ and $U \sim N(0, 1)$. The binary action $A$ is generated by the following conditional probabilities

$$\Pr(A = 1 \mid U > 0) = \epsilon, \qquad \Pr(A = 1 \mid U \leq 0) = 1 - \epsilon,$$

with different choices of $\epsilon \in [0, 1]$. The larger the $|\epsilon - 0.5|$ is, the more information of $U$ is revealed in the observed action $A$. We generate $W$ and $Z$ according to the following conditional probabilities

$$W \mid (S, U) \sim N(S + 3U, 1);$$

$$Z \mid (S, U, A) \sim N(3S + U + 0.5A, 1).$$

Moreover, $W$ and $Z$ are conditionally independent given $(S, U)$. The observed reward is computed by $R = (U - 0.5)(A - 0.5) + \epsilon$ where $\epsilon \sim N(0, 0.5)$. For this continuous setting, we compute the $Q$-bridge function via the min-max conditional moment estimation described in Appendix I by taking $\mathcal{G}, \mathcal{Q}$ as reproducing kernel Hilbert Spaces (RKHSs) equipped with Gaussian kernels. The bandwidths of Gaussian kernels are selected by the median heuristic. Tuning parameters of the penalties are selected by cross-validation. Computation details can be found in Section E of Dikkala et al. (2020). As for the projection step, we adopt kernel ridge regression (KRR) to perform the estimation, and the tuning parameter of the penalty is selected by cross-validation. In this simulation study, we take the sample size $n = 1000$.

Table 3 shows the simulation results over 50 replications. The observation is consistent with that in the tabular setting. And the super-policy outperforms the other two commonly used optimal policies when $\epsilon$ deviates from $0.5$.

## C.2 A SIMULATION STUDY FOR SEQUENTIAL DECISION MAKING

In this section, we perform a simulation study to evaluate the performance of the super-policy in the sequential decision making. Specifically, we mainly follow the data generation process

Table 4: Simulation results for the sequential decision making problem described in C.2. The simulation is performed over 50 simulated datasets. Mean regret values for estimated optimal policies under different policy classes are provided. The smaller regret values indicate better performances. Values in the parentheses are the standard deviations of the regret values.

| SONLY | SZONLY | SUPER |
|---|---|---|
| 5.4 (1.9e-01) | 5.3 (4.7e-01) | **2.2** (4.9e-01) |

described in Section F.1 of Miao et al. (2022)), and only change the reward function to $R_t = \text{expit}\{U_t(A_t - 0.5)\} + e_t$, where $e_t \sim \text{Uniform}[-0.1, 0.1]$ and $\text{expit}(x) = 1/(1 + \exp(-x))$. We take the sample size as $n = 1000$ and the length of episode $T = 20$. Note that this setting satisfies the memoryless assumption (i.e., Assumption 6). We implement Algorithm 2 to estimate the optimal policies from three policy classes considered in Section C.1 by adjusting the projection step accordingly. We again use the RKHS modeling to perform the min-max conditional moment estimation for obtaining a sequence of $Q$-bridge functions and implement KRR to estimate the projections at every iteration. See implementation details in the discussion of the continuous setting in Section C.1. To obtain the regret value, we estimate the optimal policy which depends on both $S_t$ and $U_t$, and use it to approximate the oracle optimal value. Table 4 summarises the simulation results over 50 simulated datasets. As we can see, the super policy performs significantly better than the other two commonly used optimal policies.

## D   REAL DATA APPLICATIONS

### D.1   APPLICATION TO RHC DATA

In this section, we evaluate the performance of our method on the dataset from the Study to Understand Prognoses and Preferences for Outcomes and Risks of Treatments (SUPPORT Connors et al., 1996). SUPPORT examined the effectiveness and safety of direct measurement of cardiac function by Right Heart Catheterization (RHC) for certain critically ill patients in intensive care units (ICU). This dataset has been studied by many existing works (e.g. Qi et al., 2021; Tchetgen Tchetgen et al., 2020). Our goal is to find an optimal policy on the usage of RHC that maximizes 30-day survival rates 30-day survival rates of critically ill patients from the day admitted or transferred to ICU.

This dataset corresponds to the setting of contextual bandits. There are 5735 patients, of whom 2184 were measured by RHC in the first 24 hours ($A = 1$) and the remaining were considered in the control group ($A = 0$). If a patient survived or censored at day 30, we let the response $Y = 1$, otherwise, we take the response as $Y = -1$. Following the data pre-processing steps in Qi et al. (2021), we consider 71 covariates including demographics, diagnosis, estimated survival probability, comorbidity, vital signs, and physiological status among others in this study. See the full list of covariates in `https://hbiostat.org/data/repo/rhc.html`. In particular, we take the action proxy $Z = (\text{pafi1}, \text{paco21})$ and the reward proxy $W = (\text{ph1}, \text{hema1})$. For more details and justifications of the choices of proxy variables, we refer readers to Section 6.1 of Tchetgen Tchetgen et al. (2020).

We compare the super-policy with the following two policies considered in Qi et al. (2021): $d_1(L, Z)$ and $d_1(L)$, where $d_1(L, Z)$ corresponds to the policy in the policy class SZONLY and $d_1(L)$ corresponds to the policy in the policy class SONLY. To make it more comparable, we use the same estimating procedure for the bridge functions considered in these three methods. In addition, the RKHS modeling for the min-max conditional moment estimation is taken to obtain the $Q$-bridge function. See details of the RKHS modeling in the continuous setting in Section C.1. Since Qi et al. (2021) adopt the linear modeling for the decision functions $d_1(L, Z)$ and $d_1(L)$, we also use the linear regression to obtain the projection (line 4) in Algorithm 3.

To evaluate the value by different policies, we randomly separate 40% of the data and use it as the evaluation set $\mathcal{E}$. More specifically, after obtaining the estimated optimal policies using 60% of the data, we perform the policy evaluation of these three estimated optimal policies using the remaining 40% of the data. Take $\hat{q}$ as the estimated bridge function using $\mathcal{E}$. The evaluation is

Table 5: Evaluation results of the optimal policies learned from three different policy classes using the RHC data. The averages of evaluation values over 20 random splits are presented. Larger values indicate better performances. Values in the parentheses are standard deviations.

| SONLY | SZONLY | SUPER |
|---|---|---|
| 0.55 (5.80e-02) | 0.55 (5.78e-02) | **0.69** (1.10e-02) |

Table 6: Evaluation results of the optimal policies learned from three different policy classes using the MIMIC-III data. The averages of evaluation values over 20 random splits are presented. Larger values indicate better performances. Values in the parentheses are standard deviations.

| SONLY | SZONLY | SUPER |
|---|---|---|
| -2.83 (5.30e-02) | -2.81 (5.03e-02) | **-1.75** (1.14e-02) |

conducted as follows. $\mathcal{V}(\nu) = \hat{\mathbb{E}}\{\sum_{a\in\mathcal{A}} \hat{q}(W,S,a)\nu(a \mid S,Z,A)\}$, for $\nu \in$ SUPER; $\mathcal{V}(\pi) = \hat{\mathbb{E}}\{\sum_{a\in\mathcal{A}} \hat{q}(W,S,a)\pi(a \mid S,Z)\}$, for $\pi \in$ SZONLY; $\mathcal{V}(\pi) = \hat{\mathbb{E}}\{\sum_{a\in\mathcal{A}} \hat{q}(W,S,a)\nu(a \mid S)\}$, for $\pi \in$ SONLY. The expectation $\hat{\mathbb{E}}$ refers to the average with respect to the evaluation set $\mathcal{E}$.

Table 5 shows the evaluation results over 20 random splits. As we can see, the super-policy produces higher policy values compared with the other two methods.

### D.2 APPLICATION TO MIMIC3 DATA

In this section, we use the Multi-parameter Intelligent Monitoring in Intensive Care (MIMIC-III) dataset (`https://physionet.org/content/mimiciii/1.4/`) to demonstrate the performance of estimated optimal policies from three policy classes (SONLY, SZONLY and SUPER). This dataset records the longitudinal information (including information of demographics, vitals, labs and scores, see details in Section 4.3 of Nanayakkara et al. (2022)) of patients who satisfied the sepsis criteria, and the goal is to learn an optimal personalized treatment strategy for sepsis. Despite the richness of data collected at the ICU, the mapping between true patient states and clinical observations is usually ambiguous (Nanayakkara et al., 2022), and therefore makes this dataset fit into the setting of a confounded POMDP.

We obtain a clean dataset following the same data pre-processing steps described in Raghu et al. (2017). Based on it, we take (vasopressor administration, fluid administration) as the action variable, (-1)*SOFA as the reward function. We take (Weight, Temperature) as the reward proxy $W$ since they are not directly related to the action. All the remaining variables except for aforementioned ones are treated as observed state variables. The action proxy is taken as (Weight, Temperature) observed from the last time step. And it is natural to assume that (Weight, Temperature) observed from the last time step is not directly related to the response at the current time step. To simplify the complexity of the action space, we discretize vasopressor and fluid administrations into 2 bins, instead of 5 as in the previous work (Raghu et al., 2017). This results in a 4-dimensional action space. The numbers of episode length for every patient differ in the dataset. We decide to fix the horizon $T = 2$, and exclude those patients with records less than 2 time steps.

Following the estimation steps described in Section C.2, we estimate the optimal policies under policy classes SONLY, SZONLY and SUPER respectively. We also adopt the idea of "random splitting" described in Section D.1 to evaluate different policies. Basically, we randomly divide the data into two parts with equal sample sizes. We use one part as the training data to learn optimal policies. The other part is used for evaluating the corresponding policies. We implement the off-policy evaluation method proposed by Miao et al. (2022) in the confounded POMDP to calculate the policy values.

 summarizes the evaluation results over 20 random splits. As we can see, the super-policy produces higher policy values compared to the other two methods.

## E    TECHNICAL PROOFS IN SECTION 3

*Proof of Lemma 3.1.*

$$\mathcal{V}(\pi^*) = \mathbb{E}\left\{\sum_{a\in\mathcal{A}} R(a)\pi^*(a\mid S)\right\} = \mathbb{E}\left[\mathbb{E}\left\{\sum_{a\in\mathcal{A}} R(a)\pi^*(a\mid S)\mid S,Z,A\right\}\right]$$

$$\leq \mathbb{E}\left[\mathbb{E}\left\{\sum_{a\in\mathcal{A}} R(a)\nu^*(a\mid S,Z,A)\mid S,Z,A\right\}\right] = \mathcal{V}(\nu^*).$$

The first inequality is due to the optimality of $\nu^*$. Similarly, for the behavior policy $\pi^b$, we can show that

$$\mathcal{V}(\pi^b) = \mathbb{E}\left[\mathbb{E}\left\{\sum_{a\in\mathcal{A}} R(a)\mathbf{1}(a=A)\mid S,Z,A\right\}\right]$$

$$\leq \mathbb{E}\left[\mathbb{E}\left\{\sum_{a\in\mathcal{A}} R(a)\nu^*(a\mid S,Z,A)\mid S,Z,A\right\}\right] = \mathcal{V}(\nu^*).$$

$\square$

*Proof of Lemma 3.2.*

$$\mathbb{E}\left[R(a)\mid S=s, A=a'\right] = \mathbb{E}\left[\mathbb{E}\left\{R(a)\mid U,S=s,A=a'\right\}\mid S=s,A=a'\right]$$

$$= \mathbb{E}\left[\mathbb{E}\left\{R(a)\mid U,S=s\right\}\mid S=s,A=a'\right] \tag{18}$$

$$= \mathbb{E}\left[\mathbb{E}\left\{R\mid U,S=s,A=a\right\}\mid S=s,A=a'\right]$$

$$= \mathbb{E}\left[\mathbb{E}\left\{q(W,a,S)\mid U,S=s,A=a\right\}\mid S=s,A=a'\right] \tag{19}$$

$$= \mathbb{E}\left[\mathbb{E}\left\{q(W,a,S)\mid U,S=s,A=a'\right\}\mid S=s,A=a'\right] \tag{20}$$

$$= \mathbb{E}\left[q(W,a,S)\mid S=s,A=a'\right],$$

where equation 18 is because of Assumption 1(c), equation 19 is from equation 3 in Assumption 1 and equation 20 is due to Assumption 1(b). $\square$

To close this section, we prove Lemma 3.3. The following regularity condition is imposed. For a probability measure function $\mu$, let $\boldsymbol{L}^2\{\mu(x)\}$ denote the space of all squared integrable functions of $x$ with respect to measure $\mu(x)$, which is a Hilbert space endowed with the inner product $\langle g_1, g_2 \rangle = \int g_1(x)g_2(x)\mathrm{d}\mu(x)$. For all $s,a,t$, define the following operator

$$K_{s,a} : \boldsymbol{L}^2\left\{\mu_{W\mid S,A}(w\mid s,a)\right\} \to \boldsymbol{L}^2\left\{\mu_{Z\mid S,A}(z\mid s,a)\right\}$$
$$h \mapsto \mathbb{E}\left\{h(W)\mid Z=z, S=s, A=a\right\},$$

and its adjoint operator

$$K_{s,a}^* : \boldsymbol{L}^2\left\{\mu_{Z\mid S,A}(z\mid s,a)\right\} \to \boldsymbol{L}^2\left\{\mu_{W\mid S,A}(w\mid s,a)\right\}$$
$$g \mapsto \mathbb{E}\left\{g(Z)\mid W=w, S=s, A=a\right\}.$$

**Assumption 7** (Regularity conditions for contextual bandits). For any $Z=z, S=s, W=w, A=a$,

(a)  $\iint_{\boldsymbol{W}\times\boldsymbol{Z}} f_{W\mid Z,S,A}(w\mid z,s,a)f_{Z\mid W,S,A}(z\mid w,s,a)\mathrm{d}w\mathrm{d}z < \infty$, where $f_{W_t\mid Z_t,S_t,A_t}$ and $f_{Z_t\mid W_t,S_t,A_t}$ are conditional density functions.

(b)

$$\int_{\boldsymbol{Z}} \left[\mathbb{E}\left\{R_t\mid Z=z,S=s,A=a\right\}\right]^2 f_{Z\mid S,A}(z\mid s,a)\mathrm{d}z < \infty.$$

(c) There exists a singular decomposition $(\lambda_{s,a;\nu}, \phi_{s,a;\nu}, \psi_{s,a;\nu})_{\nu=1}^{\infty}$ of $K_{s,a}$ such that,

$$\sum_{\nu=1}^{\infty} \lambda_{s,a;\nu}^{-2} |\langle \mathbb{E}\{R_t \mid Z = z, S = s, A = a\}, \psi_{s,a;t;\nu}\rangle|^2 < \infty.$$

*Proof of Lemma 3.3.* From equation 6, we have

$$\begin{aligned}
0 &= \mathbb{E}[R - q(W, A, S) \mid Z, S, A] \\
&= \mathbb{E}\{\mathbb{E}[R - q(W, A, S) \mid U, Z, S, A] \mid Z, S, A\} \\
&= \mathbb{E}\{\mathbb{E}[R - q(W, A, S) \mid U, S, A] \mid Z, S, A\},
\end{aligned} \tag{21}$$

where equation 21 is due to Assumption 1(b). Then by Assumption 2, we have

$$\mathbb{E}[R - q(W, A, S) \mid U, S, A] = 0,$$

which is exactly equation 3. In addition, by Proposition 1 in Miao et al. (2018a), the solution to equation 6 exists under Assumption 7. Then Lemma 3.3 is proved. □

## F  TECHNICAL PROOFS IN SECTION 4

*Proof of Theorem 4.1.* First of all, note that there is one-to-one corresponding policy of $\pi_b$ and $\pi^*$ in $\Omega$ respectively. Specifically, for $\{\pi_t^b\}_{t=1}^T$, we can let $\nu_t^{\pi_b}(a \mid S_t, a') = \mathbf{1}(a = a')$ almost surely to recover $\pi_b$. For $\pi^*$, we can always choose $\nu^{\pi^*}$ such that $\nu^{\pi^*}(a \mid S_t, A_t) = \pi^*(a \mid S_t)$. This completes our proof that $\nu^*$ achieves the super-optimality. □

Next, to show Theorem 4.3, we need to make some additional conditions.

**Assumption 8.** $(Z_{t+1}, A_{t+1}) \perp\!\!\!\perp Z_t \mid (U_t, S_t, A_t)$ for $1 \le t \le T - 1$.

**Assumption 9** (Completeness). For any $(s, a) \in \mathcal{S} \times \mathcal{A}$, $t = 1, \dots, T$,

   (a) For any square-integrable function $g$, $\mathbb{E}\{g(U_t) \mid Z_t, S_t = s, A_t = a\} = 0$ a.s. if and only if $g = 0$ a.s;

   (b) For any square-integrable function $g$, $\mathbb{E}\{g(Z_t) \mid W_t, S_t = s, A_t = a\} = 0$ a.s. if and only if $g = 0$ a.s.

For a probability measure function $\mu$, let $\boldsymbol{L}^2\{\mu(x)\}$ denote the space of all squared integrable functions of $x$ with respect to measure $\mu(x)$, which is a Hilbert space endowed with the inner product $\langle g_1, g_2 \rangle = \int g_1(x) g_2(x) \mathrm{d}\mu(x)$.

**Assumption 10** (Regularity conditions). For all $s, a, t$, define the following operator

$$\begin{aligned}
K_{s,a;t} : \boldsymbol{L}^2\left\{\mu_{W_t|S_t,A_t}(w \mid s, a)\right\} &\to \boldsymbol{L}^2\left\{\mu_{Z_t|S_t,A_t}(z \mid s, a)\right\} \\
h &\mapsto \mathbb{E}\left\{h(W_t) \mid Z_t = z, S_t = s, A_t = a\right\}.
\end{aligned}$$

Take $K_{s,a;t}^*$ as the adjoint operator of $K_{s,a,t}$.

For any $Z_t = z, S_t = s, W_t = w, A_t = a$ and $1 \le t \le T$, following conditions hold:

   (a) $\iint_{\boldsymbol{W} \times \boldsymbol{Z}} f_{W_t|Z_t,S_t,A_t}(w \mid z, s, a) f_{Z_t|W_t,S_t,A_t}(z \mid w, s, a) \mathrm{d}w\mathrm{d}z < \infty$, where $f_{W_t|Z_t,S_t,A_t}$ and $f_{Z_t|W_t,S_t,A_t}$ are conditional density functions.

   (b) For any $g \in \boldsymbol{G}^{(t+1)}$,

   $$\int_{\boldsymbol{Z}} [\mathbb{E}\{R_t + g(W_{t+1}, S_{t+1}, Z_{t+1}, A_{t+1}) \mid Z_t = z, S_t = s, A_t = a\}]^2 f_{Z_t|S_t,A_t}(z \mid s, a)\mathrm{d}z < \infty.$$

   (c) There exists a singular decomposition $(\lambda_{s,a;t;\nu}, \phi_{s,a;t;\nu}, \psi_{s,a;t;\nu})_{\nu=1}^{\infty}$ of $K_{s,a;t}$ such that for all $g \in \boldsymbol{G}^{(t+1)}$,

   $$\sum_{\nu=1}^{\infty} \lambda_{s,a;t;\nu}^{-2} |\langle \mathbb{E}\{R_t + g(W_{t+1}, S_{t+1}, Z_{t+1}, A_{t+1}) \mid Z_t = z, S_t = s, A_t = a\}, \psi_{s,a;t;\nu}\rangle|^2 < \infty.$$

(d) For all $1 \le t \le T$, $v_t^\pi \in \mathbf{G}^{(t)}$ where $\mathbf{G}^{(t)}$ satisfies the regularity conditions (b) and (c) above.

**Now we are ready to prove Theorem 4.3.**

*Proof of Theorem 4.3.* **Part I.** We suppose there exists $q_t^\pi$ satisfying equation 10, $1 \le t \le T$. Define $V_t^\nu(W_t, S_t, Z_t, A_t) = \sum_{a \in \mathcal{A}} q_t^\nu(W_t, S_t, a)\pi(a \mid S_t, Z_t, A_t)$ and $V_{T+1}^\nu = 0$. Then

$$\mathbb{E}\left\{R_t + V_{t+1}^\nu(W_{t+1}, S_{t+1}, Z_{t+1}, A_{t+1}) \mid Z_t, S_t, A_t\right\}$$
$$=\mathbb{E}\left[\mathbb{E}\left\{R_t + V_{t+1}^\nu(W_{t+1}, S_{t+1}, Z_{t+1}, A_{t+1}) \mid U_t, Z_t, S_t, A_t\right\} \mid Z_t, S_t, A_t\right]$$
$$=\mathbb{E}\left[\mathbb{E}\left\{R_t + V_{t+1}^\nu(W_{t+1}, S_{t+1}, Z_{t+1}, A_{t+1}) \mid U_t, S_t, A_t\right\} \mid Z_t, S_t, A_t\right] \text{ by Assumption 3 and 8,}$$

and

$$\mathbb{E}\left\{q_t^\nu(W_t, S_t, A_t) \mid Z_t, S_t, A_t\right\}$$
$$=\mathbb{E}\left[\mathbb{E}\left\{q_t^\nu(W_t, S_t, A_t) \mid U_t, Z_t, S_t, A_t\right\} \mid Z_t, S_t, A_t\right]$$
$$=\mathbb{E}\left[\mathbb{E}\left\{q_t^\nu(W_t, S_t, A_t) \mid U_t, S_t, A_t\right\} \mid Z_t, S_t, A_t\right] \text{ by Assumption 3.}$$

Therefore, by Assumption 9 (a), we have

$$\mathbb{E}\left\{R_t + V_{t+1}^\nu(W_{t+1}, S_{t+1}, Z_{t+1}, A_{t+1}) \mid U_t, S_t, A_t\right\} = \mathbb{E}\left\{q_t^\nu(W_t, S_t, A_t) \mid U_t, S_t, A_t\right\} \quad \text{a.s.}$$

and for any $a \in \mathcal{A}$,

$$\mathbb{E}\left\{R_t + V_{t+1}^\nu(W_{t+1}, S_{t+1}, Z_{t+1}, A_{t+1}) \mid U_t, S_t, A_t = a\right\}$$
$$=\mathbb{E}\left\{q_t^\nu(W_t, S_t, A_t) \mid U_t, S_t, A_t = a\right\} = \mathbb{E}\left\{q_t^\nu(W_t, S_t, a) \mid U_t, S_t, A_t = a\right\}$$
$$=\mathbb{E}\left\{q_t^\nu(W_t, S_t, a) \mid U_t, S_t\right\}. \tag{22}$$

Next, we prove that

$$\mathbb{E}^\nu\left\{R_t + V_{t+1}^\nu(W_{t+1}, S_{t+1}, Z_{t+1}, A_{t+1}) \mid U_t, S_t, Z_t, A_t\right\} = \mathbb{E}\left\{\sum_{a \in \mathcal{A}} q_t^\nu(W_t, S_t, a)\nu_t(a \mid S_t, Z_t, A_t) \mid U_t, S_t, Z_t, A_t\right\} \quad \text{a.s.} \tag{23}$$

Take $W_{t+1}(a), S_{t+1}(a), Z_{t+1}(a), U_{t+1}(a)$ as the counterfactual variables had the action $a$ is taken at the current time $t$ as $a$. For any $a \in$,

$$\mathbb{E}^\nu\left\{R_t + V_{t+1}^\nu(W_{t+1}, S_{t+1}, Z_{t+1}, A_{t+1}) \mid U_t, S_t, Z_t, A_t = a\right\}$$
$$= \sum_{a' \in \mathcal{A}} \mathbb{E}\left[R_t(a') + V_{t+1}^\nu(W_{t+1}(a'), S_{t+1}(a'), Z_{t+1}(a'), \pi_b(S_{t+1}(a'), U_{t+1}(a'))) \mid U_t, S_t, Z_t, A_t = a\right]$$
$$\qquad \nu_t(a' \mid S_t, Z_t, A_t = a)$$
$$= \sum_{a' \in \mathcal{A}} \mathbb{E}\left[R_t(a') + V_{t+1}^\nu(W_{t+1}(a'), S_{t+1}(a'), Z_{t+1}(a'), \pi_b(S_{t+1}(a'), U_{t+1}(a'))) \mid U_t, S_t, A_t = a\right]$$
$$\qquad \nu_t(a' \mid S_t, Z_t, A_t = a) \text{ by Assumption 3}$$
$$= \sum_{a' \in \mathcal{A}} \mathbb{E}\left[R_t(a') + V_{t+1}^\nu(W_{t+1}(a'), S_{t+1}(a'), Z_{t+1}(a'), \pi_b(S_{t+1}(a'), U_{t+1}(a'))) \mid U_t, S_t\right]\nu_t(a' \mid S_t, Z_t, A_t = a)$$
$$= \sum_{a' \in \mathcal{A}} \mathbb{E}\left[R_t + V_{t+1}^\pi(W_{t+1}, S_{t+1}, Z_{t+1}, A_{t+1}) \mid U_t, S_t, A_t = a'\right]\nu_t(a' \mid S_t, Z_t, A_t = a)$$
$$= \sum_{a' \in \mathcal{A}} \mathbb{E}\left[q_t^\nu(W_t, S_t, a') \mid U_t, S_t\right]\nu_t(a' \mid S_t, Z_t, A_t = a) \text{ by equation 22}$$
$$= \sum_{a' \in \mathcal{A}} \mathbb{E}\left[q_t^\nu(W_t, S_t, a') \mid U_t, S_t, Z_t, A_t = a\right]\nu_t(a' \mid S_t, Z_t, A_t = a),$$

where the fourth, fifth and last equations are based on the unconfoundedness assumption once $U_t$ is given and $W_t$ is independent of $(A_t, Z_t)$ given $U_t, S_t$. Therefore, equation 23 is verified.

**Part II.** We will use this Bellman-like equation equation 23 to verify equation 9 and thus establish the identification results. First, at time $T$, by equation 23 and $V_{T+1}^\nu = 0$,

$$\mathbb{E}^\nu\left(R_T \mid U_T, S_T, Z_T, A_T\right) = \mathbb{E}\left[\sum_{a\in\mathcal{A}} q_T^\nu(W_T, S_T, a)\nu_T(a \mid S_T, Z_T, A_T) \mid U_T, S_T, Z_T, A_T\right].$$

By induction, suppose that at time $t+1$, $\mathbb{E}^\nu\left[\sum_{t'=t+1}^T R_{t'} \mid S_{t+1}, U_{t+1}, Z_{t+1}, A_{t+1}\right] = \mathbb{E}\left\{V_{t+1}^\nu(W_{t+1}, S_{t+1}, Z_{t+1}, A_{t+1}) \mid S_{t+1}, U_{t+1}, Z_{t+1}, A_{t+1}\right\}$. Then at time $t$,

$$\mathbb{E}^\nu\left(\sum_{t'=t}^T R_{t'} \,\Big|\, U_t, S_t, Z_t, A_t\right)$$

$$=\mathbb{E}^\nu\left\{R_t + \mathbb{E}^\nu\left(\sum_{t'=t+1}^T R_{t'} \,\Big|\, U_{t+1}, S_{t+1}, Z_{t+1}, A_{t+1}, U_t, S_t, Z_t, A_t\right) \,\Big|\, U_t, S_t, Z_t, A_t\right\}$$

$$=\mathbb{E}^\nu\left\{R_t + \mathbb{E}^\nu\left(\sum_{t'=t+1}^T R_{t'} \,\Big|\, U_{t+1}, S_{t+1}, Z_{t+1}, A_{t+1}\right) \,\Big|\, U_t, S_t, Z_t, A_t\right\} \text{ by Assumption 3}$$

$$=\mathbb{E}^\nu\left\{R_t + \mathbb{E}\left(V_{t+1}^\nu(W_{t+1}, S_{t+1}, Z_{t+1}, A_{t+1}) \,\Big|\, U_{t+1}, S_{t+1}, Z_{t+1}, A_{t+1}\right) \,\Big|\, U_t, S_t, Z_t, A_t\right\}$$

$$=\mathbb{E}^\nu\left\{R_t + \mathbb{E}\left(V_{t+1}^\nu(W_{t+1}, S_{t+1}, Z_{t+1}, A_{t+1}) \,\Big|\, U_{t+1}, S_{t+1}, Z_{t+1}, A_{t+1}, U_t, S_t, Z_t, A_t\right) \,\Big|\, U_t, S_t, Z_t, A_t\right\}$$

by Assumption 3

$$=\mathbb{E}^\nu\left\{R_t + \mathbb{E}^\nu\left(V_{t+1}^\nu(W_{t+1}, S_{t+1}, Z_{t+1}, A_{t+1}) \,\Big|\, U_{t+1}, S_{t+1}, Z_{t+1}, A_{t+1}, U_t, S_t, Z_t, A_t\right) \,\Big|\, U_t, S_t, Z_t, A_t\right\}$$

$$=\mathbb{E}^\nu\left\{R_t + V_{t+1}^\nu(W_{t+1}, S_{t+1}, Z_{t+1}, A_{t+1}) \mid U_t, S_t, Z_t, A_t\right\} \text{ by the law of total expectation}$$

$$=\mathbb{E}\{\sum_{a\in\mathcal{A}} q_t^\nu(W_t, S_t, a)\nu_t(a \mid S_t, Z_t, A_t) \mid U_t, S_t, Z_t, A_t\} \text{ by equation 23.}$$

**Part III.** Now we prove the existence of the solution to equation 10.

For $t = T, \ldots, 1$, by Assumption 10 (a), $K_{s,a;t}$ is a compact operator for each $(s,a) \in \boldsymbol{S} \times \mathcal{A}$ (Carrasco et al., 2007, Example 2.3), so there exists a singular value system stated in Assumption 10 (c). Then by Assumption 9 (b), we have $\mathrm{Ker}(K_{s,a;t}^*) = 0$, since for any $g \in \mathrm{Ker}(K_{s,a;t}^*)$, we have, by the definition of Ker, $K_{s,a;t}^* g = \mathbb{E}\left[g(Z_t) \mid W_t, S_t = s, A_t = a\right] = 0$, which implies that $g = 0$ a.s. Therefore $\mathrm{Ker}(K_{s,a;t}^*) = 0$ and $\mathrm{Ker}(K_{s,a;t}^*)^\perp = \boldsymbol{L}^2(\mu_{Z_t|S_t,A_t}(z \mid s, a))$. By Assumption 10 (b), $\mathbb{E}\left\{R_t + g(W_{t+1}, S_{t+1}, Z_{t+1}, A_{t+1}) \mid Z_t = \cdot, S_t = s, A_t = a\right\} \in \mathrm{Ker}(K_{s,a,;t}^*)$ for given $(s,a) \in \boldsymbol{S}_t \times \mathcal{A}$ and any $g \in \boldsymbol{G}^{(t+1)}$. Now condition (a) in Theorem 15.16 of Kress (1989) has been verified. The condition (b) is satisfied given Assumption 10 (c). Recursively applying the above argument from $t = T$ to $t = 1$ yields the existence of the solution to equation 10. $\square$

Next, we show our generalized identification results stated in Section 4.3. Before that, we make the following assumptions.

**Assumption 11** (Completeness conditions for history-dependent policies). For any $a \in \mathcal{A}$, $t = 1, \ldots, T$,

(a) For any square-integrable function $g$, $\mathbb{E}\{g(U_t, Z_t) \mid Z_t, O_0, A_t = a\} = 0$ a.s. if and only if $g = 0$ a.s;

(b) For any square-integrable function $g$, $\mathbb{E}\{g(Z_t, O_o) \mid W_t, Z_t, A_t = a\} = 0$ a.s. if and only if $g = 0$ a.s.

**Assumption 12** (Regularity Conditions for history-dependent policies). For all $z, a, t$, define the following operator

$$K_{z,a;t} : \boldsymbol{L}^2\left\{\mu_{W_t|Z_t,A_t}(w \mid z, a)\right\} \to \boldsymbol{L}^2\left\{\mu_{O_0|Z_t,A_t}(z \mid o, a)\right\}$$
$$h \mapsto \mathbb{E}\left\{h(W_t) \mid Z_t = z, O_0 = o, A_t = a\right\}.$$

Take $K^*_{z,a;t}$ as the adjoint operator of $K_{z,a,t}$.

For any $Z_t = z, O_0 = o, W_t = w, A_t = a$ and $1 \leq t \leq T$, following conditions hold:

(a) $\iint_{\boldsymbol{W} \times \mathcal{O}} f_{W_t|Z_t,O_0,A_t}(w \mid z, o, a) f_{O_0|W_t,Z_t,A_t}(o \mid w, z, a) \mathrm{d}w \mathrm{d}o < \infty$, where $f_{W_t|Z_t,O_0,A_t}$ and $f_{O_0|W_t,Z_t,A_t}$ are conditional density functions.

(b) For any $g \in \boldsymbol{G}^{(t+1)}$,

$$\int_{\boldsymbol{Z}} \left[ \mathbb{E}\left\{ R_t + g(W_{t+1}, Z_{t+1}, A_{t+1}) \mid Z_t = z, O_0 = o, A_t = a \right\} \right]^2 f_{O_0|Z_t,A_t}(o \mid z, a) \mathrm{d}z < \infty.$$

(c) There exists a singular decomposition $(\lambda_{z,a;t;\nu}, \phi_{z,a;t;\nu}, \psi_{z,a;t;\nu})_{\nu=1}^{\infty}$ of $K_{z,a;t}$ such that for all $g \in \boldsymbol{G}^{(t+1)}$,

$$\sum_{\nu=1}^{\infty} \lambda_{z,a;t;\nu}^{-2} \left| \langle \mathbb{E}\left\{ R_t + g(W_{t+1}, Z_{t+1}, A_{t+1}) \mid Z_t = z, O_0 = o, A_t = a \right\}, \psi_{z,a;t;\nu} \rangle \right|^2 < \infty.$$

(d) For all $1 \leq t \leq T$, $v_t^\pi \in \boldsymbol{G}^{(t)}$ where $\boldsymbol{G}^{(t)}$ satisfies the regularity conditions (b) and (c) above.

**Now we are ready to prove Theorem 4.4.**

*Proof of Theorem 4.4.* The structure of the proof and related arguments are similar to the proof of Theorem 4.3. Mainly, we will show the solution of equation 11 satisfies the following equation

$$\mathbb{E}^\nu \left[ \sum_{t'=t}^{T} R_{t'} \mid U_t, A_t \right] = \mathbb{E}\left[ \sum_{a \in \mathcal{A}} q_t^\nu(W_t, Z_t, a) \nu_t(a \mid Z_t, A_t) \mid U_t, A_t \right],$$

where $\mathbb{E}^\nu$ refers to expectation taken with respect to $\{\nu_t\}_{t=t}^T$. Therefore we only list several key steps in the corresponding three parts of the proof. Take $V_t^\nu(W_t, Z_t, A_t) = \sum_{a \in \mathcal{A}} q_t^\nu(W_t, Z_t, a) \nu(a \mid Z_t, A_t)$.

**Part I.** By Assumption 3 and 4, we have

$$\mathbb{E}\left\{ R_t + V_{t+1}^\nu(W_{t+1}, Z_{t+1}, A_{t+1}) \mid U_t, Z_t, O_0, A_t \right\}$$
$$= \mathbb{E}\left\{ R_t + V_{t+1}^\nu(W_{t+1}, Z_{t+1}, A_{t+1}) \mid U_t, Z_t, A_t \right\}$$

and

$$\mathbb{E}\left\{ q_t^\nu(W_t, Z_t, A_t) \mid U_t, Z_t, O_0, A_t \right\} = \mathbb{E}\left\{ q_t^\nu(W_t, Z_t, A_t) \mid U_t, Z_t, A_t \right\}.$$

Then by Assumption 11 (a), we have

$$\mathbb{E}\left\{ R_t + V_{t+1}^\nu(W_{t+1}, Z_{t+1}, A_{t+1}) \mid U_t, Z_t, A_t \right\} = \mathbb{E}\left\{ q_t^\nu(W_t, Z_t, A_t) \mid U_t, Z_t, A_t \right\} \quad \text{a.s.}$$

and therefore

$$\mathbb{E}^\nu \left\{ R_t + V_{t+1}^\nu(W_{t+1}, Z_{t+1}, A_{t+1}) \mid U_t, Z_t, A_t \right\} = \mathbb{E}\left\{ \sum_{a \in \mathcal{A}} q_t^\nu(W_t, Z_t, a) \nu_t(a \mid Z_t, A_t) \mid U_t, Z_t, A_t \right\} \quad \text{a.s.,}$$
(24)

where $\mathbb{E}^\nu$ refers to expectation taken with respect to $\{\nu_t\}_{t=t}^T$.

**Part II.** Following the same induction idea, we can show that if $\mathbb{E}^\nu \left[ \sum_{t'=t+1}^{T} R_{t'} \mid U_{t+1}, Z_{t+1}, A_{t+1} \right] = \mathbb{E}\left\{ V_{t+1}^\nu(W_{t+1}, Z_{t+1}, A_{t+1}) \mid U_{t+1}, Z_{t+1}, A_{t+1} \right\}$, then by utilizing equation 25, at time $t$, we can obtain

$$\mathbb{E}^\nu \left( \sum_{t'=t}^{T} R_{t'} \mid U_t, Z_t, A_t \right) = \mathbb{E}\left\{ \sum_{a \in \mathcal{A}} q_t^\nu(W_t, Z_t, a) \nu_t(a \mid Z_t, A_t) \mid U_t, Z_t, A_t \right\},$$

where $\mathbb{E}^\nu$ refers to expectation taken with respect to $\{\nu_t\}_{t=t}^T$. **Part III.** The existence of the solution to equation 11 can be verified by utilizing Assumption 11(b) and Assumption 12. $\square$

Lastly, in order to show our generalized identification results stated in Section 4.4, we adapt the completeness and regularity assumptions as follows.

**Assumption 13** (Completeness conditions for k-step history-dependent policies). For any $a \in \mathcal{A}$, $t = k+1, \ldots, T$,

    (a) For any square-integrable function $g$, $\mathbb{E}\{g(U_t, \tilde{Z}_t) \mid Z_t, A_t = a\} = 0$ a.s. if and only if $g = 0$;

    (b) For any square-integrable function $g$, $\mathbb{E}\{g(Z_t) \mid W_t, \tilde{Z}_t, A_t = a\} = 0$ a.s. if and only if $g = 0$ a.s.

**Assumption 14** (Regularity Conditions for k-step history-dependent policies). Define the following conditional expectation operator:

$$K_{s,a;t} : \boldsymbol{L}^2 \left\{ \mu_{(W_t, \tilde{Z}_t)|S_t, A_t}((w, \tilde{z}) \mid s, a) \right\} \to \boldsymbol{L}^2 \left\{ \mu_{Z_t|S_t, A_t}(z \mid s, a) \right\}$$

$$h \mapsto \mathbb{E}\left\{ h(W_t, \tilde{Z}_t) \mid Z_t = z, S_t = s, A_t = a \right\},$$

and take $K_{s,a;t}^*$ as its adjoint operator. For any $\tilde{Z}_t = \tilde{z}, Z_t = z, S_t = s, W_t = w, A_t = a$ and $k+1 \le t \le T$,

    (a) $\iint f_{(W_t, \tilde{Z}_t)|Z_t, S_t, A_t}((w, \tilde{z}) \mid z, s, a) f_{Z_t|W_t, \tilde{Z}_t, S_t, A_t}(z \mid w, \tilde{z}, s, a) \mathrm{d}w \mathrm{d}\tilde{z} \mathrm{d}z < \infty$, where $f_{Z_t|W_t, \tilde{Z}_t, S_t, A_t}$ and $f_{(W_t, \tilde{Z}_t)|Z_t, S_t, A_t}$ are conditional density functions.

    (b) For any $g \in \boldsymbol{G}^{(t+1)}$,

$$\int_{\boldsymbol{Z}} \left[\mathbb{E}\left\{R_t + g(W_{t+1}, S_{t+1}, Z_{t+1}, A_{t+1}) \mid Z_t = z, S_t = s, A_t = a\right\}\right]^2 f_{Z_t|S_t, A_t}(z \mid s, a) \mathrm{d}z < \infty.$$

    (c) There exists a singular decomposition $(\lambda_{s,a;t;\nu}, \phi_{s,a;t;\nu}, \psi_{s,a;t;\nu})_{\nu=1}^{\infty}$ of $K_{s,a;t}$ such that for all $g \in \boldsymbol{G}^{(t+1)}$,

$$\sum_{\nu=1}^{\infty} \lambda_{s,a;t;\nu}^{-2} \left|\langle \mathbb{E}\left\{R_t + g(W_{t+1}, S_{t+1}, Z_{t+1}, A_{t+1}) \mid Z_t = z, S_t = s, A_t = a\right\}, \psi_{s,a;t;\nu}\rangle\right|^2 < \infty.$$

    (d) For all $k+1 \le t \le T$, $v_t^{\pi} \in \boldsymbol{G}^{(t)}$ where $\boldsymbol{G}^{(t)}$ satisfies the regularity conditions (b) and (c) above.

**Now we are ready to prove Theorem 4.5.**

*Proof of Theorem 4.5.* The results for $1 \le t \le k$ can be obtained by directly applying the proof of Theorem 4.4. Here we only show the proof for the case when $t > k$. The proof structure and argument are quite similar to the proof of Theorem 4.3. Therefore, we list several important steps in three parts of the proof. Take $v_t^{\nu}(W_t, Z_t, A_t) = \sum_{a \in \mathcal{A}} q_t^{\nu}(W_t, \tilde{Z}_t, a)\nu(a \mid Z_t, A_t)$.

**Part I.** By Assumption 3,

$$\mathbb{E}\left\{R_t + V_{t+1}^{\nu}(W_{t+1}, Z_{t+1}, A_{t+1}) \mid U_t, Z_t, A_t\right\} = \mathbb{E}\left\{R_t + V_{t+1}^{\nu}(W_{t+1}, Z_{t+1}, A_{t+1}) \mid U_t, \tilde{Z}_t, A_t\right\}$$

and

$$\mathbb{E}\left\{q_t^{\nu}(W_t, \tilde{Z}_t, A_t) \mid U_t, Z_t, A_t\right\} = \mathbb{E}\left\{q_t^{\nu}(W_t, \tilde{Z}_t, A_t) \mid U_t, \tilde{Z}_t, A_t\right\}.$$

Then by Assumption 13 (a), we have

$$\mathbb{E}\left\{R_t + V_{t+1}^{\nu}(W_{t+1}, Z_{t+1}, A_{t+1}) \mid U_t, \tilde{Z}_t, A_t\right\} = \mathbb{E}\left\{q_t^{\nu}(W_t, \tilde{Z}_t, A_t) \mid U_t, \tilde{Z}_t, A_t\right\} \quad \text{a.s.}$$

and therefore

$$\mathbb{E}^{\nu}\left\{R_t + V_{t+1}^{\nu}(W_{t+1}, Z_{t+1}, A_{t+1}) \mid U_t, Z_t, A_t\right\} = \mathbb{E}\left\{\sum_{a \in \mathcal{A}} q_t^{\nu}(W_t, \tilde{Z}_t, a)\nu_t(a \mid Z_t, A_t) \mid U_t, Z_t, A_t\right\} \quad \text{a.s.,}$$

$$\tag{25}$$

where $\mathbb{E}^{\nu}$ refers to expectation taken with respect to $\{\nu_t\}_{t=t}^{T}$.

**Part II.** Following the same induction idea, we can show that if $\mathbb{E}^{\nu}\left[\sum_{t'=t+1}^{T} R_{t'} \mid U_{t+1}, Z_{t+1}, A_{t+1}\right] = \mathbb{E}\left\{V_{t+1}^{\nu}(W_{t+1}, Z_{t+1}, A_{t+1}) \mid U_{t+1}, Z_{t+1}, A_{t+1}\right\}$, then by utilizing equation 25, at time $t$,

$$\mathbb{E}^{\nu}\left(\sum_{t'=t}^{T} R_{t'} \,\Big|\, U_t, Z_t, A_t\right) = \mathbb{E}\left\{\sum_{a\in\mathcal{A}} q_t^{\nu}(W_t, \tilde{Z}_t, a)\nu_t(a \mid Z_t, A_t) \mid U_t, Z_t, A_t\right\},$$

where $\mathbb{E}^{\nu}$ refers to expectation taken with respect to $\{\nu_t\}_{t=t}^{T}$. **Part III.** The existence of the solution to equation 14 can be verified by utilizing Assumption 13(b) and Assumption 14. □

## G  TECHNICAL PROOFS IN SECTION 4.5

*Proof of Theorem 4.6.* We notice that $O_{t-1} \perp\!\!\!\perp (R_t, O_t, U_{t+1})|(U_t, A_t)$. Consequently, the conditional distributions of $(R_t, O_t)$ and $(U_{t+1}, O_t)$ given $(A_t, U_t)$ shall satisfy

$$\Pr(R_t = r, O_t = o|A_t = a, U_t = u)\Pr(U_t = u|A_t = a, O_{t-1} = o)$$
$$= \underbrace{\Pr(R_t = r, O_t = o|A_t = a, O_{t-1} = o)}_{P_{oa}^{(t,r)}},$$

$$\Pr(U_{t+1} = u, O_t = o|A_t = a, U_t = u)\Pr(U_t = u|A_t = a, O_{t-1} = o)$$
$$= \underbrace{\Pr(U_{t+1} = u, O_t = o|A_t = a, O_{t-1} = o)}_{P_{oa}^{(t,u)}},$$

$$\Pr((U_{t+1} = u, A_{t+1} = a', O_t = o|A_t = a, U_t = u)\Pr(U_t = u|A = a, O_{t-1} = o)$$
$$= \underbrace{\Pr(U_{t+1} = u, A_{t+1} = a', O_t = o|A_t = a, O_{t-1} = o)}_{P_{o,a',a}^{(t,u)}},$$

$$\Pr((O_t = o|U_t = u)\Pr(U_t = u|A_t = a, O_{t-1} = o) = \underbrace{\Pr(O_t = o|A_t = a, O_{t-1} = o)}_{P_a^{(t)}}.$$

Accordingly, $\Pr(R_t = r, O_t = o|A_t = a, U_t = u)$, $\Pr(U_{t+1} = u, O_t = o|A_t = a, U_t = u)$, $\Pr((U_t = u, A_{t+1} = a', O_t = o|A_t = a, U_t = u)$ and $\Pr((O_t = o|U_t = u)$ correspond to the matrices consisting of all conditional probabilities. When $P_a^{(t)}$ and $\Pr(U_t = u|A_t = a, O_{t-1} = o)$ are invertible, it allows us to represent $\Pr(R_t = r, O_t = o|A_t = a, U_t = u)$ and $\Pr(U_{t+1} = u, O_t = o|A_t = a, U_t = u)$ and $\Pr((U_{t+1} = u, A_{t+1} = a', O_t = o|A_t = a, U_t = u)$ by

$$\Pr(R_t = r, O_t = o|A_t = a, U_t = u) = P_{oa}^{(t,r)}[P_a^{(t)}]^{-1}\Pr((O_t = o|U_t = u)$$
$$\Pr(U_{t+1} = u, O_t = o|A_t = a, U_t = u) = P_{oa}^{(t,u)}[P_a^{(t)}]^{-1}\Pr((O_t = o|U_t = u)$$
$$\Pr((U_{t+1} = u, A_{t+1} = a', O_t = o|A_t = a, U_t = u) = P_{o,a',a}^{(t,u)}[P_a^{(t)}]^{-1}\Pr((O_t = o|U_t = u)$$

respectively. We first represent $\mathbb{E}^{\nu} R_1$ using the observed data. Notice that

$$\mathbb{E}^{\nu} R_1 = \sum_{a',u}\left[\mathbb{E}\left\{\sum_a R_1(a)\nu_1(a|O_1, A_1)|A_1 = a', U_1 = u\right\}\right]\Pr(A_1 = a', U_1 = u)$$

$$= \sum_{a'}\left[\sum_{a,o}\nu_1(a \mid o, a')r^{\top}\Pr(R_1 = r, O_1 = o|A_1 = a, U_1 = u)\right]\Pr(A_1 = a', U_1 = u)$$

$$= \sum_{a'}\left[\sum_{a,o}\nu_1(a \mid o, a')r^{\top}P_{oa}^{(1,r)}[P_a^{(1)}]^{-1}\Pr((O_1 = o|U_1 = u)\right]\Pr(A_1 = a', U_1 = u)$$

$$= \sum_{o,a,a'}\nu_1(a \mid o, a')r^{\top}P_{oa}^{(1,r)}[P_a^{(1)}]^{-1}\Pr(O_1 = o, A_1 = a')$$

Next, consider $\mathbb{E}^\nu R_2$. According to the Markov property, $R_2$ and $O_2$ are conditionally independent of $(A_1, U_0, O_1)$ given $(A_1, U_1)$. As such, we have that

$$\mathbb{E}^\nu R_2 = \sum_{o_1, a_1', a_1} \nu_1(a_1|o_1, a_1')$$

$$\sum_{s_2, a_2', o_2} \left\{ \left[ \sum_{a_2} \nu_2(a_2|o_2, o_1, a_2', a_1') \boldsymbol{r}^\top \Pr(R_2 = \boldsymbol{r}, O_2 = o_2 | A_2 = a_2, U_2 = \boldsymbol{u}) \right] \right.$$

$$\left. \Pr(U_2 = \boldsymbol{u}, A_2 = a_2', O_1 = o_1 | A_1 = a_1, U_1 = \boldsymbol{u}) \right\} \Pr(U_1 = \boldsymbol{u}, A_1 = a_1').$$

$$= \sum_{o_1, a_1', a_1} \nu_1(a_1|o_1, a_1') \sum_{a_2', o_2} \left\{ \left[ \sum_{a_2} \nu_2(a_2|o_2, a_1', o_1, a_2') \boldsymbol{r}^\top \boldsymbol{P}^{(2,r)}_{o_2, a_2} [\boldsymbol{P}^{(2)}_{a_2}]^{-1} \Pr((O_2 = \boldsymbol{o}|U_2 = \boldsymbol{u}) \right] \boldsymbol{P}^{(1,u)}_{o_1, a_2', a_1} \right\}$$

$$[\boldsymbol{P}^{(1)}_{a_1}]^{-1} \Pr((O_1 = \boldsymbol{o}|U_1 = \boldsymbol{u}) \Pr(U_1 = \boldsymbol{u}, A_1 = a_1')$$

$$= \sum_{o_1, a_1', a_1} \nu_1(a_1|o_1, a_1') \left[ \sum_{a_2', o_2, a_2} \nu_2(a_2|o_2, a_1', o_1, a_2') \boldsymbol{r}^\top \boldsymbol{P}^{(2,r)}_{o_2, a_2} [\boldsymbol{P}^{(2)}_{a_2}]^{-1} \boldsymbol{P}^{(1,o)}_{o_1, a_2', a_1} \right]$$

$$[\boldsymbol{P}^{(1)}_{a_1}]^{-1} \Pr(O_1 = \boldsymbol{o}, A_1 = a_1').$$

where $\boldsymbol{P}^{(t,o)}_{o_t, a_{t+1}', a_t} = \Pr(O_{t+1} = \boldsymbol{o}, A_t = a_{t+1}', O_t = o_t | A_t = a_t, O_{t-1} = \boldsymbol{o})$. Follow the similar argument, one can derive the identification formula for $t = 3, \ldots, T$. □

## H PROOF IN SECTION 5

*Proof of Lemma 5.1.*

$$\mathcal{V}(\nu^*) - \mathcal{V}(\hat{\nu}^*)$$

$$= \mathbb{E} \left\{ \mathbb{E} \left[ \sum_{a \in \mathcal{A}} q(W, S, a)\nu^*(a \mid S, Z, A) \mid S, Z, A \right] - \mathbb{E} \left[ \sum_{a \in \mathcal{A}} q(W, S, a)\hat{\nu}^*(a \mid S, Z, A) \mid S, Z, A \right] \right\}$$

$$\leq \mathbb{E} \left\{ \mathbb{E} \left[ \sum_{a \in \mathcal{A}} q(W, S, a)\nu^*(a \mid S, Z, A) \mid S, Z, A \right] - \hat{\mathbb{E}} \left[ \sum_{a \in \mathcal{A}} \hat{q}(W, S, a)\nu^*(a \mid S, Z, A) \mid S, Z, A \right] \right\}$$

$$+ \mathbb{E} \left\{ \hat{\mathbb{E}} \left[ \sum_{a \in \mathcal{A}} \hat{q}(W, S, a)\hat{\nu}^*(a \mid S, Z, A) \mid S, Z, A \right] - \mathbb{E} \left[ \sum_{a \in \mathcal{A}} q(W, S, a)\hat{\nu}^*(a \mid S, Z, A) \mid S, Z, A \right] \right\} \quad (26)$$

$$\leq 2\xi_n + \mathbb{E} \left\{ \sum_{a \in \mathcal{A}} q(W, S, a)\nu^*(a \mid S, Z, A) - \sum_{a \in \mathcal{A}} \hat{q}(W, S, a)\nu^*(a \mid S, Z, A) \right\} \quad (27)$$

$$+ \mathbb{E} \left\{ \sum_{a \in \mathcal{A}} \hat{q}(W, S, a)\hat{\nu}^*(a \mid S, Z, A) - \sum_{a \in \mathcal{A}} q(W, S, a)\hat{\nu}^*(a \mid S, Z, A) \right\}$$

$$= 2\xi_n + \mathbb{E} \left\{ (q(W, S, A') - \hat{q}(W, S, A')) \frac{\sum_{a \in \mathcal{A}} \pi_b(a \mid U, S)\nu^*(A' \mid Z, S, a)}{\pi_b(A' \mid U, S)} \right\} \quad (28)$$

$$+ \mathbb{E} \left\{ (q(W, S, A') - \hat{q}(W, S, A')) \frac{\sum_{a \in \mathcal{A}} \pi_b(a \mid U, S)\hat{\nu}^*(A' \mid Z, S, a)}{\pi_b(A' \mid U, S)} \right\} \leq 2(\xi_n + p_{\max}\zeta_n),$$

where equation 26 is due to the optimality of $\hat{q}$ and equation 27 is due to the definition of $\xi_n$. □

*Proof of Theorem 5.1.* The bound in Theorem 5.1 can be derived by combining the results of Theorem I.2, Theorem I.4 and Lemma I.2. □

In the following, we derive the regret bound stated in Section 5.2. Before that, we present the following regret decomposition lemma. Define function class $\tilde{Q}^{(t)}$ over $\mathcal{W} \times \mathcal{S}$ such that $\tilde{Q}^{(t)} := \{q(\cdot, \cdot, a) : q \in \mathcal{Q}^{(t)}, a \in \mathcal{A}\}$.

**Lemma H.1.** Suppose $f_t \in \mathcal{Q}^{(t)} \subset \mathcal{W} \times \mathcal{S} \times \mathcal{A}$ and take the policy $\nu_f = \{\nu_{f,t}\}_{t=1}^T$ as the one that is greedy with respect to $\hat{\mathbb{E}}[f_t(W_t, S_t, a) \mid S_t, Z_t, A_t]$. Take $g_t(S_t, Z_t, A_t; \tilde{q}) := \mathbb{E}[\tilde{q}(W_t, S_t) \mid S_t, Z_t, A_t]$ and $\hat{g}_t(S_t, Z_t, A_t; \tilde{q}) := \hat{\mathbb{E}}[\tilde{q}(W_t, S_t) \mid S_t, Z_t, A_t]$ for $\tilde{q} \in \tilde{Q}^{(t)}$. Define the projection error

$$\xi_{t,n} := \sup_{\tilde{q} \in \tilde{Q}^{(t)}} \|g_t(\cdot, \cdot, \cdot; \tilde{q}) - \hat{g}_t(\cdot, \cdot, \cdot; \tilde{q})\|_2,$$

and

$$\zeta_{t,n}^f := \left\| \mathbb{E}\left\{ f_t(W_t, S_t, A_t) - \left[ R_t + \sum_{a \in \mathcal{A}} f_{t+1}(W_{t+1}, S_{t+1}, A_{t+1})\nu_{f,t+1}(a \mid S_{t+1}, Z_{t+1}, A_{t+1}) \right] \mid S_t, Z_t, A_t \right\} \right\|_2.$$

Define

$$p_{t,\max} := \sup_{s,z,a} \frac{p_t^{\nu^*}(S_t = s, Z_t = z, A_t = a)}{p_t^{\pi_b}(S_t = s, Z_t = z, A_t = a)},$$

and

$$p_{\max,t}^{\nu} = \sup_{s,z,a} \omega_t^{\nu}(S_t = s, Z_t = z, A_t = a),$$

where

$$\omega_t^{\nu}(S_t, Z_t, A_t) := \frac{\sum_{a \in \mathcal{A}}(\int_{u \in \mathcal{U}} \pi_b(a \mid U_t = u, S_t)p_t^{\pi_b}(u \mid S_t)du)\nu(A_t \mid S_t, Z_t, a)}{\int_u \pi_b(A_t \mid U_t = u, S_t)p_t^{\pi}(u \mid S_t, Z_t)du} \frac{p_t^{\nu}(S_t, Z_t)}{p_t^{\pi_b}(S_t, Z_t)}.$$

$$\tag{29}$$

Then under Assumption 3, 8, 9 and 10, together with Assumption 6, we can obtain the following regret bound

$$\mathcal{V}(\nu^*) - \mathcal{V}(\nu_f) \leq \left( \sum_{t=1}^T 2p_{t,\max}\xi_{t,n} \right) + \sqrt{T \sum_{t=1}^T [(p_{t,\max}^{\nu^*})^2 + (p_{t,\max}^{\nu_f})^2](\zeta_{t,n}^f)^2}.$$

*Proof of Lemma H.1.* We start from the decomposition

$$\mathcal{V}(\nu^*) - \mathcal{V}(\nu_f)$$

$$\leq \mathcal{V}(\nu^*) - \mathbb{E}\left[ \sum_{a \in \mathcal{A}} f_1(W_1, S_1, a)\nu_1^*(a \mid S_1, Z_1, A_1) \right]$$

$$+ \mathbb{E}\left[ \sum_{a \in \mathcal{A}} f_1(W_1, S_1, a)\nu_1^*(a \mid S_1, Z_1, A_1) \right] - \mathbb{E}\left[ \hat{\mathbb{E}}\left\{ \sum_{a \in \mathcal{A}} f_1(W_1, S_1, a)\nu_1^*(a \mid S_1, Z_1, A_1) \mid S_1, Z_1, A_1 \right\} \right]$$

$$+ \mathbb{E}\left[ \hat{\mathbb{E}}\left\{ \sum_{a \in \mathcal{A}} f_1(W_1, S_1, a)\hat{\nu}_{f,1}(a \mid S_1, Z_1, A_1) \mid S_1, Z_1, A_1 \right\} \right] - \mathbb{E}\left[ \sum_{a \in \mathcal{A}} f_1(W_1, S_1, a)\nu_{f,1}(a \mid S_1, Z_1, A_1) \right]$$

$$+ \mathbb{E}\left[ \sum_{a \in \mathcal{A}} f_1(W_1, S_1, a)\nu_{f,1}(a \mid S_1, Z_1, A_1) \right] - \mathcal{V}(\hat{\nu}^*)$$

$$\leq 2\xi_{1,n} + \mathcal{V}(\nu^*) - \mathbb{E}\left[ \sum_{a \in \mathcal{A}} f_1(W_1, S_1, a)\nu_1^*(a \mid S_1, Z_1, A_1) \right] + \mathbb{E}\left[ \sum_{a \in \mathcal{A}} f_1(W_1, S_1, a)\nu_{f,1}(a \mid S_1, Z_1, A_1) \right] - \mathcal{V}(\nu_f)$$

$$\tag{30}$$

First, we can show that for any policy $\nu \in \Omega$,

$$
\mathbb{E}\left\{\sum_{a \in \mathcal{A}} f_1(W_1, S_1, a)\nu_1(a \mid S_1, Z_1, A_1)\right\} - \mathcal{V}(\nu)
$$

$$
=\mathbb{E}\left\{\sum_{a \in \mathcal{A}} f_1(W_1, S_1, a)\nu_1(a \mid S_1, Z_1, A_1)\right\} - \mathbb{E}^\nu\left[\sum_{t=1}^T R_t\right]
$$

$$
=\mathbb{E}^\nu \sum_{t=1}^T \left\{\left[\sum_{a \in \mathcal{A}} f_t(W_t, S_t, a)\nu_t(a \mid S_t, Z_t, A_t)\right] - \mathbb{E}^\nu\left[R_t + \sum_{a \in \mathcal{A}} f_{t+1}(W_{t+1}, S_{t+1}, a)\nu_{t+1}(a \mid S_{t+1}, Z_{t+1}, A_{t+1})\right]\right\}
$$

$$(31)$$

At time $t$, because of the optimality of $\nu_{f,t}$, we have

$$
\hat{\mathbb{E}}\left\{\sum_{a \in \mathcal{A}} f_t(W_t, S_t, a)\nu_t^*(a \mid S_t, Z_t, A_t) \mid S_t, Z_t, A_t\right\} \le \hat{\mathbb{E}}\left\{\sum_{a \in \mathcal{A}} f_t(W_t, S_t, a)\nu_{f,t}(a \mid S_t, Z_t, A_t) \mid S_t, Z_t, A_t\right\}.
$$

Then

$$
\mathbb{E}\left\{\sum_{a \in \mathcal{A}} f_t(W_t, S_t, a)\nu_t^*(a \mid S_t, Z_t, A_t) \mid S_t, Z_t, A_t\right\} - \mathbb{E}\left\{\sum_{a \in \mathcal{A}} f_t(W_t, S_t, a)\nu_{f,t}(a \mid S_t, Z_t, A_t) \mid S_t, Z_t, A_t\right\}
$$

$$
\le\mathbb{E}\left\{\sum_{a \in \mathcal{A}} f_t(W_t, S_t, a)\nu_t^*(a \mid S_t, Z_t, A_t) \mid S_t, Z_t, A_t\right\} - \hat{\mathbb{E}}\left\{\sum_{a \in \mathcal{A}} f_t(W_t, S_t, a)\nu_t^*(a \mid S_t, Z_t, A_t) \mid S_t, Z_t, A_t\right\}
$$

$$
+ \hat{\mathbb{E}}\left\{\sum_{a \in \mathcal{A}} f_t(W_t, S_t, a)\nu_{f,t}(a \mid S_t, Z_t, A_t) \mid S_t, Z_t, A_t\right\} - \mathbb{E}\left\{\sum_{a \in \mathcal{A}} f_t(W_t, S_t, a)\nu_{f,t}(a \mid S_t, Z_t, A_t) \mid S_t, Z_t, A_t\right\},
$$

$$(32)$$

and

$$
\mathbb{E}\left\{\sum_{a \in \mathcal{A}} f_t(W_t, S_t, a)\nu_t^*(a \mid S_t, Z_t, A_t) - \sum_{a \in \mathcal{A}} f_t(W_t, S_t, a)\nu_{f,t}(a \mid S_t, Z_t, A_t)\right\}
$$

$$
\le\mathbb{E}^{1/2}\left\{\left[\mathbb{E}\sum_{a \in \mathcal{A}} f_t(W_t, S_t, a)\nu_t^*(a \mid S_t, Z_t, A_t) - \sum_{a \in \mathcal{A}} f_t(W_t, S_t, a)\nu_{f,t}(a \mid S_t, Z_t, A_t) \mid S_t, Z_t, A_t\right]^2\right\} \le 2\xi_{t,n}.
$$

$$(33)$$

The last inequality is due to the decomposition equation 32 and the definition of $\xi_{t,n}$.

Note that

$$
\mathbb{E}^{\nu^*}\left\{\sum_{a \in \mathcal{A}} f_t(W_t, S_t, a)\nu_t^*(a \mid S_t, Z_t, A_t) \mid S_t, Z_t, A_t\right\} - \mathbb{E}^{\nu^*}\left\{\sum_{a \in \mathcal{A}} f_t(W_t, S_t, a)\nu_{f,t}(a \mid S_t, Z_t, A_t) \mid S_t, Z_t, A_t\right\}
$$

$$
=\mathbb{E}^{\nu^*}\left\{\mathbb{E}^{\nu^*}\left[\sum_{a \in \mathcal{A}} f_t(W_t, S_t, a)\left(\nu_t^*(a \mid S_t, Z_t, A_t) - \nu_{f,t}(a \mid S_t, Z_t, A_t)\right) \mid U_t, S_t, Z_t, A_t\right] \mid S_t, Z_t, A_t\right\}
$$

$$
=\mathbb{E}^{\nu^*}\left\{\mathbb{E}\left[\sum_{a \in \mathcal{A}} f_t(W_t, S_t, a)\left(\nu_t^*(a \mid S_t, Z_t, A_t) - \nu_{f,t}(a \mid S_t, Z_t, A_t)\right) \mid U_t, S_t, Z_t, A_t\right] \mid S_t, Z_t, A_t\right\}
$$

$$
=\mathbb{E}\left\{\frac{p_t^{\nu^*}(U_t \mid S_t, Z_t, A_t)}{p_t^b(U_t \mid S_t, Z_t, A_t)}\mathbb{E}\left[\sum_{a \in \mathcal{A}} f_t(W_t, S_t, a)\left(\nu_t^*(a \mid S_t, Z_t, A_t) - \nu_{f,t}(a \mid S_t, Z_t, A_t)\right) \mid U_t, S_t, Z_t, A_t\right] \mid S_t, Z_t, A_t\right\}.
$$

Due to Assumption 6, we have $p_t^{\nu^*}(U_t \mid S_t) = p_t^{\pi_b}(U_t \mid S_t)$ and

$$
p_t^{\nu^*}(U_t \mid S_t, Z_t, A_t) = \frac{p_t^{\nu^*}(Z_t, A_t \mid U_t, S_t)p_t^{\nu^*}(U_t \mid S_t)}{\int_{u \in \mathcal{U}} p_t^{\nu^*}(Z_t, A_t \mid U_t = u, S_t)p_t^{\nu^*}(U_t = u \mid S_t)du}
$$

$$
= \frac{p_t^{\pi_b}(Z_t, A_t \mid U_t, S_t)p_t^{\pi_b}(U_t \mid S_t)}{\int_{u \in \mathcal{U}} p_t^{\pi_b}(Z_t, A_t \mid U_t = u, S_t)p_t^{\pi_b}(U_t = u \mid S_t)du} = p_t^{\pi_b}(U_t \mid S_t, Z_t, A_t).
$$

Therefore,

$$
\mathbb{E}^{\nu^*}\left[\mathbb{E}^{\nu^*}\left\{\sum_{a\in\mathcal{A}}f_t(W_t,S_t,a)\nu_t^*(a\mid S_t,Z_t,A_t)\mid S_t,Z_t,A_t\right\}-\mathbb{E}^{\nu^*}\left\{\sum_{a\in\mathcal{A}}f_t(W_t,S_t,a)\nu_{f,t}(a\mid S_t,Z_t,A_t)\mid S_t,Z_t,A_t\right\}\right]
$$

$$
=\mathbb{E}^{\nu^*}\left[\mathbb{E}\left\{\sum_{a\in\mathcal{A}}f_t(W_t,S_t,a)\nu_t^*(a\mid S_t,Z_t,A_t)\mid S_t,Z_t,A_t\right\}-\mathbb{E}\left\{\sum_{a\in\mathcal{A}}f_t(W_t,S_t,a)\nu_{f,t}(a\mid S_t,Z_t,A_t)\mid S_t,Z_t,A_t\right\}\right]
$$

$$
=\mathbb{E}\left[\frac{p_t^{\nu^*}(S_t,Z_t,A_t)}{p_t^{\pi_b}(S_t,Z_t,A_t)}\mathbb{E}\left\{\sum_{a\in\mathcal{A}}f_t(W_t,S_t,a)(\nu_t^*(a\mid S_t,Z_t,A_t)-\nu_{f,t}(a\mid S_t,Z_t,A_t))\mid S_t,Z_t,A_t\right\}\right]
$$

$$
\leq 2p_{\max,t}\xi_{t,n}. \tag{34}
$$

The last inequality is due to equation 33 and the definition of $p_{\max,t}$.
Now let's go back to equation 30, we have

$$
\mathbb{E}\left[\sum_{a\in\mathcal{A}}f_1(W_1,S_1,a)\nu_{f,1}(a\mid S_1,Z_1,A_1)\right]-\mathcal{V}(\nu_{f,t})
$$

$$
=\mathbb{E}^{\nu_f}\sum_{t=1}^T\left\{\left[\sum_{a\in\mathcal{A}}f_t(W_t,S_t,a)\nu_t(a\mid S_t,Z_t,A_t)\right]-\mathbb{E}^{\nu_f}\left[R_t+\sum_{a\in\mathcal{A}}f_{t+1}(W_{t+1},S_{t+1},a)\nu_{f,t+1}(a\mid S_{t+1},Z_{t+1},A_{t+1})\right]\right\},
$$

because of equation 31, and

$$
\mathbb{E}\left[\sum_{a\in\mathcal{A}}f_1(W_1,S_1,a)\nu_t^*(a\mid S_1,Z_1,A_1)\right]-\mathcal{V}(\nu^*)
$$

$$
=\mathbb{E}^{\nu^*}\sum_{t=1}^T\left\{\left[\sum_{a\in\mathcal{A}}f_t(W_t,S_t,a)\nu_t^*(a\mid S_t,Z_t,A_t)\right]-\mathbb{E}^{\nu^*}\left[R_t+\sum_{a\in\mathcal{A}}f_{t+1}(W_{t+1},S_{t+1},a)\nu_{t+1}^*(a\mid S_{t+1},Z_{t+1},A_{t+1})\right]\right\}
$$

$$
\geq\sum_{t=1}^T\mathbb{E}^{\nu^*}\left[\sum_{a\in\mathcal{A}}f_t(W_t,S_t,a)\nu_t^*(a\mid S_t,Z_t,A_t)\right]-\mathbb{E}^{\nu^*}\left[R_t+\sum_{a\in\mathcal{A}}f_{t+1}(W_{t+1},S_{t+1},a)\nu_{f,t+1}(a\mid S_{t+1},Z_{t+1},A_{t+1})\right]
$$

$$
-2p_{t+1,\max}\xi_{t+1,n}
$$

because of equation 34. Then

$$
\mathcal{V}(\nu^*)-\mathcal{V}(\nu_f)
$$

$$
\leq 2\xi_{1,n}+\mathbb{E}^{\nu_f}\sum_{t=1}^T\left\{\left[\sum_{a\in\mathcal{A}}f_t(W_t,S_t,a)\nu_{f,t}(a\mid S_t,Z_t,A_t)\right]\right.
$$

$$
-\mathbb{E}^{\nu_f}\left[R_t+\sum_{a\in\mathcal{A}}f_{t+1}(W_{t+1},S_{t+1},a)\nu_{f,t+1}(a\mid S_{t+1},Z_{t+1},A_{t+1})\mid S_t,Z_t,A_t\right]\right\}
$$

$$
-\mathbb{E}^{\nu^*}\sum_{t=1}^T\left\{\left[\sum_{a\in\mathcal{A}}f_t(W_t,S_t,a)\nu_t^*(a\mid S_t,Z_t,A_t)\right]\right.
$$

$$
+\mathbb{E}^{\nu^*}\left[R_t+\sum_{a\in\mathcal{A}}f_{t+1}(W_{t+1},S_{t+1},a)\nu_{f,t+1}(a\mid S_{t+1},Z_{t+1},A_{t+1})\mid S_t,Z_t,A_t\right]\right\}+\sum_{t=2}^T 2p_{t,\max}\xi_{t,n}
$$

We know that for $\nu,\in\{\nu^*,\nu_f\}$,

$$
\mathbb{E}^{\nu}\left[R_t+\sum_{a\in\mathcal{A}}f_{t+1}(W_{t+1},S_{t+1},a)\nu_{f,t+1}(a\mid S_{t+1},Z_{t+1},A_{t+1})\mid U_t,S_t,Z_t,A_t\right]
$$

$$
=\mathbb{E}^{\nu}\left[\sum_{a\in\mathcal{A}}\mathbb{E}\left\{R_t+\sum_{a'\in\mathcal{A}}f_{t+1}(W_{t+1},S_{t+1},a')\nu_{f,t+1}(a'\mid S_{t+1},Z_{t+1},A_{t+1})\mid U_t,S_t,Z_t,A_t=a\right\}\nu_t(a\mid S_t,Z_t,A_t)\right]
$$

Take

$$\omega_t^\nu(S_t, Z_t, A_t) = \frac{\sum_{a \in \mathcal{A}}(\int_{u \in \mathcal{U}} \pi_b(a \mid U_t = u, S_t) p_t^{\pi_b}(u \mid S_t, Z_t) du) \nu(A_t \mid S_t, Z_t, a)}{\int_u \pi_{\pi_b}(A_t \mid U_t = u, S_t) p_t^\pi(u \mid S_t, Z_t) du} \frac{p_t^\nu(S_t, Z_t)}{p_t^{\pi_b}(S_t, Z_t)}.$$

Then at any $t$,

$$\mathbb{E}^{\nu_f} \left\{ \left[ \sum_{a \in \mathcal{A}} f_t(W_t, S_t, a) \nu_{f,t}(a \mid S_t, Z_t, A_t) \right] \right.$$

$$\left. - \mathbb{E}^{\nu_f} \left[ R_t + \sum_{a \in \mathcal{A}} f_{t+1}(W_{t+1}, S_{t+1}, a) \nu_{f,t+1}(a \mid S_{t+1}, Z_{t+1}, A_{t+1}) \mid S_t, Z_t, A_t \right] \right\}$$

$$= \mathbb{E}^{\nu_f} \left\{ \sum_{a \in \mathcal{A}} \nu_{f,t}(a \mid S_t, Z_t, A_t) \right.$$

$$\left. \mathbb{E} \left[ f_t(W_t, S_t, A_t) - \left( R_t + \sum_{a' \in \mathcal{A}} f_{t+1}(W_{t+1}, S_{t+1}, a') \nu_{f,t+1}(a' \mid S_{t+1}, Z_{t+1}, A_{t+1}) \right) \mid U_t, S_t, Z_t, A_t = a \right] \right\}$$

$$= \mathbb{E}^{\nu_f} \left\{ \sum_{a \in \mathcal{A}} \nu_{f,t}(a \mid S_t, Z_t, A_t) \right.$$

$$\left. \mathbb{E} \left[ f_t(W_t, S_t, A_t) - \left( R_t + \sum_{a' \in \mathcal{A}} f_{t+1}(W_{t+1}, S_{t+1}, a') \nu_{f,t+1}(a' \mid S_{t+1}, Z_{t+1}, A_{t+1}) \right) \mid S_t, Z_t, A_t = a \right] \right\}$$

$$= \mathbb{E} \left\{ \omega^{\nu_f}(S_t, Z_t, A_t) \left[ f_t(W_t, S_t, A_t) - \left( R_t + \sum_{a' \in \mathcal{A}} f_{t+1}(W_{t+1}, S_{t+1}, a') \nu_{f,t+1}(a' \mid S_{t+1}, Z_{t+1}, A_{t+1}) \right) \right] \right\}.$$

The second equality is due to that $p_t^{\pi_b}(U_t \mid S_t, Z_t, A_t = a) = p_t^{\nu_f}(U_t \mid S_t, Z_t, A_t = a)$.

$$\left| \sum_{t=1}^T \mathbb{E}^{\nu_f} \left\{ \left[ \sum_{a \in \mathcal{A}} f_t(W_t, S_t, a) \nu_{f,t}(a \mid S_t, Z_t, A_t) \right] - \left[ R_t + \sum_{a \in \mathcal{A}} f_{t+1}(W_{t+1}, S_{t+1}, a) \nu_{f,t+1}(a \mid S_{t+1}, Z_{t+1}, A_{t+1}) \right] \right\} \right|$$

$$\leq \left( T \sum_{t=1}^T (\mathbb{E} \{ \omega^{\nu_f}(S_t, Z_t, A_t) [f_t(W_t, S_t, A_t) \right.$$

$$\left. - \left( R_t + \sum_{a' \in \mathcal{A}} f_{t+1}(W_{t+1}, S_{t+1}, a') \nu_{f,t+1}(a' \mid S_{t+1}, Z_{t+1}, A_{t+1}) \right) \right] \mid S_t, Z_t, A_t \})^2 \right)^{1/2}$$

$$\leq \sqrt{T \sum_{t=1}^T (p_{\max,t}^{\nu_f})^2 (\zeta_{t,n}^f)^2}$$

Similarly, we have

$$\left| \sum_{t=1}^T \mathbb{E}^{\nu_f} \left\{ \left[ \sum_{a \in \mathcal{A}} f_t(W_t, S_t, a) \nu_{f,t}(a \mid S_t, Z_t, A_t) \right] - \left[ R_t + \sum_{a \in \mathcal{A}} f_{t+1}(W_{t+1}, S_{t+1}, a) \nu_{f,t+1}(a \mid S_{t+1}, Z_{t+1}, A_{t+1}) \right] \right\} \right|$$

$$\leq \sqrt{T \sum_{t=1}^T (p_{\max,t}^{\nu^*})^2 (\zeta_{t,n}^f)^2}$$

Therefore, overall we have

$$\mathcal{V}(\nu^*) - \mathcal{V}(\nu_f) \leq \left( \sum_{t=1}^T 2 p_{t,\max} \xi_{t,n} \right) + \sqrt{T \sum_{t=1}^T [(p_{t,\max}^{\nu^*})^2 + (p_{t,\max}^{\nu_f})^2](\zeta_{t,n}^f)^2}.$$

$\square$

*Proof of Lemma 5.2.* Proof of Lemma 5.2 is a direct adaption of Lemma H.1. □

*Proof of Theorem 5.2.* The result is concluded by directly combining Theorems I.1, I.3 and Lemma I.1. □

# I MIN-MAX CONDITIONAL MOMENT ESTIMATION AND PROJECTION ESTIMATION

## I.1 MIN-MAX CONDITIONAL MOMENT ESTIMATION

We take the min-max estimation procedure to solve the estimation equation equation 10. More specifically, we follow the construction in Dikkala et al. (2020) and propose the following estimators for $Q$-bridge functions. For the following discussion, without loss of generality, we assume $\max |R_t| \leq 1$ for $t = 1, \ldots, T$, and function spaces $\mathcal{Q}^{(t)}$, $\mathcal{G}^{(t)}$ $\mathcal{H}^{(t)}$ below are classes of bounded functions whose image is a subset of $[-1, 1]$. Take $\hat{q}_{T+1} = 0$. For $t = T, \ldots, 1$,

$$\hat{q}_t = (T - t + 1)\arg\min_{q \in \mathcal{Q}^{(t)}} \sup_{g \in \mathcal{G}^{(t)}} \Psi_n(q, \hat{V}_{t+1}, g) - \lambda\left(\|g\|_{\mathcal{G}^{(t)}}^2 + \frac{U}{\Delta^2}\|g\|_{2,n}^2\right) + \lambda\mu\|q\|_{\mathcal{Q}^{(t)}}^2, \quad (35)$$

where $\| \cdot \|_{2,n}$ is the empirical norm, $\lambda$, $U$, $\delta$ and $\mu$ are positive tuning parameters, and

$$\Psi_n(q, \hat{V}_{t+1}, g) = \frac{1}{n}\sum_{i=1}^n\left\{q(W_{i,t}, S_{i,t}, A_{i,t}) - \frac{R_{i,t} + \hat{V}_{t+1}(W_{i,t+1}, S_{i,t+1}, Z_{i,t+1}, A_{i,t+1})}{T - t + 1}\right\}g(Z_{i,t}, S_{i,t}, A_{i,t}),$$

$$\hat{V}_{t+1}(W_{i,t+1}, S_{i,t+1}, Z_{i,t+1}, A_{i,t+1}) = \sum_{a \in \mathcal{A}}\hat{q}_{t+1}(W_{i,t+1}, S_{i,t+1}, a)\hat{\nu}_{t+1}^*(a \mid S_{i,t+1}, Z_{i,t+1}, A_{i,t+1}).$$
$$(36)$$

In the following, we utilize a uniform error bound to study $\xi_{t,n}$. Define the operator $\mathcal{T}_t = \bar{\mathcal{T}}_t^{-1}\tilde{\mathcal{T}}_t$, where $[\tilde{\mathcal{T}}_t h](S_t, Z_t, A_t) = \mathbb{E}[h(R_t, W_{t+1}, S_{t+1}, Z_{t+1}, A_{t+1}) \mid S_t, Z_t, A_t]$ for $h \in \mathcal{L}^2\{\mathcal{R} \times \mathcal{W} \times \mathcal{S} \times \mathcal{Z} \times \mathcal{A}\}$ and $[\bar{\mathcal{T}}_t q](S_t, Z_t, A_t) = \mathbb{E}[q(W_t, S_t, A_t) \mid S_t, Z_t, A_t]$ for $h \in \mathcal{L}^2\{\mathcal{W} \times \mathcal{S} \times \mathcal{A}\}$. And take $[\langle\nu, q\rangle](W_t, S_t, Z_t, A_t) = \sum_{a \in \mathcal{A}}\hat{q}(W_t, S_t, a)\hat{\nu}(a \mid S_t, Z_t, A_t)$. For a function space $\mathcal{F}$, we define $\alpha\mathcal{F} = \{\alpha f : f \in \mathcal{F}\}$ and $\mathcal{F}_B = \{f \in \mathcal{F} : \|f\|_{\mathcal{F}}^2 \leq B\}$.

**Assumption 15.** The following conditions hold for $t = 1, \ldots, T$.

(a) For any $\nu \in \mathcal{V}$ and $q \in \mathcal{Q}^{(t)}$, $\langle\nu, q\rangle \in \mathcal{H}^{(t)}$. For any $h \in \mathcal{H}^{(t+1)}$, $\mathcal{T}_t(h + R_t) \in \mathcal{Q}^{(t)}$.

(b) For any $q \in (T - t)\mathcal{Q}^{(t+1)}$ and any $\nu \in \mathcal{V}$, we have $\left\|\mathcal{T}_t\left(\frac{R_t + \langle\nu, q\rangle}{T - t + 1}\right)\right\|_{\mathcal{Q}^{(t)}}^2 \leq \left\|\frac{q}{T - t}\right\|_{\mathcal{Q}^{(t+1)}}^2$.

(c) For any $q \in \mathcal{Q}^{(t)}$ and $\nu \in \mathcal{V}$, we have $\|\langle\nu, q\rangle\|_{\mathcal{H}^{(t)}}^2 \leq C_v\|q\|_{\mathcal{Q}^{(t)}}^2$ for some constant $C_v > 0$.

(d) There exists $L > 0$ such that $\|g^* - \bar{\mathcal{T}}_t q_t\|_2 \leq \varrho_{t,n}$, where $g^* \in \arg\min_{g \in \mathcal{G}^{(t)}_{L^2\|q_t\|_{\mathcal{Q}^{(t)}}^2}}\|g - \bar{\mathcal{T}}_t q_t\|_2$ for all $q_t \in \mathcal{Q}^{(t)}$.

Take $\mathcal{Q}_B^{(t)}$, $\mathcal{H}_D^{(t)}$ and $\mathcal{G}_{3U}^{(t)}$ as balls in $\mathcal{Q}^{(t)}$, $\mathcal{H}^{(t)}$ and $\mathcal{G}^{(t)}$ respectively for some fixed constants $B, D, U > 0$ such that functions in $\mathcal{Q}_B^{(t)}$, $\mathcal{H}_D^{(t)}$ and $\mathcal{G}_{3U}^{(t)}$ are uniformly bounded by 1. Consider the following two spaces:

$$\boldsymbol{\Omega}^{(t)} = \{(w_t, s_t, z_t, a_t, w_{t+1}, s_{t+1}, z_{t+1}, a_{t+1}) \mapsto r[q_h^*(w_t, s_t, a_t) - h(w_{t+1}, s_{t+1}, z_{t+1}, a_{t+1})]g(z_t, s_t, a_t) :$$
$$h \in \mathcal{H}_D^{(t+1)}, g \in \mathcal{G}_{3U}^{(t)}, r \in [0, 1]\}$$

$$\boldsymbol{\Xi}^{(t)} = \left\{(w_t, s_t, z_t, a_t) \mapsto r[q - q_h^*(w_t, s_t, a_t)]g^{L^2 B}(z_t, s_t, a_t) : \right.$$
$$\left. q \in \mathcal{Q}^{(t)}, q - q_h^* \in \mathcal{Q}_B^{(t)}, h \in \mathcal{H}_D^{(t+1)}, r \in [0, 1]\right\},$$

where $q_h^* \in \mathcal{Q}^{(t)}$ is the solution to $\mathbb{E}[q(W_t, S_t, A_t) - h(W_{t+1}, S_{t+1}, Z_{t+1}, A_{t+1}) \mid Z_t, S_t, A_t] = 0$ and $g^{L^2 B} = \arg\min_{g \in \mathcal{G}_{L^2 B}^{(t)}} \|g - \tilde{\mathcal{T}}_t(q - q_h^*)\|_2$ for a given $L > 0$.

We use the Rademacher complexity to characterize the complexity of a function class. For a generic real-valued function space $\mathcal{F} \subset \mathbb{R}^X$, the local Rademacher complexity with radius $\delta > 0$ is defined as

$$\mathcal{R}_n(\mathcal{F}, r) = \left( \sup_{f \in \mathcal{F}, \|f\|_2 \le r^2} \left| \frac{1}{n} \sum_{i=1}^n \epsilon_i f(X_i) \right| \right),$$

where $\{X_i\}_{i=1}^n$ are i.i.d. copies of $X$ and $\{\epsilon_i\}_{i=1}^n$ are i.i.d. Rademacher random variables.

Suppose $\mathcal{F}$ is star-shape and $\|f\|_\infty \le 1$ for $f \in \mathcal{F}$. The critical radius of the local Rademacher complexity $\mathcal{R}_n(\mathcal{F}, r)$, denoted by $r^*$, is the smallest value satisfying $r^2 \ge \mathcal{R}_n(\mathcal{F}, r)$.

**Theorem I.1.** Suppose $\mathcal{G}^{(t)}$, $t = 1, \ldots, T$ are symmetric and start-convex set of test functions and $\|\mathcal{T}_T(R_T)\|_{\mathcal{Q}^T} \le M_Q$. Under Assumption 15, take $\Delta = \tilde{\Delta}_{t,n} + c_0 \sqrt{\log(c_1 T/\delta)/n}$ for some universal constants $c_0, c_1 > 0$, where $\tilde{\Delta}_{t,n}$ is the maximum of critical radius of $\mathcal{G}_{3U}^{(t)}$, $\boldsymbol{\Omega}^{(t)}$ and $\boldsymbol{\Xi}^{(t)}$. Assume that $\varrho_{t,n}$ in Assumption 15(d) $\le \Delta$. Then $(R_t + \hat{V}_{t+1})/(T - t + 1) \in \mathcal{H}_D^{(t+1)}$ with $D = C_v(T - t + 1)M_Q$.

If we further assume tuning parameters satisfy $U\lambda \asymp (\Delta)^2$ and $\mu \ge \mathcal{O}(L^2 + U/B)$, then the following equality holds uniformly for all $t = 1, \ldots, T$ with probability $1 - \delta$:

$$\|\hat{q}_t/(T - t + 1)\|_{\mathcal{Q}^{(t)}}^2 \le (T - t + 2)M_Q,$$

where $\hat{q}_t$ is the solution of equation 35; and

$$\zeta_{t,n} \lesssim M_Q(T - t + 1)^2 (\tilde{\Delta}_{t,n} + \sqrt{\log(c_1 T/\delta)/n}),$$

where

$$\zeta_{t,n} = \left\| \mathbb{E}\left\{ \hat{q}_t(W_t, S_t, A_t) - \left( R_t + \sum_{a \in \mathcal{A}} \hat{V}_{t+1}(W_{t+1}, S_{t+1}, Z_{t+1}, A_{t+1}) \right) \mid S_t, Z_t, A_t \right\} \right\|_2 \quad (37)$$

with $\hat{V}_{t+1}$ defined in equation 36.

*Proof of Theorem I.1.* Proof of Theorem I.1 is a direct adaption of Theorem 6.2 and Lemma D.2 in Miao et al. (2022). □

**Remark 1.** Under the setting of contextual bandits, the $Q$ function estimation can be considered as a special case of equation 35 by setting $t = T$. Then the result of bounding $\zeta_n$ can be adopted from Theorem I.1 accordingly. And we have the following theorem.

**Theorem I.2.** Suppose there exists $q^* \in \mathcal{G}$ that satisfy the $\mathbb{E}[q^* - R \mid S, Z, A] = 0$. The functions in $\mathcal{G}$ and $\mathcal{Q}$ are uniformly bounded by 1. $|R| \le 1$. Take $\Delta = \tilde{\Delta}_n + c_0 \sqrt{\log(c_1/\delta)/n}$ with some positive universal constants $c_0$ and $c_1$, and $\tilde{\Delta}_n$ the maximum of critical radius of $\mathcal{G}_{3U}$ and

$$\boldsymbol{\Xi} = \left\{ (w, s, z, a) \mapsto r[q - q^*](w, s, a)g^{L^2 B}(z, s, a) : q - q^* \in \mathcal{Q}_B, r \in [0, 1] \right\},$$

where $g^{L^2 B} = \arg\min_{f \in \mathcal{G}_{L^2 B}} \|g - \mathbb{E}(q - q^* \mid S, Z, A)\|_2$. In addition, we suppose that for any $q \in \mathcal{Q}$, $\|g^{L^2}\|h - h^*\|_2^2 - \mathbb{E}(q - q^* \mid S, Z, A)\|_2 \lesssim \eta_n \lesssim \Delta$. By taking the tuning parameters $\lambda \approx \Delta^2/U$ and $\mu \gtrsim L^2 + \Delta^2/(B\lambda)$, with probability at least $1 - \delta$, we have

$$\zeta_n \lesssim \tilde{\Delta}_n + \sqrt{\log(c_1/\delta)/n}.$$

## I.2 PROJECTION ESTIMATION

In this section, we discuss how to perform the projection step $\hat{\mathbb{E}}[\hat{q}_t(W_t, S_t, a) \mid S_t = s, Z_t = z, A_t = a']$ in Algorithm 2. Take $\tilde{Q}^{(t)}$ as a space defined over $\mathcal{W} \times \mathcal{S}$ such that $\tilde{\mathcal{Q}}^{(t)} := \{q(\cdot, \cdot, a) : q \in$

$\mathcal{Q}^{(t)}, a \in \mathcal{A}\}$. Take $g_t^*(s, z, a; \tilde{q}) := \mathbb{E}[\tilde{q}(W_t, S_t) \mid S_t = s, Z_t = z, A_t = a]$ for $\tilde{q} \in \tilde{\mathcal{Q}}^{(t)}$. We estimate $g^*$ by

$$\hat{g}_t(\cdot, \cdot, \cdot; \tilde{q}) := \underset{g \in \mathcal{Q}^{(t)}}{\arg\min} \frac{1}{n} \sum_{i=1}^{n} [g(S_{i,t}, Z_{i,t}, A_{i,t}) - \tilde{q}(W_{i,t}, S_{i,t})]^2 + \mu \|g\|_{\mathcal{G}^{(t)}}^2;$$

$$\text{and } \hat{\mathbb{E}}[\hat{q}_t(W_t, S_t, a) \mid S_t = s, Z_t = z, A_t = a'] = (T - t + 1)\hat{g}_t(\cdot, \cdot, \cdot; \hat{q}_t(\cdot, \cdot, a)/(T - t + 1)).$$
$$\tag{38}$$

Take $\tilde{\mathcal{Q}}_{\tilde{B}}^{(t)}$ and $\mathcal{G}_M^{(t)}$ as balls in $\tilde{\mathcal{Q}}$ and $\mathcal{Q}^{(t)}$ respectively for some fixed constants $\tilde{B}$ and $M$ such that functions in $\tilde{\mathcal{Q}}_{\tilde{B}}^{(t)}$ and $\mathcal{G}_M^{(t)}$ are uniformly bounded by 1.

Consider the following space:

$$\Upsilon^{(t)} = \Big\{ (w_t, s_t, z_t, a_t) \mapsto [g(s_t, z_t, a_t) - \tilde{q}(w_t, s_t)]^2 - [g^*(s_t, z_t, a_t; \tilde{q}) - \tilde{q}(w_t, s_t)]^2 :$$
$$g, g^* \in \mathcal{G}_M^{(t)}, \tilde{q} \in \tilde{\mathcal{Q}}_{\tilde{B}}^{(t)} \Big\}$$

**Theorem I.3.** Suppose for any $q \in \mathcal{Q}^{(t)}$ and $a \in \mathcal{A}$, $\|q(\cdot, \cdot, a)\|_{\tilde{\mathcal{Q}}^{(t)}}^2 \le \tilde{C}_v \|q\|_{\mathcal{Q}^{(t)}}^2$; for any $\tilde{q} \in \tilde{\mathcal{Q}}^{(t)}$, $g^*(\cdot, \cdot, \cdot; \tilde{q}) \in \mathcal{G}^{(t)}$ and $\|g^*(\cdot, \cdot, \cdot; \tilde{q})\|_{\mathcal{G}^{(t)}}^2 \le C_g \|\tilde{q}\|_{\tilde{\mathcal{Q}}^{(t)}}^2$. Take $\kappa_{t,n} = \tilde{\kappa}_{t,n} + c_0 \sqrt{\log(c_1 T/\delta)/n}$ for some universal positive constants $c_0$ and $c_1$, where $\tilde{\kappa}_n^{(t)}$ is the critical radius of function space $\Upsilon^{(t)}$. If we further assume the tuning parameter $\mu$ in equation 38 satisfying $\mu \gtrsim (\kappa_{t,n})^2$, then with probability at least $1 - \delta$, we have

$$\text{for any } t = 1, \dots, T, \quad \xi_{t,n} \lesssim (T - t + 1) \left( \kappa_{t,n} \sqrt{1 + \|\hat{q}_t^\pi/(T - t + 1)\|_{\mathcal{Q}^{(t)}}^2} + \sqrt{\mu \|\hat{q}_t^\pi/(T - t + 1)\|_{\mathcal{Q}^{(t)}}^2} \right)$$
$$\lesssim (T - t + 1)^{1.5} \sqrt{M_Q} \kappa_{t,n}.$$

**Remark 2.** Under the setting of contextual bandits, the estimation for the projection (equation 17) can be considered as a special case of equation 38 by setting $t = T$. Then the corresponding result for bounding $\xi_n$ can be obtained by taking $t = T$, $\tilde{\mathcal{Q}} = \{q(\cdot, \cdot, a) : q \in \mathcal{G}, a \in \mathcal{A}\}$ in Theorem I.3. And we obtain

**Theorem I.4.** Suppose for any $q \in \mathcal{Q}$ and $a \in \mathcal{A}$, $\|q(\cdot, \cdot, a)\|_{\tilde{\mathcal{Q}}}^2 \le \tilde{C}_v \|q\|_{\mathcal{Q}}^2$; for any $\tilde{q} \in \tilde{\mathcal{Q}}$, $g^*(\cdot, \cdot, \cdot; \tilde{q}) \in \mathcal{G}$ and $\|g^*(\cdot, \cdot, \cdot; \tilde{q})\|_{\mathcal{G}}^2 \le C_g \|\tilde{q}\|_{\tilde{\mathcal{Q}}}^2$. Take $\kappa_n = \tilde{\kappa}_n + c_0 \sqrt{\log(c_1/\delta)/n}$ for some universal positive constants $c_0$ and $c_1$, where $\tilde{\kappa}_n$ is the critical radius of function space

$$\Upsilon = \Big\{ (w, s, z, a) \mapsto [g(s, z, a) - \tilde{q}(w, s)]^2 - [g^*(s, z, a; \tilde{q}) - \tilde{q}(w, s)]^2 : g, g^* \in \mathcal{G}_M, \tilde{q} \in \tilde{\mathcal{Q}}_{\tilde{B}} \Big\}$$

If we further assume the tuning parameter $\mu$ in equation 17 satisfying $\mu \gtrsim (\kappa_n)^2$, then with probability at least $1 - \delta$, we have

$$\xi_n \lesssim \left( \kappa_n \sqrt{1 + \|\hat{q}^\pi\|_{\mathcal{Q}}^2} + \sqrt{\mu \|\hat{q}^\pi\|_{\mathcal{Q}}^2} \right) \lesssim \kappa_n.$$

*Proof of Theorem I.3.* First, we note that for any $g \in \mathcal{G}^{(t)}$,

$$\mathbb{E}\left[ g(S_t, Z_t, A_t) - \tilde{q}(W_t, S_t) \right]^2 - \mathbb{E}\left[ g^*(S_t, Z_t, A_t; \tilde{q}) - \tilde{q}(W_t, S_t) \right]^2$$
$$= \mathbb{E}\left[ \{g(S_t, Z_t, A_t) - g^*(S_t, Z_t, A_t; \tilde{q})\} \{g(S_t, Z_t, A_t) + g^*(S_t, Z_t, A_t; \tilde{q}) - 2\tilde{q}(W_t, S_t)\} \right]$$
$$= \mathbb{E}\left[ \{g(S_t, Z_t, A_t) - g^*(S_t, Z_t, A_t; \tilde{q})\} \{g(S_t, Z_t, A_t) - g^*(S_t, Z_t, A_t; \tilde{q}) + 2g^*(S_t, Z_t, A_t; \tilde{q}) - 2\tilde{q}(W_t, S_t)\} \right]$$
$$= \mathbb{E}\left[ \{g(S_t, Z_t, A_t) - g^*(S_t, Z_t, A_t; \tilde{q})\}^2 \right] \tag{39}$$

The last equality is due to the fact that $\mathbb{E}g(S_t, Z_t, A_t)[g^*(S_t, Z_t, A_t; \tilde{q}) - \tilde{q}(W_t, S_t)] = 0$ for any $g \in \mathcal{G}^{(t)}$. From the basic inequality, we have

$$\frac{1}{n} \sum_{i=1}^{n} [\hat{g}(S_{i,t}, Z_{i,t}, A_{i,t}) - \tilde{q}(W_{i,t}, S_{i,t})]^2 \le \frac{1}{n} \sum_{i=1}^{n} [g^*(S_{i,t}, Z_{i,t}, A_{i,t}; \tilde{q}) - \tilde{q}(W_{i,t}, S_{i,t})]^2 + \mu \|g^*\|_{\mathcal{G}^{(t)}}^2 - \mu \|\hat{g}\|_{\mathcal{G}^{(t)}}^2.$$
$$\tag{40}$$

Next, we will establish the different between

$$\mathbb{E}\left[g(S_t, Z_t, A_t) - \tilde{q}(W_t, S_t)\right]^2 - \mathbb{E}\left[g^*(S_t, Z_t, A_t; \tilde{q}) - \tilde{q}(W_t, S_t)\right]^2$$

and

$$\left\{\frac{1}{n}\sum_{i=1}^{n}\left[g(S_{i,t}, Z_{i,t}, A_{i,t}) - \tilde{q}(W_{i,t}, S_{i,t})\right]^2\right\} - \left\{\frac{1}{n}\sum_{i=1}^{n}\left[g^*(S_{i,t}, Z_{i,t}, A_{i,t}; \tilde{q}) - \tilde{q}(W_{i,t}, S_{i,t})\right]^2\right\},$$

to study the bound for $\mathbb{E}\left[\{g(S_t, Z_t, A_t) - g^*(S_t, Z_t, A_t; \tilde{q})\}^2\right]$.

To begin with, for any $g, g^* \in \mathcal{G}^{(t)}$ and $\tilde{q} \in \tilde{\mathcal{Q}}^{(t)}$,

$$\mathrm{Var}\left\{\left[g(S_t, Z_t, A_t) - \tilde{q}(W_t, S_t)\right]^2 - \left[g^*(S_t, Z_t, A_t; \tilde{q}) - \tilde{q}(W_t, S_t)\right]^2\right\}$$

$$\leq \mathbb{E}\left\{\left[g(S_t, Z_t, A_t) - \tilde{q}(W_t, S_t)\right]^2 - \left[g^*(S_t, Z_t, A_t; \tilde{q}) - \tilde{q}(W_t, S_t)\right]^2\right\}^2$$

$$\leq 16\mathbb{E}\left\{g(S_t, Z_t, A_t) - g^*(S_t, Z_t, A_t; \tilde{q})\right\}^2$$

$$= 16\mathbb{E}\left\{\left[g(S_t, Z_t, A_t) - \tilde{q}(W_t, S_t)\right]^2 - \left[g^*(S_t, Z_t, A_t; \tilde{q}) - \tilde{q}(W_t, S_t)\right]^2\right\},$$

where the second inequality is due to the uniform boundness of $g$ and $\tilde{q}$, and the last equality is from equation 39.

Then we apply Corollary of Theorem 3.3 in Bartlett et al. (2005) to the function class $\boldsymbol{\Upsilon}^{(t)}$. For any function $f \in \boldsymbol{\Upsilon}^{(t)}$, $\|f\|_\infty \leq 1$, and $\mathrm{Var}(f) \leq 16\mathbb{E}f$. Take the functional $T$ in Theorem 3.3 of Bartlett et al. (2005) as $T(f) = \mathbb{E}f^2$ and define $r^*$ as the fixed point of a sub-root function $\psi$ such that for any $r \geq r^*$,

$$\psi(r) \geq 16\mathbb{E}\mathcal{R}_n(\boldsymbol{\Upsilon}^{(t)}, T(f) \leq r).$$

Then with probability at least 1- $\delta$, the following inequality holds for any $f \in \boldsymbol{\Upsilon}^{(t)}$,

$$\mathbb{E}f \lesssim 2\frac{1}{n}\sum_{i=1}^{n}f(W_{i,t}, S_{i,t}, Z_{i,t}, A_{i,t}) + r^* + \frac{\log(1/\delta)}{n}.$$

If we take $\tilde{\kappa}_{t,n} = c\sqrt{r^*}$ for some universal constant $c$, and the sub-root function $\psi$ as the identity function. Then $\kappa_n$ is the critical radius of $\mathcal{R}_n(\boldsymbol{\Upsilon}^{(t)})$.

Therefore, for any $g \in \mathcal{G}_M^{(t)}$, $\tilde{q} \in \mathcal{Q}_{\tilde{B}}^{(t)}$, we have

$$\mathbb{E}\left[\{g(S_t, Z_t, A_t) - g^*(S_t, Z_t, A_t; \tilde{q})\}^2\right] \tag{41}$$

$$\lesssim \frac{1}{n}\sum_{i=1}^{n}\left[g(S_{i,t}, Z_{i,t}, A_{i,t}) - \tilde{q}(W_{i,t}, S_{i,t})\right]^2 - \frac{1}{n}\sum_{i=1}^{n}\left[g^*(S_{i,t}, Z_{i,t}, A_{i,t}; \tilde{q}) - \tilde{q}(W_{i,t}, S_{i,t})\right]^2 + \tilde{\kappa}_{t,n}^2 + \frac{\log(1/\delta)}{n} \tag{42}$$

Therefore, for any $g \in \mathcal{G}^{(t)}$, $\tilde{q} \in \mathcal{Q}^{(t)}$, if $\|g\|_{\mathcal{G}^{(t)}}^2 \leq M$ and $\|g\|_{\tilde{\mathcal{Q}}^{(t)}}^2 \leq \tilde{B}$, then equation 41 is still valid. Otherwise, take $z = \|\tilde{q}\|_{\mathcal{Q}^{(t)}}/\min\{\sqrt{\tilde{B}}, \sqrt{M/C_g}\} + \|g\|_{\mathcal{G}^{(t)}}/\sqrt{M}$, we can verify that

$$\|g/z\|_{\mathcal{G}^{(t)}}^2 \leq M$$
$$\|\tilde{q}/z\|_{\tilde{\mathcal{Q}}^{(t)}}^2 \leq \tilde{B}$$
$$\|g^*(\cdot, \cdot, \cdot; \tilde{q}/z)\|_{\mathcal{G}^{(t)}}^2 \leq C_g\|\tilde{q}/z\|_{\tilde{\mathcal{Q}}^{(t)}}^2 \leq M.$$

Then

$$
\mathbb{E}\left[\left\{g(S_t, Z_t, A_t)/z - g^*(S_t, Z_t, A_t; \tilde{q})/z\right\}^2\right]
$$
$$
\lesssim \frac{1}{n}\sum_{i=1}^{n}[g(S_{i,t}, Z_{i,t}, A_{i,t})/z - \tilde{q}(W_{i,t}, S_{i,t})/z]^2 - \frac{1}{n}\sum_{i=1}^{n}[g^*(S_{i,t}, Z_{i,t}, A_{i,t}; \tilde{q})/z - \tilde{q}(W_{i,t}, S_{i,t})/z]^2 + \tilde{\kappa}_{t,n}^2 + \frac{\log(1/\delta)}{n}
$$

$$
\mathbb{E}\left[\left\{g(S_t, Z_t, A_t) - g^*(S_t, Z_t, A_t; \tilde{q})\right\}^2\right]
$$
$$
\lesssim \frac{1}{n}\sum_{i=1}^{n}[g(S_{i,t}, Z_{i,t}, A_{i,t}) - \tilde{q}(W_{i,t}, S_{i,t})]^2 - \frac{1}{n}\sum_{i=1}^{n}[g^*(S_{i,t}, Z_{i,t}, A_{i,t}; \tilde{q}) - \tilde{q}(W_{i,t}, S_{i,t})]^2
$$
$$
+ \max\left\{1, \frac{\|g\|_{\mathcal{G}^{(t)}}^2}{M} + \frac{\|\tilde{q}\|_{\mathcal{Q}^{(t)}}^2}{\min\{\tilde{B}, M/C_g\}}\right\}\left[\kappa_n^2 + \frac{\log(1/\delta)}{n}\right].
$$

hold with probability at least $1 - \delta$.

Then combine with the basic inequality equation 40, with probability at least $1 - \delta$, we have

$$
\|\hat{g}(S_t, Z_t, A_t) - g^*(S_t, Z_t, A_t; \tilde{q})\|_2^2 \lesssim \max\left\{1, \frac{\|\hat{g}\|_{\mathcal{G}^{(t)}}^2}{M} + \frac{\|\tilde{q}\|_{\mathcal{Q}^{(t)}}^2}{\min\{\tilde{B}, M/C_g\}}\right\}\left[r^* + \frac{\log(1/\delta)}{n}\right] + \mu\|g^*\|_{\mathcal{G}^{(t)}}^2 - \mu\|\hat{g}\|_{\mathcal{G}^{(t)}}^2
$$
$$
\lesssim \max\left\{1, \frac{\|\tilde{q}\|_{\mathcal{Q}^{(t)}}^2}{\min\{\tilde{B}, M/C_g\}}\right\}\left[\kappa_n^2 + \frac{\log(1/\delta)}{n}\right] + \mu\|g^*\|_{\mathcal{G}^{(t)}}^2.
$$

The last inequality is from the condition of tuning parameter $\mu$.

$\square$

## I.3 BOUND THE CRITICAL RADIUS

In this section, we characterize the bound of critical radius mentioned above.

**Lemma I.1.** Suppose $\mathcal{G}^{(t)}$, $\mathcal{H}^{(t+1)}$ and $\mathcal{Q}^{(t)}$ are VC-subgraph classed with VC dimensions $\mathbb{V}(\mathcal{G}^{(t)})$, $\mathbb{V}(\mathcal{H}^{(t)})$ and $\mathbb{V}(\mathcal{G}^{(t)})$ respectively, then we have

$$
\tilde{\Delta}_{t,n} \lesssim (T - t + 1)^{1/2}\sqrt{\frac{\max\left\{\mathbb{V}(\mathcal{G}^{(t)}), \mathbb{V}(\mathcal{H}^{(t+1)}), \mathbb{V}(\mathcal{Q}^{(t)})\right\}}{n}} \tag{43}
$$

$$
\tilde{\kappa}_{t,n} \lesssim \sqrt{\frac{\max\left\{\mathbb{V}(\mathcal{G}^{(t)}), \mathbb{V}(\mathcal{Q}^{(t)})\right\}}{n}} \tag{44}
$$

*Proof.* Note that for any $h \in \mathcal{H}^{(t+1)}$, we have $\|h\|_{\mathcal{H}^{(t+1)}}^2 \lesssim C_v(T - t + 1)M_Q$ by Theorem I.1. And equation 43 is derived directly from Section D.3.1 in Miao et al. (2022). As for equation 44, note that

$$
\boldsymbol{\Upsilon}^{(t)} = \{(w_t, s_t, z_t, a_t) \mapsto [g(s_t, z_t, a_t) - g^*(s_t, z_t, a_t; \tilde{q})][g(s_t, z_t, a_t) + g^*(s_t, z_t, a_t; \tilde{q}) - 2\tilde{q}(w_t, s_t)] :
$$
$$
g, g^* \in \mathcal{G}_M^{(t)}, \tilde{q} \in \tilde{\mathcal{Q}}_{\tilde{B}}^{(t)}\}.
$$

By the similar argument in bounding $\log N_n(t, \Omega^{(t)})$ in Section D.4.2 in Miao et al. (2022), we have

$$
\log N_n(t, \boldsymbol{\Upsilon}^{(t)}) \lesssim \log N_n(t, \mathcal{G}_M^{(t)}) + \log N_n(t, \tilde{\mathcal{Q}}_{\tilde{B}}^{(t)})
$$
$$
\lesssim \log N_n(t, \mathcal{G}_M^{(t)}) + \log N_n(t, \mathcal{Q}_B^{(t)}),
$$

where $N_n(\epsilon, \mathcal{G})$ denotes the smallest empirical $\epsilon$-covering of $\mathcal{G}$. And the bound in equation 44 is obtained by bounding the local Rademacher complexity by entropy integral (See Section D.3.1 in Miao et al. (2022)). $\square$

Similar results apply to $\tilde{\Delta}_n$ and $\tilde{\kappa}_n$ and we get

**Lemma I.2.** Suppose $\mathcal{G}$ and $\mathcal{Q}$ are VC-subgraph classed with VC dimensions $\mathbb{V}(\mathcal{G})$ and $\mathbb{V}(\mathcal{G})$ respectively, then we have

$$\tilde{\Delta}_n + \tilde{\kappa}_n \lesssim \sqrt{\frac{\max\{\mathbb{V}(\mathcal{G}), \mathbb{V}(\mathcal{Q})\}}{n}}. \tag{45}$$

**Lemma I.3.** Suppose $\mathcal{G}^{(t)}$, $\mathcal{Q}^{(t)}$ and $\mathcal{H}^{(t+1)}$ are RKHSs endowed with reproducing kernel $K_{\mathcal{G}}$, $K_{\mathcal{Q}}$ and $K_{\mathcal{G}}$ with decreasing sorted eigenvalues $\{\lambda_j(K_{\mathcal{G}})\}_{j=1}^{\infty}$, $\{\lambda_j(K_{\mathcal{Q}})\}_{j=1}^{\infty}$ and $\{\lambda_j(K_{\mathcal{H}})\}_{j=1}^{\infty}$, respectively.
Then $\tilde{\Delta}_{t,n}$ is upper bounded by $\delta$ satisfies

$$\sqrt{\frac{1}{n}}\sqrt{\sum_{i,j=1}^{\infty} \min\{\lambda_i(K_{\mathcal{G}})\lambda_j(K_{\mathcal{Q}}), \delta^2\}} \lesssim \delta^2$$

$$\sqrt{\frac{(T-t+1)}{n}}\sqrt{\sum_{i,j=1}^{\infty} \min\{[\lambda_i(K_{\mathcal{G}}) + \lambda_i(K_{\mathcal{H}})]\lambda_j(K_{\mathcal{Q}}), \delta^2\}} \lesssim \delta^2$$

Then $\tilde{\kappa}_{t,n}$ is upper bounded by $\delta$ satisfies

$$\sqrt{\frac{(T-t+1)}{n}}\sqrt{\sum_{i,j=1}^{\infty} \min\{[\lambda_i(K_{\mathcal{G}}) + \lambda_i(K_{\mathcal{Q}})]\lambda_j(K_{\mathcal{Q}}), \delta^2\}} \lesssim \delta^2.$$

*Proof.* The proof follows the similar argument in the proof of Lemma D.7 in Miao et al. (2022). (See Section D.4.3 in Miao et al. (2022).) □

With different decay rates of eigenvalues, by directly applying Lemma I.3, we obtain the following corollary.

**Corollary I.1.** With the same conditions in Lemma I.3, if $\lambda_j(K_{\mathcal{Q}}) \propto j^{-2\alpha_{\mathcal{Q}}}$, $\lambda_j(K_{\mathcal{G}}) \propto j^{-2\alpha_{\mathcal{G}}}$, $\lambda_j(K_{\mathcal{H}}) \propto j^{-2\alpha_{\mathcal{H}}}$, where $\alpha_{\mathcal{G}}, \alpha_{\mathcal{H}}, \alpha_{\mathcal{Q}} > 1/2$, then we have

$$\tilde{\Delta}_{t,n} \lesssim \sqrt{(T-t+1)}n^{\frac{1}{2+\max\{1/\alpha_{\mathcal{Q}}, 1/\alpha_{\mathcal{G}}, 1/\alpha_{\mathcal{H}}\}}}\log n,$$

$$\tilde{\kappa}_{t,n} \lesssim n^{\frac{1}{2+\max\{1/\alpha_{\mathcal{Q}}, 1/\alpha_{\mathcal{G}}\}}}\log n.$$

Similar results apply to $\tilde{\Delta}_n$ and $\tilde{\kappa}_n$.

**Corollary I.2.** Suppose $\mathcal{G}$, $\mathcal{Q}$ are RKHSs endowed with reproducing kernel $K_{\mathcal{G}}$, $K_{\mathcal{Q}}$ and $K_{\mathcal{G}}$ with decreasing sorted eigenvalues $\{\lambda_j(K_{\mathcal{G}})\}_{j=1}^{\infty}$, $\{\lambda_j(K_{\mathcal{Q}})\}_{j=1}^{\infty}$ respectively.
Then if $\lambda_j(K_{\mathcal{Q}}) \propto j^{-2\alpha_{\mathcal{Q}}}$, $\lambda_j(K_{\mathcal{G}}) \propto j^{-2\alpha_{\mathcal{G}}}$, we have

$$\tilde{\Delta}_n + \tilde{\kappa}_n \lesssim n^{\frac{1}{2+\max\{1/\alpha_{\mathcal{Q}}, 1/\alpha_{\mathcal{G}}\}}}\log n.$$

