# OpenReview forum: "Blessing from Experts: Super Reinforcement Learning in Confounded Environments"
_ICLR.cc/2023/Conference — Submitted to ICLR 2023_

### Official Review · Reviewer_SFSz · 2022-10-21

**Confidence:** 2
**Correctness:** 3
**Technical Novelty And Significance:** 3
**Empirical Novelty And Significance:** 2
**Recommendation:** 6

**Clarity, Quality, Novelty And Reproducibility:**

Overall the paper is well-written and easy to follow. However, I would appreciate a proper paper ending with a conclusion or discussion section. In its current form, the paper seems to end abruptly on page 9.

The paper is well-motivated by real-world applications, and it proposed a relatively comprehensive theoretical analysis of the problem of finding an optimal policy in the presence of the confounder. The practical implementable algorithms are helpful in evaluating them in real-world environments (even in simulation). However, I am rather surprised that no such empirical evaluation is presented in the paper. The use of expert recommendations in decision-making is important and should improve existing policy learning. However, the practicality would depend on the availability of the expert recommendation.

My understanding is that the algorithm takes input the expert-recommended action at every timestep and then makes a decision based on both history and expert demonstration. However, I am a bit confused about the offline data generation process (history data). What are the assumptions on offline data generation? How is the offline data collected? Is it from a human agent or an optimally learned policy that uses confounded observation data? That raises the question of how much expert-recommended actions are needed to learn an optimal policy in the presence of a confounder.

Moreover, having an expert action at each timestep might be infeasible in many RL tasks. For example, in learning robotic locomotion tasks (e.g., humanoid walk, quadruped run), access to expert recommendations would be infeasible. It is unclear how the proposed method can be leveraged in those settings. Are the confounded environments restricted where such demonstrations are feasible only? Clarifying these points in the abstract and intro would be helpful for the reader to understand. One suggestion would be to specify expert recommendations in the title. Alternatively, a justification would be helpful for how the proposed method is generalized for all the confounded environments, including environments where direct access to expert action at each timestep is not feasible (robotic locomotion).



**Strength And Weaknesses:**

Strength:

Important problem finding an optimal policy in confounded offline reinforcement learning.
Overall well-written, and the assumptions are stated in the context of the theorem.
The practical derivation of the proposed algorithms helps leverage them in a real-world task.

Weakness:

Real-world evaluation of the proposed practical algorithms is missing.
While assumptions are stated, it is unclear if all those are held in a real-world setup.
Missing empirical evaluation and discussion.



**Summary Of The Paper:**

This paper analyzes offline policy learning in a confounded setting for contextual bandits and sequential decision-making. The paper introduces the notion of super policy learning in the setup where expert-recommended actions are available at each step in addition to the past historical data. Under this setup, the author(s) propose two algorithms for the contextual bandit and sequential decision-making (RL) settings and derive corresponding regret bounds.


**Summary Of The Review:**

The paper tackles the important problem of finding an optimal policy in confounded offline reinforcement learning. The proposed practical algorithms have the potential to be implemented for real-world tasks. However, the justification of the assumptions on the real-world task is missing. Adding such empirical evaluation would eventually improve the justification of the paper's claims.

---

> ### Author Response · Authors · 2022-11-18
> **Response (1/2)**
>
> We thank the reviewer for suggestive feedback and we have updated our paper accordingly (marked in blue). Our one-to-one response to the reviewer’s questions and concerns is given below.
>
> - **Empirical Evaluation**. Our primary goal in this paper is to provide a theoretical framework for the super reinforcement learning by leveraging the recommended action as an input for decision making under unmeasured confounding. Since our proposal is substantially different from all existing approaches, it requires a lot of efforts to develop new algorithms and theories. Therefore, we did not perform numerical studies in our original submission.
> Nonetheless, following your suggestion, we have conducted extensive numerical experiments (i.e., **three simulation studies and two real-world applications**) in this revision to illustrate the superior performance of our proposed super policy learning. In particular:
>   - First, we conduct two simulation studies in contextual bandits and one simulation study in the sequential decision making, respectively, to compare the proposed super-policy against estimated policies computed via existing RL algorithms. Details of simulation settings and results can be found in Section C of the supplementary material. The super-policy produces significantly smaller regret values when the observed action contains information about the unobserved state variables.
>   - Second, we apply our methods to the right heart catheterization (RHC) dataset from the Study to Understand Prognoses and Preferences for Outcomes and Risks for Treatments (SUPPORT). The same dataset is evaluated by Qi et al. (2021), Kallus et al. (2021) and Cui et al. (2020) under the framework of proximal causal inference. A more detailed description is presented in Section D.1 in the supplementary material. Our proposed super-policy produces higher evaluated values compared to other estimated policies via existing RL algorithms.
>   - Third, we apply our methods to the MIMIC-III dataset (Raghu et al., 2017) to study the optimal treatment strategy for sepsis. This dataset serves as an example of the setting of confounded sequential decision making. More details can be found in Section D.2 in the supplementary material. Again our proposed super-policy shows appealing performance.
>
> - **Clarification on the “ expert’s recommendation”**. Sorry for the confusion about the “expert’s recommendation”. We mainly refer the behavior policy as the “expert’s recommendation”. Indeed, the observed action in our offline data does not need to be generated by the expert (or a policy close to the optimal one). As long as the behavior policy depends on the hidden information $U$, the super-optimality is guaranteed. For example, as seen in Table 1, when $\epsilon = 1$, the performance of the behavior policy is very bad but our proposed sup-policy can still achieve the global optimality. Since the requirement on the behavior policy is very mild, our method is indeed practically feasible. Refer to our next response for details. Therefore the only assumption on the offline data generation is that the behavior policy depends on the unobserved state $U$.
>
> ### Reference
>
> Qi, Z., R. Miao, and X. Zhang (2021). Proximal learning for individualized treatment regimes under unmeasured confounding. arXiv preprint arXiv:2105.01187.
>
> Kallus, N., X. Mao, and M. Uehara (2021). Causal inference under unmeasured confounding with negative controls: A minimax learning approach. arXiv preprint arXiv:2103.14029.
>
> Cui, Y., H. Pu, X. Shi, W. Miao, and E. T. Tchetgen (2020). Semiparametric proximal causal inference. arXiv preprint arXiv:2011.08411.
>
> Raghu, A., M. Komorowski, I. Ahmed, L. Celi, P. Szolovits, and M. Ghassemi (2017). Deep reinforcement learning for sepsis treatment. arXiv preprint arXiv:1711.09602.

---

> > ### Author Response · Authors · 2022-11-18
> > **Response (2/2)**
> >
> > - **Motivation and availability of the expert’s recommendation**. As you have commented, our methods require human experts to continue to recommend actions in the future. This assumption is plausible in some applications and we consider two examples below to elaborate. Meanwhile, our method can be generalized to the setting where the recommended action is not accessible at each decision point. For example, if we know when human agents will provide a recommended action in the testing environment on a regular basis, we are able to adapt the current method to learn a super-policy that leverages such recommendations at those decision points. Overall, our approach can be regarded as an example of Human-in-the-loop.
> >
> >    - **Urgent care**. As discussed in the main text, clinicians leverage visual observations or communications with patients to recommend treatments, where such unstructured information is hard to quantify and often not recorded (McDonald, 1996), leading to a confounded dataset. Suppose clinicians can continue to recommend treatments in the future which is commonly seen in practice, our proposal can be naturally applied to these examples for super-policy learning.
> >
> >       Following your suggestion, we consider a concrete example of managing sepsis patients and use the MIMIC3 dataset (Nanayakkara et al., 2022) to empirically investigate the performance of the proposed method. This dataset is likely to contain unmeasured confounders, as discussed in (Namkoong et al., 2020). To infer the latent confounders, we use the weight and temperature at the current time step as the reward proxy $W_t$, since it is reasonable to assume that they are not directly related to the action. We take these two variables at the previous time step as the action proxy $Z_t$ since it is reasonable to assume that they are not directly related to the reward at the current time step.
> >   - **Deep brain stimulation**. Due to recent advances in deep brain stimulation technology, it becomes feasible to instantly collect electroencephalogram data, based on which we are able to provide adaptive stimulation to specific regions in the brain so as to treat patients with neurological disorders including Parkinson’s disease, essential tremor, dystonia, etc (see e.g., Lozano et al., 2019). In these applications, the patient herself is allowed to determine the behavior policy (e.g., when to turn on/off the stimulation, for how long, how much amplitude, etc) based on information **only known to herself** (e.g., how she feels at the moment), therefore generating an offline dataset with unmeasured confounders. Our proposed super-policy can be naturally applied when the patient continues to recommend actions in the future.
> >
> >   We have included the related discussions in the introduction section (the second paragraph) and Section 4.1 (the third paragraph) of the revised paper.
> >
> >
> >
> > ### Reference
> >
> > McDonald, C. J. (1996). Medical heuristics: the silent adjudicators of clinical practice.
> >
> > Nanayakkara, T., G. Clermont, C. J. Langmead, and D. Swigon (2022). Unifying cardiovascular modeling with deep reinforcement learning for uncertainty aware control of sepsis treatment. PLOS Digital Health 1(2), e0000012.
> >
> > Namkoong, H., R. Keramati, S. Yadlowsky, and E. Brunskill (2020). Off-policy policy evaluation for sequential decisions under unobserved confounding. Advances in Neural Information Processing Systems 33, 18819–18831.
> >
> > Lozano, A. M., N. Lipsman, H. Bergman, P. Brown, S. Chabardes, J. W. Chang, K. Matthews, C. C. McIntyre, T. E. Schlaepfer, M. Schulder, et al. (2019). Deep brain stimulation: current challenges and future directions. Nature Reviews Neurology 15(3), 148–160.

---

> > > ### Author Response · Authors · 2022-11-22
> > > **Looking forward to your feedback**
> > >
> > > Dear Reviewer SFSz,
> > >
> > > Thank you for your invaluable feedback. We were wondering whether our response and the revised manuscript addressed your concerns. If you have any additional comments, please let us know, we would be happy to address them. We kindly ask you to consider raising your scores if the concerns were appropriately addressed.

---

### Official Review · Reviewer_FcTZ · 2022-10-27

**Confidence:** 3
**Clarity, Quality, Novelty And Reproducibility:** See above.
**Correctness:** 4
**Technical Novelty And Significance:** 3
**Empirical Novelty And Significance:** Not applicable
**Recommendation:** 5

**Strength And Weaknesses:**

Strengths: this paper has a clear problem statement and exposition. I have no checked the proofs in detail, but they seem to be correct. There may be situations where the proposed scheme is useful.

Weaknesses: the motivation is not totally clear to me. Particularly, the proxies for the unobserved confounder are introduced as an assumption to get around the non-identifiability of the target, expert-recommendation enhanced, policy. How realistic is this assumption? Are there particular motivating problems in mind? It would also be nice to have some experiments showing the scope of the benefits.

**Summary Of The Paper:**

This paper considers offline reinforcement learning in the presence of unobserved confounding. There are two observations. First, including the "expert recommended" action (from the behavioral policy) as part of the state is helpful, because this action was informed by the unobserved confounder. Second, if there are proxies for the unobserved confounder available, these can be leveraged to learn the optimal policy under the (better) expert-recommendations available setting. This later result is a direct application of recent proxy identification methods in causal inference.

------------

I read the rebuttal comments and the revised paper. I think the point about past/future state serving as proxies is a good one (though I haven't given much thought to its correctness). The authors have also added some experimental demonstration. Unfortunately, these are substantial revisions and I don't have the capacity to do more than a cursory evaluation of them.

Overall, I remain weakly positive on this paper. It's not earth shattering, but it seems to be a solid execution of a reasonably interesting idea.  I've updated my score to an 8, but with the intention that this should be parsed literally ("I think it's a good paper worth accepting") and not as a very strong endorsement.

(FWIW though, I find the brain stimulation example pretty unconvincing---if patient actions are available at runtime, then we'd presumably just ignore whatever the learned policy is and defer to the patient)

-----------
A second update:

Following the reviewer discussion, I'm downgrading my score to a weak reject. The crux of this is:
1. we all agree that proxy identification results from causal inference translate to results for policy learning for unobserved confounding. This paper seems to do the translation correctly, and I'm willing to accept that it's likely technically correct.*
2. however, a main claim of the paper is that expert recommended actions can serve / are especially good choices for the required proxies. None of us were able to articulate clearly how this part of the argument goes. (AFAICT, the real world healthcare examples don't use expert recommendations in this role)
3. thus, it seems that there's at least a clarity problem, and potentially a technical correctness problem. Based on the discussion, this is severe enough that the paper needs a major revision. If no reviewer is able to explain the gist of the main idea, then the paper is not communicating its development clearly enough.

*Although, here, I wonder if the presentation could be improved substantially simply by taking the building block as the identification result rather than the tool used for identification. That is, explain directly how to translate from such-and-such is causally identified, so the policy learning can now be achieved as such-and-such. The technical details of how proxy identification results work don't seem obviously different than how it works for identification, but they're involved enough to be distracting



**Summary Of The Review:**

This paper is clear and (apparently) correct. While I am not fully convinced of its importance, I think it's a nice demo of the use of proxy methods in reinforcement learning, and may have some real-world applicability.

---

> ### Author Response · Authors · 2022-11-18
> **Response (1/2)**
>
> We thank the reviewer for suggestive feedback and we have updated our paper accordingly. Our one-to-one response to the reviewer's questions and concerns is given below.
> - **Motivations and real-world examples**.  Proxy variables exist in a large variety of applications. In particular, as shown in Tennenholtz et al. (2020), past and future observations can be served as the two proxies in confounded partially observable Markov
> decision processes (POMDPs). See also Nair and Jiang (2021); Shi et al. (2021). More specifically, as discussed in Section 4.3 and A.2 of the supplementary material, $Z_t$ could be set to the past history up to $t − 1$ and $W_t$ could be set to the current observation. In addition, the technical assumptions we require for super-policy identification are very similar to those imposed in these papers. As such, our method can be applied to most confounded sequential decision-making problems where human experts will continue to recommend actions in the future. We consider two examples to elaborate.
>    - **Urgent care**. As discussed in the main text, clinicians leverage visual observations or communications with patients to recommend treatments, where such unstructured information is hard to quantify and often not recorded (McDonald, 1996), leading to a
> confounded dataset. Suppose clinicians can continue to recommend treatments in the future which is commonly seen in practice, our proposal can be naturally applied to these examples for super-policy learning.
>
>        Following your suggestion, we consider a concrete example of managing sepsis patients and use the MIMIC3 dataset (Nanayakkara et al., 2022) to empirically investigate the performance of the proposed method. This dataset is likely to contain unmeasured confounders, as discussed in (Namkoong et al., 2020). To infer the latent confounders, we use the weight and temperature at the current time step as the reward proxy $W_t$, since it is reasonable to assume that they are not directly related to the action. We take these two variables at the previous time step as the action proxy $Z_t$ since it is reasonable to assume that they are not directly related to the reward at the current time step.
>   - **Deep brain stimulation**. Due to recent advances in deep brain stimulation technology, it becomes feasible to instantly collect electroencephalogram data, based on which we are able to provide adaptive stimulation to specific regions in the brain so as to treat patients with neurological disorders including Parkinson’s disease, essential tremor, dystonia, etc (see e.g., Lozano et al., 2019). In these applications, the patient herself is allowed to determine the behavior policy (e.g., when to turn on/off the stimulation, for how long, how much amplitude, etc) based on information **only known to herself** (e.g., how she feels at the moment), therefore generating an offline dataset with unmeasured confounders. Our proposed super-policy can be naturally applied when the patient continues to recommend actions in the future.
>
>   We have included the related discussions in the introduction section (the second paragraph) and Section 4.1 (the third paragraph) of the revised paper.
>
>
>
> ### Reference
> Tennenholtz, G., U. Shalit, and S. Mannor (2020). Off-policy evaluation in partially observable environments. In Proceedings of the AAAI Conference on Artificial Intelligence, Volume 34, pp. 10276–10283.
>
> Nair, Y. and N. Jiang (2021). A spectral approach to off-policy evaluation for POMDPs. arXiv preprint arXiv:2109.10502.
>
> Shi, C., M. Uehara, and N. Jiang (2021). A minimax learning approach to off-policy evaluation in partially observable Markov decision processes. arXiv preprint arXiv:2111.06784.
>
> McDonald, C. J. (1996). Medical heuristics: the silent adjudicators of clinical practice.
>
> Nanayakkara, T., G. Clermont, C. J. Langmead, and D. Swigon (2022). Unifying cardiovascular modeling with deep reinforcement learning for uncertainty aware control of sepsis treatment. PLOS Digital Health 1(2), e0000012.
>
> Namkoong, H., R. Keramati, S. Yadlowsky, and E. Brunskill (2020). Off-policy policy evaluation for sequential decisions under unobserved confounding. Advances in Neural Information Processing Systems 33, 18819–18831.
>
> Lozano, A. M., N. Lipsman, H. Bergman, P. Brown, S. Chabardes, J. W. Chang, K. Matthews, C. C. McIntyre, T. E. Schlaepfer, M. Schulder, et al. (2019). Deep brain stimulation: current challenges and future directions. Nature Reviews Neurology 15(3), 148–160.

---

> > ### Author Response · Authors · 2022-11-18
> > **Response (2/2)**
> >
> > - **Empirical Evaluation**. Our primary goal in this paper is to provide a theoretical framework for the super reinforcement learning by leveraging the recommended action as an input for decision making under unmeasured confounding. Since our proposal is substantially different from all existing approaches, it requires a lot of efforts to develop new algorithms and theories. Therefore, we did not perform numerical studies in our original submission.
> > Nonetheless, following your suggestion, we have conducted extensive numerical experiments (i.e., **three simulation studies and two real-world applications**) in this revision to illustrate the superior performance of our proposed super policy learning. In particular:
> >   - First, we conduct two simulation studies in contextual bandits and one simulation study in the sequential decision making, respectively, to compare the proposed super-policy against estimated policies computed via existing RL algorithms. Details of simulation settings and results can be found in Section C of the supplementary material. The super-policy produces significantly smaller regret values when the observed action contains information about the unobserved state variables.
> >   - Second, we apply our methods to the right heart catheterization (RHC) dataset from the Study to Understand Prognoses and Preferences for Outcomes and Risks for Treatments (SUPPORT). The same dataset is evaluated by Qi et al. (2021), Kallus et al. (2021) and Cui et al. (2020) under the framework of proximal causal inference. A more detailed description is presented in Section D.1 in the supplementary material. Our proposed super-policy produces higher evaluated values compared to other estimated policies via existing RL algorithms.
> >   - Third, we apply our methods to the MIMIC-III dataset (Raghu et al., 2017) to study the optimal treatment strategy for sepsis. This dataset serves as an example of the setting of confounded sequential decision making. More details can be found in Section D.2 in the supplementary material. Again our proposed super-policy shows appealing performance.
> >
> >
> > ### Reference
> >
> > Qi, Z., R. Miao, and X. Zhang (2021). Proximal learning for individualized treatment regimes under unmeasured confounding. arXiv preprint arXiv:2105.01187.
> >
> > Kallus, N., X. Mao, and M. Uehara (2021). Causal inference under unmeasured confounding with negative controls: A minimax learning approach. arXiv preprint arXiv:2103.14029.
> >
> > Cui, Y., H. Pu, X. Shi, W. Miao, and E. T. Tchetgen (2020). Semiparametric proximal causal inference. arXiv preprint arXiv:2011.08411.
> >
> > Raghu, A., M. Komorowski, I. Ahmed, L. Celi, P. Szolovits, and M. Ghassemi (2017). Deep reinforcement learning for sepsis treatment. arXiv preprint arXiv:1711.09602.

---

> > > ### Author Response · Authors · 2022-11-22
> > > **Looking forward to your feedback**
> > >
> > > Dear Reviewer FcTZ,
> > >
> > > Thank you for your invaluable feedback. We were wondering whether our response and the revised manuscript addressed your concerns. If you have any additional comments, please let us know, we would be happy to address them. We kindly ask you to consider raising your scores if the concerns were appropriately addressed.

---

### Official Review · Reviewer_ye4h · 2022-10-27

**Confidence:** 4
**Correctness:** 3
**Technical Novelty And Significance:** 3
**Empirical Novelty And Significance:** Not applicable
**Recommendation:** 3

**Clarity, Quality, Novelty And Reproducibility:**

Clarity: The paper is hard to follow at times. Although the motivating example was interesting, many assumptions are introduced without proper context and should be motivated with examples.

Quality: The main drawback of the paper is that the methods are not applied on any real-world dataset. It is also not clear how much of the identification results depend on prior work on proximal causal learning, and how much of it is new. Finally, the last section on finite sample guarantees makes several strong assumptions in order to obtain meaningful bounds.

**Strength And Weaknesses:**

Strengths:
- I think the application of proximal causal learning in confounded RL setting is quite interesting.
- As far as I can tell, the identification results look complete if one assumes the existence of proxy variables with desired properties.

Weaknesses:
- The paper is quite hard to follow at times. For example, the authors do not motivate the existence of proxy variables with desired properties. There is also no real-world example regarding the setting considered in section 4 (confounded RL with proxy variables for rewards and actions).
- Perhaps the major drawback of the paper is that the proposed method is not validated on any real-world dataset. So it is hard to tell where such methods can be applied.
- The final section on finite sample guarantees requires many assumptions. For example, it requires a bound on $p_\max$ which is completely a property of the behavioral policy. Second, for RL setting it requires memoryless confounding. I am not sure if confounding is memoryless one still requires such complicated method for estimation. Wouldn't it be possible to just perform posterior inference on hidden variables from the observation?
- The authors also do not discuss how to solve the sequence of linear integral equation. This problem should be hard in general and it should have been discussed. Moreover, for finite sample guarantees, one needs to solve this set of equations from finite samples and needs to argue about estimation error. However, as far as I can tell, no such error bound (in terms of samples n) was provided.



Questions for the authors:
1. Why do you need two different types of proxy variables? Can you obtain similar results if there is a single proxy variable depending on the unobserved hidden state?
2. I am not sure if the assumptions of multiple proxy variables and bridge functions with appropriate conditions are realistic. Can the authors provide some real-world examples where such multiple proxies are available and bridge functions satisfy the desired properties. I understand that it might not be feasible to verify some of the properties in practice, but does there exist any real-world datasets which suggest existence of such $Q$-bridge functions at every time-step?
3. Why does the finite sample guarantee depend on the constant $p_\max$? In general, one should expect a constant related to the overlap between the behavioral policy $\pi^b$ and optimal policy to show up in the bound but the constant $p_\max$ as well as $p^\omega_{t,\max}$ just depend on behavioral policy and can be arbitrarily large.
4. The role of the class $\mathcal{G}$ is not clear in theorem 5.1. Since this is just obtained from projecting the class of $q$-functions, why isn't bounding the complexity of bridge functions sufficient?

**Summary Of The Paper:**

This paper considers the problem of reinforcement learning in the presence of unmeasured confounders. In particular, the authors consider a setting where the data is collected according to a behavioral policy $\pi^b: S \times U \rightarrow \Delta(A)$ but the variables in $U$ are unobserved. The goal is to learn a "super" policy which might be history dependent and has return higher than the optimal policy that uses only the observed states in S.

In order to solve the super policy learning problem, the authors use techniques from proximal causal learning. In particular, they assume that there are proxies for both actions and rewards at each time step. If the proxies satisfy certain conditional independence assumption the expected return from a super policy can be expressed through a set of $Q$-bridge functions. These $Q$-bridge functions only depend on observable variables (states in S, and proxies W, Z) and they are estimated as the solution of a set of linear integral equations.

The authors finally provide sample complexity guarantees for learning optimal policies under confounded contextual bandits and confounded RL setting, but with memoryless unmeasured confounding. The bounds are derived assuming bounded complexity for the class of $Q$-bridge functions and their projections.

**Summary Of The Review:**

I am leaning towards rejecting the paper mainly because the presentation of the paper is not clear, and the author makes many assumptions without giving proper example. The methods are also not validated on any real-world datasets, and not clear how they compare with prior methods.

---

> ### Author Response · Authors · 2022-11-18
> **Response (1/3)**
>
> We thank the reviewer for suggestive feedback and we have updated our paper accordingly (marked in blue). Our one-to-one response to the reviewer’s questions and concerns is given below.
>
> - **Our contributions**. Before addressing your comments, we would like to clarify the major contribution of our paper. Specifically, we innovatively propose a framework for super reinforcement learning by including the recommended actions in the decision rule for enhanced policy learning (i.e., **treating the recommended action as a state variable when making decisions**). In the confounded environments, the recommended actions could contain the information of latent confounders, and thus can be leveraged to improve policy learning. Based on this idea, we extend proximal causal inference methods to our setting for super-policy learning and establish various identification results.
>
> - **Empirical Evaluation**. Our primary goal in this paper is to provide a theoretical framework for the super reinforcement learning by leveraging the recommended action as an input for decision making under unmeasured confounding. Since our proposal is substantially different from all existing approaches, it requires a lot of efforts to develop new algorithms and theories. Therefore, we did not perform numerical studies in our original submission.
> Nonetheless, following your suggestion, we have conducted extensive numerical experiments (i.e., **three simulation studies and two real-world applications**) in this revision to illustrate the superior performance of our proposed super policy learning. In particular:
>   - First, we conduct two simulation studies in contextual bandits and one simulation study in the sequential decision making, respectively, to compare the proposed super-policy against estimated policies computed via existing RL algorithms. Details of simulation settings and results can be found in Section C of the supplementary material. The super-policy produces significantly smaller regret values when the observed action contains information about the unobserved state variables.
>   - Second, we apply our methods to the right heart catheterization (RHC) dataset from the Study to Understand Prognoses and Preferences for Outcomes and Risks for Treatments (SUPPORT). The same dataset is evaluated by Qi et al. (2021), Kallus et al. (2021) and Cui et al. (2020) under the framework of proximal causal inference. A more detailed description is presented in Section D.1 in the supplementary material. Our proposed super-policy produces higher evaluated values compared to other estimated policies via existing RL algorithms.
>   - Third, we apply our methods to the MIMIC-III dataset (Raghu et al., 2017) to study the optimal treatment strategy for sepsis. This dataset serves as an example of the setting of confounded sequential decision making. More details can be found in Section D.2 in the supplementary material. Again our proposed super-policy shows appealing performance.
>
> - **Solving linear integral equations**. In this revision, we have provided the details of solving the sequence of linear integral equations (equation (35)) in Section G.1 of the supplementary material due to the space limit. In particular, we introduced the min-max conditional moment estimation procedure for learning the Q-bridge functions. Theorems 5.1 and 5.2 are based on such min-max conditional moment estimations. In Section G of the supplementary material, we have also characterized the convergence rate of the resulting error in terms of the sample size n.
>
> - **Two different types of proxy variables**. There are two sources of unmeasured confounding in the confounded POMDP model. The first one is the effect of $U_t$ on the rewards and future observations. The second one is the effect on the future latent variables $U_{t+1}$. Roughly speaking, $W_{t+1}$ can be regarded as a surrogate for adjusting the effect on $U_{t+1}$ whereas $Z_{t+1}$ is used to adjust the effect on the rewards and future observations. As such, the existence of one type of proxy might not be sufficient for policy value identification in general POMDPs without the memoryless unmeasured confounding assumption.
>
>
> ### Reference
>
> Qi, Z., R. Miao, and X. Zhang (2021). Proximal learning for individualized treatment regimes under unmeasured confounding. arXiv preprint arXiv:2105.01187.
>
> Kallus, N., X. Mao, and M. Uehara (2021). Causal inference under unmeasured confounding with negative controls: A minimax learning approach. arXiv preprint arXiv:2103.14029.
>
> Cui, Y., H. Pu, X. Shi, W. Miao, and E. T. Tchetgen (2020). Semiparametric proximal causal inference. arXiv preprint arXiv:2011.08411.
>
> Raghu, A., M. Komorowski, I. Ahmed, L. Celi, P. Szolovits, and M. Ghassemi (2017). Deep reinforcement learning for sepsis treatment. arXiv preprint arXiv:1711.09602.

---

> > ### Author Response · Authors · 2022-11-18
> > **Response (2/3)**
> >
> > - **Motivations and real-world examples**.  Proxy variables exist in a large variety of applications. In particular, as shown in Tennenholtz et al. (2020), past and future observations can be served as the two proxies in confounded partially observable Markov
> > decision processes (POMDPs). See also Nair and Jiang (2021); Shi et al. (2021). More specifically, as discussed in Section 4.3 and A.2 of the supplementary material, $Z_t$ could be set to the past history up to t − 1 and $W_t$ could be set to the current observation. In addition, the technical assumptions we require for super-policy identification are very similar to those imposed in these papers. As such, our method can be applied to most confounded sequential decision-making problems where human experts will continue to recommend actions in the future. We consider two examples to elaborate.
> >    - **Urgent care**. As discussed in the main text, clinicians leverage visual observations or communications with patients to recommend treatments, where such unstructured information is hard to quantify and often not recorded (McDonald, 1996), leading to a
> > confounded dataset. Suppose clinicians can continue to recommend treatments in the future which is commonly seen in practice, our proposal can be naturally applied to these examples for super-policy learning.
> >
> >        Following your suggestion, we consider a concrete example of managing sepsis patients and use the MIMIC3 dataset (Nanayakkara et al., 2022) to empirically investigate the performance of the proposed method. This dataset is likely to contain unmeasured confounders, as discussed in (Namkoong et al., 2020). To infer the latent confounders, we use the weight and temperature at the current time step as the reward proxy $W_t$, since it is reasonable to assume that they are not directly related to the action. We take these two variables at the previous time step as the action proxy $Z_t$ since it is reasonable to assume that they are not directly related to the reward at the current time step.
> >   - **Deep brain stimulation**. Due to recent advances in deep brain stimulation technology, it becomes feasible to instantly collect electroencephalogram data, based on which we are able to provide adaptive stimulation to specific regions in the brain so as to treat patients with neurological disorders including Parkinson’s disease, essential tremor, dystonia, etc (see e.g., Lozano et al., 2019). In these applications, the patient herself is allowed to determine the behavior policy (e.g., when to turn on/off the stimulation, for how long, how much amplitude, etc) based on information **only known to herself** (e.g., how she feels at the moment), therefore generating an offline dataset with unmeasured confounders. Our proposed super-policy can be naturally applied when the patient continues to recommend actions in the future.
> >
> >   We have included the related discussions in the introduction section (the second paragraph) and Section 4.1 (the third paragraph) of the revised paper.
> >
> >
> > ### Reference
> > Tennenholtz, G., U. Shalit, and S. Mannor (2020). Off-policy evaluation in partially observable environments. In Proceedings of the AAAI Conference on Artificial Intelligence, Volume 34, pp. 10276–10283.
> >
> > Nair, Y. and N. Jiang (2021). A spectral approach to off-policy evaluation for POMDPs. arXiv preprint arXiv:2109.10502.
> >
> > Shi, C., M. Uehara, and N. Jiang (2021). A minimax learning approach to off-policy evaluation in partially observable Markov decision processes. arXiv preprint arXiv:2111.06784.
> >
> > McDonald, C. J. (1996). Medical heuristics: the silent adjudicators of clinical practice.
> >
> > Nanayakkara, T., G. Clermont, C. J. Langmead, and D. Swigon (2022). Unifying cardiovascular modeling with deep reinforcement learning for uncertainty aware control of sepsis treatment. PLOS Digital Health 1(2), e0000012.
> >
> > Namkoong, H., R. Keramati, S. Yadlowsky, and E. Brunskill (2020). Off-policy policy evaluation for sequential decisions under unobserved confounding. Advances in Neural Information Processing Systems 33, 18819–18831.
> >
> > Lozano, A. M., N. Lipsman, H. Bergman, P. Brown, S. Chabardes, J. W. Chang, K. Matthews, C. C. McIntyre, T. E. Schlaepfer, M. Schulder, et al. (2019). Deep brain stimulation: current challenges and future directions. Nature Reviews Neurology 15(3), 148–160.

---

> > > ### Author Response · Authors · 2022-11-18
> > > **Response (3/3)**
> > >
> > > - **Technical assumptions**. We agree that the requirement on $p_{\max}$ may be relatively restrictive. However, we respectfully argue that there are many existing works assuming such coverage assumption for the offline data to perform the policy optimization (see e.g., Antos et al., 2007; Munos and Szepesv ́ari, 2008; Chen and Jiang, 2019). In addition, one can relax this requirement by adopting the pessimistic principle for offline policy learning (e.g. Liu et al., 2020; Jin et al., 2021). Pessimism enables us to identify an optimal policy as long as the offline data-generating process covers the optimal policy (instead of any potential policy). We believe that pessimism can be incorporated into our algorithm to relax the conditions on $p_{\max}$. Nonetheless, since this is not the main focus of our paper, we will leave it as our future work.
> > >
> > >   For the memoryless assumption, first, we would like to argue that even under this assumption, the reward and future observations could still be confounded by the unmeasured confounders (and the posterior inference cannot uniquely identify the policy value in general). As long as one cannot fully recover the unmeasured confounders, super-policy learning is always helpful in obtaining a better policy. Second, this assumption is introduced to establish the finite-sample guarantees of our algorithm. It is not needed for policy value identification. Under this assumption, we show that the proposed fitted-Q-iteration type algorithm learns the super-policy efficiently with regret guarantees. To remove this assumption, one can alternatively develop a policy iteration algorithm by jointly optimizing all actions in the trajectory. However, in this case, we have to simultaneously solve T linear integral equations to identify Q-bridge functions for the policy evaluation, which is clearly computationally difficult for a large T. We will pursue this direction in our future work.
> > >
> > > - **The role of $\mathcal{G}$**. The class $\mathcal{G}$ serves not only as the space that performs the projection of Q-bridge functions, but also as the class of test functions in the min-max conditional moment estimation. For the purpose of performing the projection, the complexity of $\mathcal{G}$ is needed for nonparametric estimation. See e.g., Lemma B.1 in Liao et al. (2020) that the complexities of both the response function class and projection function class are needed. As for the min-max conditional moment estimation, the complexity of the class of test functions also plays a role in determining the convergence rate of the estimated Q-bridge functions.
> > >
> > >
> > >
> > > ### Reference
> > > Antos, A., C. Szepesv ́ari, and R. Munos (2007). Fitted q-iteration in continuous action-space mdps. Advances in neural information processing systems 20.
> > >
> > > Munos, R. and C. Szepesv ́ari (2008). Finite-time bounds for fitted value iteration. Journal
> > > of Machine Learning Research 9(May), 815–857.
> > >
> > > Chen, J. and N. Jiang (2019). Information-theoretic considerations in batch reinforcement learning. In International Conference on Machine Learning, pp. 1042–1051. PMLR.
> > >
> > >
> > > Liu, Y., A. Swaminathan, A. Agarwal, and E. Brunskill (2020). Provably good batch off-policy reinforcement learning without great exploration. Advances in neural information processing systems 33, 1264–1274.
> > >
> > > Jin, Y., Z. Yang, and Z. Wang (2021). Is pessimism provably efficient for offline rl? In International Conference on Machine Learning, pp. 5084–5096. PMLR.
> > >
> > > Liao, P., P. Klasnja, and S. Murphy (2020). Off-policy estimation of long-term average outcomes with applications to mobile health. Journal of the American Statistical Association, 1–10.

---

> > > > ### Author Response · Authors · 2022-11-22
> > > > **Please let us know if you have further concerns**
> > > >
> > > > Dear Reviewer ye4h,
> > > >
> > > > Thank you for your invaluable feedback. We were wondering whether our response and the revised manuscript addressed your concerns. If you have any additional comments, please let us know, we would be happy to address them.
> > > >
> > > > In this revision, in order to further demonstrate the superior performance of our method, we have conducted 3 simulation studies and 2 real-world applications. We believe the paper has been improved according to the reviewers' comments. Therefore we kindly ask you to consider raising your scores if the concerns were appropriately addressed.

---

> > > > ### Author Response · Authors · 2022-12-01
> > > > **Looking forward to your feedback**
> > > >
> > > > Dear Reviewer ye4h,
> > > >
> > > > Again, many thanks for your invaluable comments! Would you please let us know if our response and the revised manuscript have addressed your concerns? Feel free to let us know if you have any additional comments, and we would be very happy to further address them.

---

### Author Response · Authors · 2022-11-18
**Summary of key changes (1/2)**

We would like to thank the three reviewers for their helpful and constructive comments. We have revised our paper thoroughly according to their suggestions. All major changes in the updated paper are marked in blue. In addition to our detailed responses below, we summarize the major changes we have made in the revision here.

First, we have conducted extensive numerical experiments (including 3 simulation studies and 2 real-world applications) in this revision to illustrate the empirical performance of our proposed super-policy learning against existing RL algorithms that take either $S$ or a combination of $S$ and $Z$ as input for the optimal policy (denoted by Sonly and SZonly). For your convenience, we attach the results below.

- Two simulation studies in contextual bandits.

Tables 1 \& 2 show the average regret values for the tabular and continuous state settings respectively. It can be seen that when the observed action does not contain any information about the unobserved confounders ($\epsilon$= 0.5), the estimated super policy achieves comparable performance with other estimated policies. As the observed action contains more information (i.e., $|\epsilon - 0.5|>0$), the proposed super-policy achieves smaller regret than other baseline policies.

**Table 1**: Simulation results for the setting of contextual bandits with tabular variables under different choices of $\epsilon$. We replicate the simulation for 50 times. Mean regret values for estimated optimal policies under different policy classes are provided (and a smaller regret value indicates better performance). Values in the parentheses are the standard deviations of the regret values.
|                 | **Sonly**      | **SZonly**          | **Super**           |   |
|-----------------|----------------|---------------------|---------------------|---|
| $\epsilon = 0.5$ | 0.25 (3.1e-04) | **0.21**  (1.7e-02) | **0.21** (1.4e-02)  |   |
| $\epsilon = 0.7$ | 0.25 (3.1e-04) | 0.22 (1.8e-02)      | **0.18**  (3.5e-02) |   |
| $\epsilon = 0.9$ | 0.25 (2.5e-04) | 0.24 (1.2e-02)      | **0.17**  (8.6e-02) |   |

**Table 2**: Simulation results for the setting of contextual bandits with continuous variables under different choices of $\epsilon$. The rest is the same as Table 1.
|                 | **Sonly**      | **SZonly**          | **Super**           |   |
|-----------------|----------------|---------------------|---------------------|---|
| $\epsilon = 0.5$ | 0.40 (9.6e-04) | 0.14 (6.1e-03) | **0.12** (2.5e-03)  |   |
| $\epsilon = 0.7$ | 0.40 (9.2e-04) | 0.12 (5.9e-03)      | **0.11** (3.5e-03) |   |
| $\epsilon = 0.9$ | 0.40 (1e-03) | 0.11 (1.3e-02)    | **0.065** (1e-02) |   |

- One simulation study in the sequential decision making.

We use an existing simulation setting in the literature to illustrate the superiority of the proposed super-policy constructed based on the estimated Q-bridge functions. As Table 3 shows, the estimated super-policy yields significantly smaller regret values compared to the other two policies.

**Table 3**: Simulation results for the sequential decision making problem. See detailed description in Table 1.
|              **Sonly**      | **SZonly**          | **Super**           |   |
|-----------------|---------------------|---------------------|---|
| 5.4 (1.9e-01)  | 5.3 (4.7e-01)  | **2.2** (4.9e-01) |   |

- An application to the RHC data.

We use RHC data as an example of contextual bandits. We randomly divide the data into two parts. We use one part as the training data to learn optimal policies. The other part is used for evaluating the corresponding policies. As Table 4 shows, the super-policy produces higher policy values compared with the other two policies.

**Table 4**: Evaluation results of the optimal policies learned from three different policy classes using the RHC data. The averages of evaluation values over 20 random splits are presented. Larger values indicate better performances. Values in the parentheses are standard deviations.
|              **Sonly**      | **SZonly**          | **Super**           |   |
|-----------------|---------------------|---------------------|---|
| 0.55 (5.80e-02)  | 0.55 (5.78e-02) | **0.69** (1.10e-02) |   |

- An application to the MIMIC-III data.

The MIMIC-III dataset serves as an example of confounded POMDPs. We also adopt the idea of “random splitting” to evaluate different policies. As can be seen from Table 5 that our proposed super policy produces higher policy values compared with the other two policies.

**Table 5**: Evaluation results of the optimal policies learned from three different policy classes using the MIMIC-III data. See detailed description in Table 4.
|              **Sonly**      | **SZonly**          | **Super**           |   |
|-----------------|---------------------|---------------------|---|
| 0.55 (5.80e-02)  | 0.55 (5.78e-02) | **0.69** (1.10e-02) |   |

More details can be found in Sections C \& D of the revision.

---

> ### Author Response · Authors · 2022-11-18
> **Summary of key changes (2/2)**
>
>
> Second, in additional to the extensive numerical experiments, we also provide more details to demonstrate the applicability of our framework. We add more motivations in the introduction. For example, we include the example of deep brain stimulation (DBS) to motivate the plausibility of including observed actions in policy learning. We provide more discussion about the choices of proxy variables. Finally, a conclusion section is added to summarize our paper.

---

> > ### Author Response · Authors · 2022-12-06
> > **Response to the update (1/2)**
> >
> > Dear All Reviewers, Thank you so much for all your invaluable comments and suggestions. Reviewer FcTZ, many thanks for your second update and insightful comments! Based on reviewer FcTZ's second update, we believe the review team might have some misunderstandings about our paper which we would like to clarify below. Please let us know if you have any follow-up comments.
> >
> > - If we understand your **Item 2** correctly, we respectfully argue that we **did not** use the expert recommended action as the required proxy. The main claim of our paper is to use the observed action in the offline data as the input of the policy class for finding a policy with super optimality. The definition of the super-policy is **irrelevant** to the identification, thus independent of the proxy variables, as you can see in Section 3 up to the toy example.
> >
> >
> >
> >
> >     We would also like to emphasize that we mainly refer the **behavior policy as the “expert’s
> >     recommendation”**. Indeed, the observed action in our offline data does not need to be generated by the expert (or a policy close to the optimal one). As long as the behavior policy depends on the hidden information $U$, the super-optimality is guaranteed. For example, as seen in Table 1, when $\epsilon = 1$, the performance of the behavior policy ("expert policy") is very bad but our proposed sup-policy can still achieve global optimality. So the requirement on the behavior policy is mild. Please refer to our response " Clarification on the  "expert’s recommendation” " to Reviewer SFSz.
> >
> >
> > -  With regard to the role of expert recommendation in the healthcare examples, first, we would like to clarify that the expert's recommendation might not be necessarily the optimal one and there are applications where the RL agent produces more reliable treatment decisions (see e.g., Komorowski et al., 2018). Specifically, clinicians or doctors may not utilize all the important information (as opposed to RL agents) in treatment recommendations. For example, they may ignore some useful testing results, which they think are irrelevant. In addition, some doctors may not be experienced enough to offer the best recommendation. Therefore, the behavior policy is not always the optimal one. However, the super policy is guaranteed to perform *no worse* than the behavior policy (doctors’ recommendations). In other words, we can still outperform experts’ recommendations if there is room for improvement. If there is no, then the super-policy
> > performs the same as behavior one. Indeed, our proposal is to incorporate both human agents’ recommendations and patients’ personal information in real-world healthcare applications under unmeasured confounding, which we believe is extremely novel.
> >
> >
> > - Moreover, in some applications, the offline data can be generated from relatively “bad” human agents. For example, a student may have adopted a certain studying strategy in order to maximize her grade. Suppose this student, who is lack of experience, deter-
> > mines her studying strategy based on some hidden information such as efficiency (i.e.,
> > how much studying improves GPA) or her unique education experience. See more de-
> > tails on such hidden information in Harris et al. (2022) which considered a very different problem. As a result, she may not achieve a desirable grade using her own strategy.
> > Therefore the behavior policy (“expert policy”) can be relatively poor. However, even though the behavior is poor, it reflects the student’s hidden ability of studying, which is unseen in the data. Our super-policy can utilize this hidden information, and outperform both the behavior policy and the standard optimal policy which only depends on the observed state variables.
> >
> > ***
> >
> > **Reference**
> >
> > Komorowski, M., L. A. Celi, O. Badawi, A. C. Gordon, and A. A. Faisal (2018). The artificial intelligence clinician learns optimal treatment strategies for sepsis in intensive care. Nature medicine 24 (11), 1716–1720.
> >
> > Harris, K., D. D. T. Ngo, L. Stapleton, H. Heidari, and S. Wu (2022). Strategic instrumental variable regression: Recovering causal relationships from strategic responses. In International Conference on Machine Learning, pp. 8502–8522. PMLR.

---

> > > ### Author Response · Authors · 2022-12-06
> > > **Response to the update (2/2)**
> > >
> > > - Also, including the observed action in the policy class is a very new idea, we need to develop new identification results for policy learning *using the observational data*. We respectfully disagree that the identification results in our paper are basically translating the identification results in the proximal causal inference to the setting of sequential decision making. The existing identification results in the proximal causal inference are *only* designed for finding standard optimal policies that depend on S (and/or Z). The proposed super-policy additionally depends on the human recommendation A and its identification is extremely challenging since A always equals the selected action in the observational data, but we need to learn the value where the selected action differs from A for super-policy learning (which seems impossible). The proposed identification results in contextual bandits are already not trivial. Extensions to the sequential setting are even more challenging to develop. We have spent a lot of efforts and successfully achieved these seemingly impossible tasks under a number of scenarios. In addition, we have implemented 3 simulation studies and two real-world applications to demonstrate the potentials of our proposed super-policy.
> > >
> > > - Finally, just wanted to clarify the deep brain stimulation example. Even though a
> > > patient’s recommended action is available, as commented earlier, it might not necessarily be the optimal one. As the proposed super-policy would yield a better outcome, it
> > > makes more sense to employ the learned policy. Importantly, the clinicians shall communicate to the patients that their recommendations at runtime are needed as input to improve treatment decision making. The selected action will take their recommendation into consideration, but might not exactly equal their recommendation to possibly maximize their health.

---

### Decision · Program_Chairs · 2023-01-20

**Decision:**

Reject

**Justification For Why Not Higher Score:**

The general concern is a lack of clarity and motivation for abstract assumptions. See above.

**Justification For Why Not Lower Score:**

N/A

**Metareview: Summary, Strengths And Weaknesses:**

This paper considers a novel policy learning problem. One has access to past data generated by experts actions. The dataset logs covariates (or observed states), chosen actions, and rewards, but omits counfounders U which influence both potential outcomes and expert's actions. The goal is to learn a super policy which continues to observe some covariates and to observe expert recommendations, but can overrule the expert and pick any action it wishes. The question is scientifically interesting, though the immediate real world relevance is a bit unclear to me. (Usually, today, humans overrule algorithms. It is not clear also that humans would make recommendations the same way if they expected to be overruled).

*Strengths*:  The paper seems to propose a novel problem. They adapt the recent literature on proximal causal inference to formalize conditions under which their super RL agent succeeds, despite the presence of unobserved confounders. The paper initially contained no examples or empirical evaluation, but during the rebuttal period the authors ran some experiments (mostly on reusing past application of proximal causal inference) and placed those in the appendix.

*Weaknesses*: This problem studied here is quite delicate. If unobserved confounders do not heavily influence expert actions, then there is little additional value in super RL, relative to standard off policy evaluation. If unobserved confounders heavily influence both expert actions and potential outcomes, unbiased evaluation of counterfacutal actions requires delicate assumptions.

During the reviewer meeting, it became clear that *none of us* could articulate what problem features make this possible. The assumptions in the paper are quite abstract. I spent substantial time reading the proximal causal inference literature to try to understand them.  But I ultimately could not formulate a sensible sounding problem where the paper's various causal assumptions make sense AND observing expert recommendations in the future is quite valuable. The reviewers also carefully read the paper, looked at the updated examples in the appendix, and also cannot articulate the main idea.  It is hard to accept a paper when this is the outcome of peer review.

This may reflect a failing on the part of the review team. I'm sorry if that is the case. At the same time, the reviewers all work on related problems and read the paper with interest. Given this, it seems likely that the paper's lack of clarity will limit its impact and that it needs a substantial rewrite.

**Summary Of Ac-Reviewer Meeting:**

As discussed above, it quickly became clear during the review meeting that no one was able to understand the various causal assumptions in this paper and map them all to a plausible sounding problem. (I include myself in that).  This meeting took place after the authors had already added some empirical results to the paper trying to address this issue. The paper needs a substantial rewrite to improve its clarity and motivation, and I think this goes beyond the revisions that are natural in the rebuttal phase of ICLR. See the discussion above.